# 790,000 years of millennial-scale Cape Horn Current variability and interhemispheric linkages

Vincent Rigalleau [1] ✉, Frank Lamy [1], Nicoletta Ruggieri[1], Henrik Sadatzki [1,2], Helge W. Arz [3], Stephen Barker[4], Lester Lembke-Jene [1], Antje Wegwerth [3], Gregor Knorr[1], Igor M. Venancio [5], Tainã M. L. Pinho [1], Ralf Tiedemann [1] & Gisela Winckler [6,7]

Millennial-scale variations in the strength and position of the Antarctic Circumpolar Current exert considerable influence on the global meridional overturning circulation and the ocean carbon cycle. The mechanistic understanding of these variations is still incomplete, partly due to the scarcity of sediment records covering multiple glacial-interglacial cycles with millennial-scale resolution. Here, we present high-resolution current strength and sea surface temperature records covering the past 790,000 years from the Cape Horn Current as part of the subantarctic Antarctic Circumpolar Current system, flowing along the Chilean margin. Both temperature and current velocity data document persistent millennial-scale climate variability throughout the last eight glacial periods with stronger current flow and warmer sea surface temperatures coinciding with Antarctic warm intervals. These Southern Hemisphere changes are linked to North Atlantic millennial-scale climate fluctuations, plausibly involving changes in the Atlantic thermohaline circulation. The variations in the Antarctic Circumpolar Current system are associated with atmospheric $CO_2$ changes, suggesting a mechanistic link through the Southern Ocean carbon cycle.

The last glacial period (71–11.7 thousand years ago, ka) shows pronounced millennial-scale (-1–10 kyr) variability in ocean circulation and global climate, widely identified in Greenland ice cores[1], speleothems records[2], and marine sediment cores[3–5]. In the Southern Hemisphere (SH), the counterparts to the Northern Hemisphere (NH) abrupt warmings (so-called Dansgaard-Oeschger; DO events) are known as the Antarctic Isotope Maxima (AIM) and exhibit more gradual, smaller-amplitude changes[6,7]. For this glacial millennial-scale climate variability, the concept of a bipolar seesaw was established,

describing a thermal asynchrony between both hemispheres related to the dynamics of the global ocean overturning circulation[8–10]. Beyond the last glacial cycle and the stratigraphic range covered by Greenland ice cores, sediment records documenting millennial-scale climatic changes across several glacial cycles are mostly restricted to the NH[2,5,11–15], in comparison to only a few SH records[16,17].

The Antarctic Circumpolar Current (ACC), located in the Southern Ocean between 40° and 60°S is the largest current system on Earth[18]. It is driven by atmospheric forcing[19] including the Southern Westerly

[1]Alfred-Wegener-Institute, Helmholtz Center for Polar and Marine Research, Bremerhaven, Germany. [2]MARUM-Center for Marine Environmental Sciences, University of Bremen, Bremen, Germany. [3]Department of Marine Geology, Leibniz Institute for Baltic Sea Research Warnemünde, Rostock, Germany. [4]School of Earth and Environmental Sciences, Cardiff University, Cardiff, UK. [5]Programa de Geociências (Geoquímica), Universidade Federal Fluminense, Niterói, Brazil. [6]Lamont-Doherty Earth Observatory, Columbia University, Palisades, NY, USA. [7]Department of Earth and Environmental Sciences, Columbia University, New York, NY, USA. ✉e-mail: vincent.rigalleau@awi.de

Winds (SWW)[20], bathymetry, and ocean density gradients originating from surface, intermediate and deep ocean temperature and salinity changes[19]. Connecting the Atlantic, Pacific, and Indian Ocean basins, the ACC interlinks the various shallow to deeper southern water masses[18,19], thereby regulating the exchange with the global deep ocean[19,20]. The ACC is a crucial component in the global carbon budget[18] and exerts a major influence on the global uptake of anthropogenic heat and carbon dioxide[21]. The major bathymetric constriction of the ACC occurs at the Drake Passage (DP). Complementing the so-called warm-water route that connects the Indian and Atlantic oceans (i.e., Agulhas leakage), the cold-water route connects the Pacific and Atlantic oceans through the DP. Over the last glacial period, substantial millennial-scale fluctuations in DP throughflow[22] and in the Agulhas leakage[23], as well as Southern Ocean warming[24,25] have been suggested as possible triggers of DO events, through regulating the Atlantic Meridional Overturning Circulation (AMOC). Therefore, it is crucial to obtain a more comprehensive understanding of millennial-scale dynamics in the Southern Ocean in order to assess its role within the climate system. Except for Antarctic ice core records[17], longer paleorecords with millennial-scale resolution covering multiple glacial cycles are not yet available from the SH. Consequently, our knowledge about the presence and recurrence of such SH millennial-scale variability in earlier parts of the Earth's Quaternary history is limited.

Here, we present high-resolution sedimentological and geochemical records obtained from International Ocean Discovery Program (IODP) Site U1542 (52°42.29′S, 75°35.77′W; 1,101 m water depth, Fig. 1)[26]. The site is located ~30 nautical miles off the Chilean coast within the central part of the Cape Horn Current (CHC), which flows along the Chilean continental margin towards the DP[27–29]. The CHC originates from the bifurcation of subantarctic water masses of the South Pacific Current (SPC) approaching the Chilean coast around ~45ºS[28,29] (Fig. 1, Supplementary Fig. 1). The CHC is ~100–150-km wide, flows along the Chilean margin southward parallel to the Northern

Boundary (NB[30]) of the ACC, and merges with the main ACC flow close to Cape Horn[27,29] (Supplementary Fig. 1). The southern CHC is considered as part of the ACC system[27], whereas the northern section is partly influenced by lower latitude forcings[27,31].

The ~250-m-long composite record covers the past 790 kyr with an average sedimentation rate of ~30 cm/kyr (see methods for details on the age model). During glacial periods, enhanced hinterland discharge contributed to an increased supply of terrigenous sediment, resulting in a bulk accumulation rate of up to five times higher compared to interglacial periods, consistent with earlier records from the region[26,32]. We focus on reconstructions of sea surface temperature (SST) using alkenone palaeothermometry[14,32–35] and near bottom current strength based on the X-ray fluorescence core scanner data (zirconium to rubidium ratio; Zr/Rb) calibrated with sortable silt data[4,22,36,37]. Both SST and current strength reconstructions have been successfully applied in that region[4,22,33,37]. Our study extends the existing sediment records from the nearby previous piston core MD07-3128 that reach back to ~60 ka[4,26,33]. Together, these records allow to explore orbital and millennial-scale climate variability of the CHC at the entrance of the DP across the Late Pleistocene in unprecedented detail.

## Results and discussion
### SST and current strength reconstructions at the southern Chilean margin

We obtained alkenone-based SSTs from 929 samples at an average temporal resolution of ~700 years (Fig. 2c). Holocene SSTs reach up to ~11.5 °C during the early Holocene, and gradually decrease to ~9.6 °C during the late Holocene. This is ~1.5 °C above the modern mean annual SST attributed to a likely seasonal bias of the alkenone-derived SSTs at the southern Chilean margin[38]. Over the past 790 kyr, the reconstructed SST at Site U1542 range from ~3 °C during the Last Glacial Maximum (LGM; sensu lato 18–28 ka), Marine Isotope Stage (MIS) 7 d (~275 ka), MIS 11b (~390 ka) to more than 12 °C during MIS 5e,

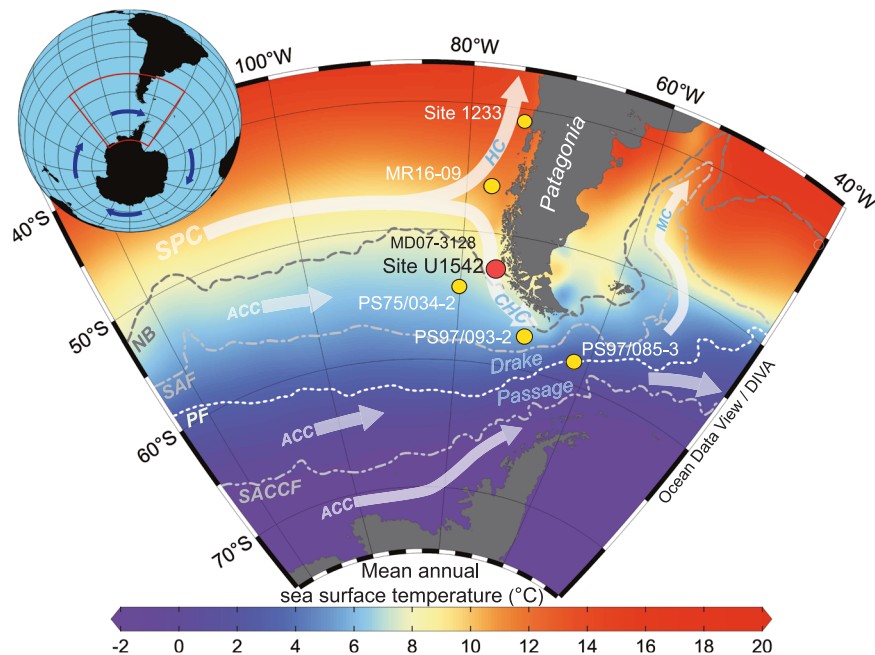

**Fig. 1 | Regional modern ocean hydrography and location of cores discussed in the study.** Map of the Drake Passage region with mean annual sea surface temperature from the World Ocean Atlas (based on 2005–2017 average observations)[75]. Yellow dots mark sediment core locations and the red dot indicates the location of the main record introduced in this study. MD07-3128 is located at the same location as Site U1542. White transparent arrows are schematic representations of major surface currents; the Antarctic Circumpolar Current (ACC), the South Pacific Current (SPC), the Cape Horn Current (CHC), the Humboldt Current (HC), and the Malvinas Current (MC). Dashed lines represent altimetry-derived ACC fronts[30]; Northern Boundary (NB), Subantarctic Front (SAF), Polar Front (PF), and Southern ACC front (SACCF). Maps were created in Ocean Data View[76].

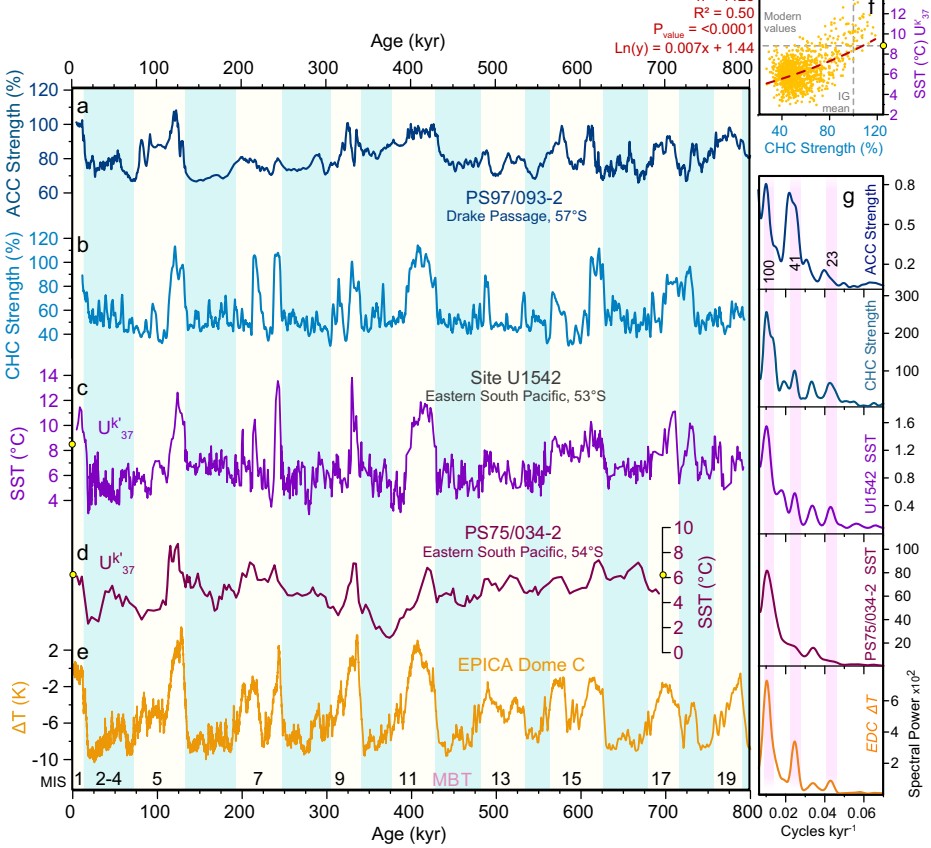

**Fig. 2 | Orbital-scale variability of sea surface temperature in the eastern South Pacific and current strength. a** Sortable silt record from sediment core PS75/093-2 representing Antarctic Circumpolar Current (ACC) strength changes at the entrance of the Drake Passage[37]. **b** Cape Horn Current (CHC) Strength. **c** Alkenone-derived sea surface temperature (SST) from Site U1542, the yellow dots indicate the modern SST at the core location). **d** Alkenone-derived SST from core PS75/034-2[34]. **e** Antarctic ice core EPICA Dome C temperature record[17] on the AICC2012 age model[73]. **f** Correlation between CHC strength and every alkenones-derived SST measurement from Site U1542. **g** Respective spectral power of (**a**) to (**f**). Timing and nomenclature of Marine Isotope Stages (MIS) follow Lisiecki and Raymo[77] and glacial periods are blue shaded. MBT corresponds to the Mid-Brunhes Transition[55].

MIS 7e and MIS 9e (Fig. 2). The mean glacial to interglacial (G/IG) temperature difference at glacial terminations is ~6 °C from 0 to 430 ka and ~4 °C before 430 ka. The transitions into glacial periods are marked by strong and abrupt coolings ranging from ~3 to -8 °C. We observe persistent and high-amplitude SST variability of ~1–3 °C on millennial timescales during glacial periods across all eight glacial periods recorded at Site U1542.

To assess the relationship between our SST record and the strength of the CHC, we reconstructed the near-bottom current speed using an extension of the classical sortable silt proxy[36], which includes the fine sand fraction (sortable silt/fine sand or SSFS[22], see methods and Supplementary Fig. 4). The calculated flow speeds depend on the sensitivity of the grain size to the bottom-current flow speed which may partly depend on local conditions such as bathymetry and sea-floor morphology. Site U1542 is located on the upper continental slope within small-scale sediment drift[26] and documents a sedimentary sequence consisting mostly of siliciclastic sediments. Ice-rafted debris (IRD) supply has been reported at the site (core MD07-3128) for the last glacial period[33] when the western Patagonian ice sheet (PIS) extended to the continental shelf in southern Chile[39,40]. High siliciclastic sediment accumulation rates at Site U1542 dilute the IRD supply and thus do not substantially affect the SSFS[41]. This is supported by grain size analyses highlighting the absence of correlation and therefore independent fluctuations between the mean SSFS and the size-fraction commonly mainly dominated by IRD (Supplementary Fig. 4e). Supplementary Fig. 4b shows that mean SSFS and the weight percentage of the SSFS component are positively correlated, providing strong evidence for primarily current-controlled grain size changes within the silt-fine sand fraction[42]. Downslope processes may also exert some influence on fine grain size distributions during peak glacials intervals, when finer grain sizes might be influenced by glaciofluvial sediment supply from the proximal PIS (Supplementary Fig. 4f). We use the discrete SSFS measurements from grain-size analyses for calibrating the XRF-based Zr/Rb count ratio in order to obtain high-resolution CHC strength records. The excellent correlation between SS and Zr/Rb ($R^2 = 0.80$) indicates that both are reliable indicators of bottom current speed at the southern Chilean margin[4] and the DP (Supplementary Fig. 4a).

Our current strength reconstruction shows high values during the Holocene, interglacial MIS 5, 7, 9, 11, and 15 with mean SSFS values of ~50 μm, corresponding to a mean current velocities of ~16.5 cm/s (Supplementary Fig. 4i)[41]. Maximum velocities of ~20 cm/s occur during peak interglacials and correspond to 120% of the interglacial mean (Fig. 2b). MIS 13 and MIS 17 show weaker bottom currents of ~42 cm/s (80%, Fig. 2b). CHC strength was significantly reduced during glacial (<10 cm/s, translating to ~50–60% weaker flow compared to the mean interglacials (Fig. 2b). This reduction is similar to previous estimates in the Chilean margin[4] and in the central DP at the Polar Front during the last glacial period[22], and in the central South Pacific[43] (Supplementary Fig. 5). However, less reduction is observed at the Pacific entrance of the DP based on deep ocean site PS97/093 (6–16% reduction, Fig. 2a)[37]. Similar to the SST changes, the Site U1542 CHC strength reconstruction exhibits pronounced millennial-scale variability with amplitudes of ~2 to 6 cm/s, that persists across all glacial stages. Altogether, our

SST and CHC strength reconstructions underline the exceptional palaeoceanographic sensitivity of the Chilean margin and provide a unique opportunity to explore in detail millennial-scale changes of the SH during G/IG cycles.

## Sensitivity of the eastern South Pacific to orbital and millennial-scale climate variability

The SST record at Site U1542 shows dominant spectral power at the eccentricity (100-kyr) band. Additionally, small amplitude spectral peaks occur at the obliquity (41-kyr) and precessional (23-19-kyr) bands (Fig. 2g). Though overall spectral power at the common orbital cyclicities is similar in both the Site U1542 SST record and the Antarctic EPICA Dome C ice core (EDC) temperature record, a direct comparison of the records reveals substantial differences. For example, the U1542 SST record shows prolonged warming trends throughout MIS 6 (-2 °C between early glaciation to termination), MIS 10 (-2 °C), and MIS 12 (-1 °C), which are not documented in the EDC record.

Comparable trends are likewise evident in the core PS75/034-2 SST record[34] and in several other subantarctic SST records[16,44,45] (Supplementary Fig. 6). However, the U1542 SST variations show overall warmer temperatures (-2–3 °C) and higher amplitude glacial-interglacial changes (2–5 °C compared to 4–6 °C) than recorded at site PS75/034 (located outside the CHC ~ 200 nautical miles offshore U1542) (Fig. 2c, d; Supplementary Fig. 6)[34]. These patterns reflect the warmer SSTs in the CHC and its larger variability in the past. Also occurring at site GeoB3327 (~43°S) in the Eastern South Pacific[34], the common warming trend observed during several glacial stages in the subantarctic Southern Ocean is closely aligned with the 100-kyr amplitude modulation of precessional variations at low latitudes, implying that these trends are a direct response to low-latitude insolation forcing by eccentricity[34,46] (Supplementary Fig. 6).

Changes in reconstructed CHC strength and SST at Site U1542 covary on G/IG timescales during the last 790 kyr (Fig. 2b, c). Increased CHC strength parallels warm SST during interglacial periods, while reduced flow speeds occur during glacial periods with colder SST. The overall direction of CHC strength changes across G/IG is similar to the reconstructed ACC variations in the central DP (core PS97/085-3[22]), at the Pacific entrance of the DP (core PS97/093-2[37]), and in the Central South Pacific (IODP Sites U1540/U1541[43]). These similarities show that large-scale ACC changes accompany the G/IG variations of the CHC suggesting common forcings. Overall, interglacial maxima in our records stand-out more prominently and initiate more abruptly from the overall glacial background compared to deep-ocean sites. These patterns are most likely related to thresholds in the CHC response to G/IG climate changes, involving e.g., sea-level and the extent of the PIS.

Site U1542 is presently located in the central CHC. Recent remote sensing-based oceanographic studies define the CHC as poleward flowing from ~50°S towards Cape Horn, where it merges with the Pacific ACC[28,29,31]. While the comparatively weak northern part of the CHC is mainly driven by pressure gradients through sea-level changes from oceanic waves propagating from the low latitudes, the stronger southern CHC is intimately linked to deep-ocean processes including ACC eddy activity[31]. It has been suggested that sediment records (i.e., sortable silt) generally correspond to the total water transport including wind-driven, barotropic, and eddy-induced transport[41,43]. However, based on the proxy data, it is impossible to distinguish the modern oceanographic processes in more detail on longer geological time scales.

The above-average CHC strength during various interglacials indicates enhanced influence of the overall stronger Pacific ACC[43] on Site U1542. Additionally, low latitude forcings (e.g., Coastally-Trapped Waves propagated from the equator[31]), might have been enhanced during strong interglacials. However, their influence on the bottom water current strength would be minor as the modern northern CHC is much weaker than the more ACC-influenced, stronger southern

section. Conversely, we interpret the weaker glacial CHC as indicative of a generally weaker ACC system in the Pacific sector[43] and, at the same time less impact of low-latitude forcings. Today the CHC reveals pronounced seasonal changes[29]. During austral fall/winter, the southward meridional transport as characteristic for the CHC, extends several degrees latitude further north and retreats southward during austral spring/summer. These large seasonal changes are connected to the seasonal migration of the South Pacific Gyre and are today related to the Southern Annular Mode[47] and thus the SWW. On paleoceanographic time-scales, these seasonal changes imply that a northward extension of subantarctic ACC waters probably connected to a stronger SPC, which would be consistent with a northward migration of the Chilean bifurcation during cold periods. Conversely, relatively warm periods would be characterized by a poleward shift of the ACC, SPC and westerly wind circulation. These analogs would serve both for orbital-scale (G/IG) as well as shorter millennial-scale variations discussed in this paper.

Glacial trends, however, diverge during MIS 2 and 8, showing long-term cooling coinciding with CHC current strengthening. This strengthening might relate to regional changes within the CHC, as the glacial trends were neither observed in the DP[22,37] nor in the central South Pacific[43] (Supplementary Fig. 5). At present, the core of the CHC is located close to the shelf break[27], but could have been shifted slightly offshore due to the PIS reaching the continental slope during the last glacial period[39]. Moreover, a lower sea level likely also deflected the CHC further offshore, thus enhancing its strength at Site U1542 but not at the deep ACC sites. Notably, these long-term changes do not influence millennial-scale, but only orbital-scale variability and trends.

The relationship between the reconstructed SST and CHC strength extends also to shorter timescales, for example for several major AIM (1, 4, 8, and 12) during the last glacial period, suggesting a connection between both orbital and millennial timescales (Fig. 3e, f). Our CHC strength record mirrors the current record from sediment core PS97/085-3[22] for the last glacial (Fig. 3d). This suggests that the CHC strength closely resembles (Fig. 3d, e), the northern subantarctic ACC entering the DP on millennial-scales. In the DP, geostrophic current velocities are highest in the vicinity of the Subantarctic and Polar Front[48], and latitudinal shifts of the fronts are linked to temperature changes[19]. This supports the idea that a reduction in the strength of the overall DP throughflow is linked to the northward shift of the Southern Ocean frontal system during glacial times[4,22,37]. Furthermore, Wu et al.[22] found a correspondence between millennial-scale maxima in the DP throughflow strength and major winter sea-ice retreat in the Scotia Sea. This potential link is consistent with our records, as both SST and CHC strength show similar orbital-scale variability as the sea-salt sodium (ssNa) record from the EDC ice core (Supplementary Fig. 7), a proxy for sea-ice extent and atmospheric circulation changes[49]. The sea-ice retreat is coupled to a strengthening and southward shift of the SWW, acting as a positive feedback mechanism that amplifies millennial-scale strengthening of the CHC flow speed[20,22].

High millennial-scale variability is also recorded in in two SST records from the Chilean margin further north, core MR16-09PC03 (~45°S) and ODP Site 1233 (~41°S) (Fig. 3)[32,40]. Like our Site U1542 SST record, suborbital SST changes range from -2 to 3 °C during the last glacial period, showing consistent timing and amplitude. Located north of Site U1542, ODP Site 1233 and core MR16-09PC03 suggest a stronger northward deflection of the SPC and ACC into the Humboldt Current system (i.e., the South Pacific Gyre) during cold periods, resulting in substantial millennial-scale variability. Taken together, these results support large-scale changes in the northward extent of the ACC and SPC, involving atmospheric variations of the SWW as important drivers for millennial-scale variations at Site U1542[4].

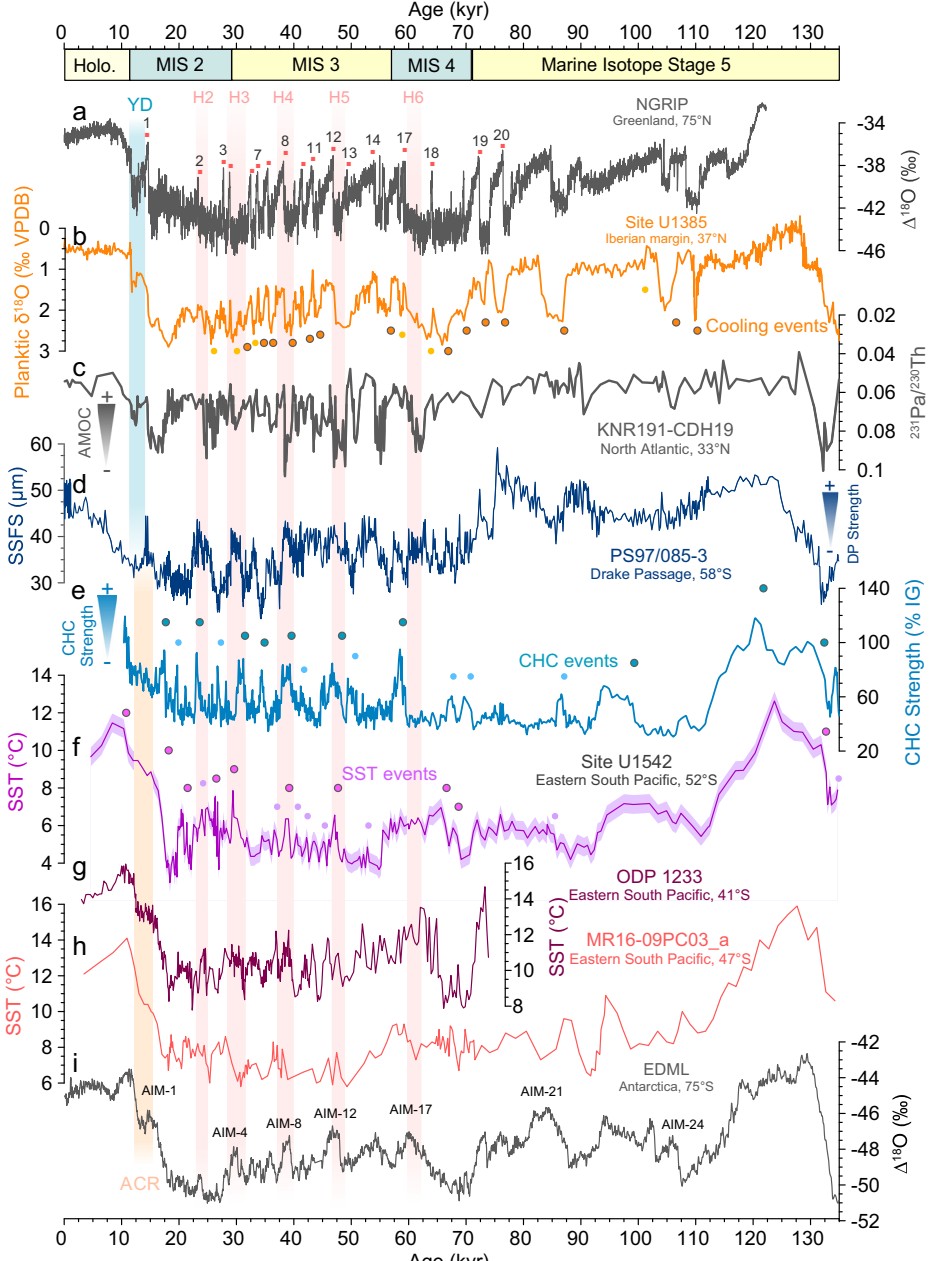

**Fig. 3 | Interhemispheric linkages during the last Glacial Period. a** Greenland climate reconstruction[7,78] recording millennial-scales abrupt events, called Dansgaard-Oeschger events (red dots). **b** Planktic δ18O from North Atlantic[13], taken as proxy for sea surface temperature (SST) changes. **c** Compilation of Pa/Th as a proxy for the Atlantic Meridional Overturning Circulation (AMOC) strength[54]. **d** Grain size-based strength of Drake Passage throughflow reconstruction[22]. **e** Cap Horn Current (CHC) strength. **f** SST with uncertainty envelope (0.5 °C; see methods) at Site U1542. **g** SST from ODP Site 1233[4]. **h** SST from MR16-09[40]. **i** Antarctic climate reconstruction at EPICA Dronning Maud Land site[7] (EDML) on the AICC2012 age model. YD Younger Dryas. ACR Antarctic Cold Reversal. H Henrich events. AIM Antarctic Isotopic Maxima. MIS Marine Isotope Stage.

## 790,000 years of Southern Hemisphere millennial-scale variability

We used a threshold detection approach to distinguish the occurrence of climatic events on the SST and CHC strength records at Site U1542 (Supplementary Fig. 8). A SH event is defined in our record by an abrupt CHC strengthening (108 events identified in total) or SST warming (103 events) (see method, Supplementary Fig. 8). We classified each event by defining two categories based on the distribution of their amplitudes (Supplementary Fig. 8g, h, i). We identified 66 major SST events characterized by a warming exceeding 1.6 °C and 51 major CHC strengthening events displaying a strengthening greater than 3.6 cm/s (corresponding to 24% of

interglacial value, see Method, Supplementary Fig. 8). Not all strengthening events in the CHC are necessarily associated with an SST warming event, as SST events tend to occur only with the stronger CHC events.

This partly different behavior of the bottom currents and SST within the CHC might be due to varying millennial-scale sensitivities and thresholds for SST and CHC strength changes. While both the SST and the current strength within the CHC are related to large-scale atmospheric and oceanic circulation changes (as discussed for the G/ IG variations), their individual response to ACC changes and low latitude forcings might vary. For example, SST changes might be more strongly linked to the advection of lower latitude water masses[28],

whereas the bottom water strength changes at our site are more strongly influenced by the ACC.

Although the amplitude of associated warming and strengthening events appear to be independent of each other (Supplementary Fig. 9), the identification of a millennial-scale event in both SST and CHC strength records, particularly for major events, serves as a robust indicator of climate dynamics that can be related to both oceanic and atmospheric circulation. For instance, during the last glacial period, we observe a major increase of flow strength (increasing from 50% to 80% of the interglacial mean) co-occurring with a temperature rise of ~2 to 3 °C toward several major AIM events (i.e., AIM 4, 8, 12, and 17). We found 57 (46 related to millennial-scale events) events monitoring a CHC acceleration concomitant with SST warming within a timeframe shorter than 2 kyr.

Millennial-scale SST and CHC strength events at Site U1542 often occur in the absolute SST range from ~4 to 6 °C and 35 to 55 % of interglacial CHC strength (Supplementary Fig. 9b, c), corresponding to intermediate glacial periods (Fig. 2)[17]. This suggests that the CHC exhibited an enhanced sensitivity to climate during an intermediate climate state (i.e., transitional periods leading to full glacial conditions) or that events are larger during these periods, consistent with findings from NH records[5,11,13,15]. Several studies have suggested that a prolonged intermediate climate state, such as MIS 3, provides favorable conditions for high amplitude DO-type variability[13,50]. During full glacial boundary conditions (e.g., LGM), NH records suggest a relatively stable climate with reduced millennial-scale variability[5].

For both SST and CHC strength records, the frequency of millennial-scale events recorded at Site U1542 seems to follow a stochastic pattern, lacking any discernible cyclic behavior (Supplementary Fig. 10). The recurrence of the millennial-scale warming events slightly increases after the Mid-Brunhes Transition (MBT)[51] (Supplementary Fig. 9d, e), likely due to an increase of glacial period duration. This implies that the magnitude of millennial-scale events, particularly SST events, changes with background climate and that the MBT affects climate variability from orbital to millennial timescales. In contrast, the amplitude of millennial-scale events is smaller during warmer periods, consistent with the relative stability observed during extended interglacial periods in the Northern Hemisphere[5,13,15].

## A persistent interhemispheric teleconnection

Over the last glacial period, SST records from the eastern South Pacific (ESP) region have been shown to reveal an "Antarctic timing" of millennial-scale temperature patterns[33,52,53], i.e., the SST pattern follows the Antarctic temperature reconstruction known from Antarctic ice cores. Millennial-scale climate events at Site U1542 are found to be contemporaneous with several AIM events[7] (Fig. 3). Additionally, our SST and CHC strength records is consistent in timing and amplitude with a Southwest Pacific Mg/Ca SST record resolving millennial-scale changes and spanning the past three glacial cycles (Fig. 5f; Supplementary Fig. 12)[16]. This suggests that the SST changes in both records represent surface changes of the wider subantarctic Southern Ocean. In addition, the timing of our millennial-scale climate events coincides with that recorded in DP sediment core PS97/085-3[22] over the last glacial period (Fig. 3d). This record further reveals that the ACC accelerated during Antarctic warming events, in parallel with the weakening of the AMOC during Heinrich Stadials[9] in the NH, as indicated by high $^{231}Pa/^{230}Th$ ratios[54] (Fig. 3c). According to the bipolar seesaw concept[8], NH stadial events are expected to be associated with SST warming in the SH, and ACC strengthening events as observed during the last glacial period[22] (Fig. 3b). These comparisons over the more recent G/IG cycles suggests that our U1542 records are consistent with the interhemispheric timing predicted by the bipolar seesaw.

To robustly assess interhemispheric connections across the past 790 kyr, we identified cooling events at Site U1385 (located at the Iberian margin in the North Atlantic)[13], a high-resolution record

spanning the last million years. The planktic $\delta^{18}O$ signal from Site U1385 primarily reflects surface temperature conditions and is an indicator of NH millennial-scale surface water changes. By applying the same thresholding approach used for Site U1542, we identified 110 NH cooling stadial events (69 major events) at Site U1385 over the past 800 kyr, consistent with the findings of Hodell et al.[13]. We subsequently compared amplitude and number of events per glacial cycle between the records to mitigate age model uncertainties.

The interhemispheric comparison reveals similarities between both records in the amplitude and number of events per glacial cycle (Fig. 4). For instance, the average amplitude of events recorded during MIS 8 and MIS 9 (243–337 ka,) shows the highest values, gradually decreasing during the last two glacial cycles. These similarities are less evident for the older glacial cycles. For instance, during MIS 12 and 13, the average CHC strength amplitude is higher than in the NH record. likely because of the prominent CHC event at 490 kyr. The number of events per 10 kyr period for each glacial cycle (Fig. 4b) is also similar for both records. This includes the number of cooling events in the NH being strongly correlated with the number of CHC strengthening events for the last 621 kyr, while it shows a moderate correlation with the number of SST warming events. The interhemispheric disparity in several cycles likely arises from the enhanced local sensitivity, notably influenced by the presence of the PIS that reached the continental shelf edge at glacial maxima[39]. During the LGM, records from the Chilean margin and DP depict higher amplitude millennial-scale events (Fig. 3d–h) compared to NH records (Fig. 3a–c). In both sediment records from Site U1542 and U1385, we observe an intensification of amplitudes and recurrence times of millennial-scale events after the MBT (Fig. 4). This suggests that the MBT, induced by changes in insolation[55], not only amplified the G/IG signal[51] but also influenced the climate on the millennial timescale.

At Site U1385, a notable out-of-phase relationship between benthic and planktic oxygen isotopes has been documented, representing a southern signal in deep waters versus a northern signal in surface waters within a single record[56]. This out-of-phase pattern aligns with Antarctic and Greenland climate records, respectively, and illustrates the bipolar seesaw mechanism, extended over the mid-to-late Pleistocene[13,56]. We find millennial-scale features of SST and CHC strength at Site U1542 matching the North Atlantic benthic $\delta^{18}O$ record, within age model uncertainties (Supplementary Fig. 12). For instance, during the penultimate glacial period, SST and CHC strength millennial-scale climate events at Site U1542 are in phase with benthic $\delta^{18}O$ fluctuations observed in the Iberian margin sediment core (referred to as AIM6i to AIM6vi, between 180 and 155 ka, Supplementary Fig. 12a)[56], themselves synchronous with AIM events[56]. Expanding upon these observations, we find that several major surface warming and CHC strengthening episodes recorded at Site U1542 can be associated with negative benthic $\delta^{13}C$ incursions in the North Atlantic at Site U1308 (Fig. 5b, Supplementary Fig. 12), and/or increased IRD concentrations at ODP 983 (Fig. 5a, Supplementary Fig. 12): both parameters associated with a reduced AMOC during these events[11,12]. This suggests that these important variations of DP throughflow that have been linked to AMOC instability over the last glacial cycle[22], persisted over the past 800 kyr. Furthermore, this one-to-one relationship reinforces previous findings highlighting strong interhemispheric climate linkages on orbital and millennial time scales[25,57].

In addition to these oceanic mechanisms for interhemispheric climate linkages related to the bipolar seesaw, atmospheric teleconnections have been proposed to operate, at least for the last glacial cycle. For example, strong millennial-scale SST warming during Heinrich event 1 at Site 1233 has been linked to southward displacements and/or strengthening of the SWW induced by a southward shift of the Intertropical Convergence Zone[58]. This atmospheric interhemispheric teleconnection would plausibly affect both SSTs and CHC strength, as the CHC is also connected to the migration of the South Pacific Gyre.

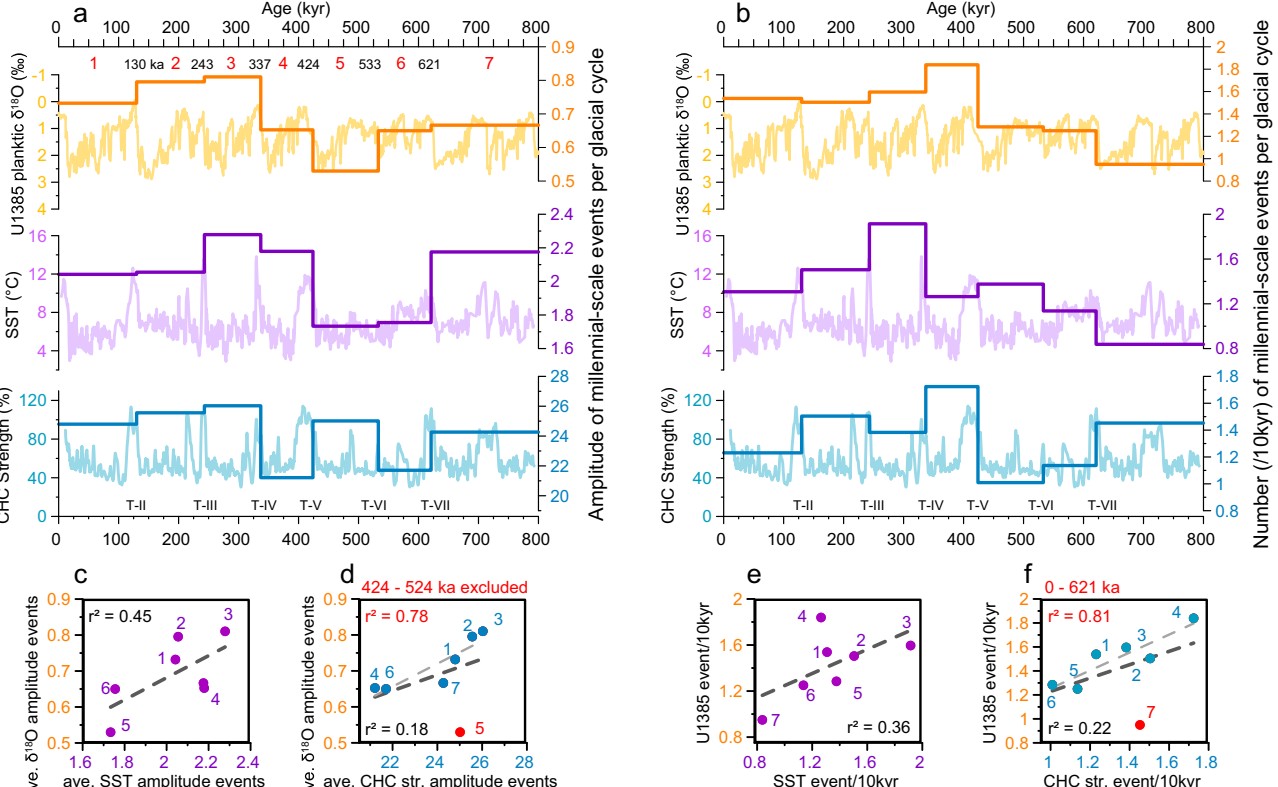

**Fig. 4 | The amplitude and number of millennial-scale events for each glacial cycle.** Comparing all events within a single glacial cycle reduces the impact of age model uncertainties between the two locations, as glacial terminations are marked abrupt and distinct patterns, making them more reliable indicators (original data are shown in background). **a** Average amplitude of stadial events at Site U1385 (orange), sea surface temperature (SST) warming events (purple), and Cape Horn Current (CHC) strengthening events (blue) at Site U1542 for each glacial cycle, highlighting a correlation between the amplitude in one hemisphere and that in the other. A high variability observed in one hemisphere during a glacial period often

corresponds to high variability in the other hemisphere. **b** Number of events per 10 kyr for each glacial period. For example, during the penultimate glacial cycle (130–243 ka), there was an average of 1.5 CHC events, 1.5 SST warming events, and 1.5 cooling events in the northern hemisphere per 10 kyr. An interhemispheric correlation between the average amplitude (**c, d**) and the number (**e, f**) of events for each glacial cycles shows that glacial cycles with higher event frequencies or amplitude in one hemisphere tend to have similarly high frequencies or amplitude in the other hemisphere, as shown in (**c–f**). Number on the figures **c–f** refers to the red number in (**a**) and (**b**) indicating the glacial cycle.

Studies of anthropogenic climate change using both models and observations detect an acceleration of averaged zonal flow on the northern flank of the ACC and identify anthropogenic ocean warming as the dominant driver[59]. Moreover, considering the widely anticipated weakening of the AMOC in response to anthropogenic warming[60,61], state-of-the-art climate models seem to reinforce the idea of the teleconnection between Southern Ocean conditions and circulation in the North Atlantic. Although occurring on a shorter time scale, this corroborates the fundamental physical relationship between SST, CHC/ACC strength, and ocean circulation that form the baseline of this study.

### Role of ACC in $CO_2$ exchanges

The Southern Ocean modulates the exchange of $CO_2$ between the deep sea and the atmosphere[19,20]. The latitudinal shifts of the SWW exert control over the balance between the biological carbon pump, which sequesters carbon into the deep ocean through the sinking of biological carbon from the surface ocean[62,63] and the release of $CO_2$ from ventilation of deep water upwelling in response to surface ocean stratification[60,64,65]. The mechanistic relationship between surface ocean conditions and atmospheric $CO_2$ changes is closely tied to shifts in the SWW, a major driver of $CO_2$ upwelling[64], as well as ACC strength[4,22].

To evaluate the role of ACC strength variability in driving atmospheric carbon dioxide ($CO_2$) variability, we compare the rate of atmospheric $CO_2$ changes smoothed over a 2-kyr-period (Fig. 6)[11] with

our current strength reconstruction that mirrors both orbital- (Fig. 2a, b) and millennial-scale (Fig. 3d, e, Supplementary Fig. 5) changes in the northern, subantarctic CHC/ACC entering the DP. We found 57 atmospheric $CO_2$ release events, in the same order as major CHC strengthening events (51). Millennial-scale CHC strengthening events range from 24% to 50% (Fig. 6a), while strengthening related to glacial terminations reaches 70%. Similarly, atmospheric $CO_2$ release events range from 5 ppmv to ~35 ppmv (Fig. 6b), while atmospheric $CO_2$ release related to glacial terminations reaches 80 ppmv. Glacial terminations present higher $CO_2$ rise compared to millennial-scale events, likely because other processes than CHC/ACC changes affect $CO_2$ rise during glacial terminations. Although taking into account of age model uncertainties, we associated 31 (10 related to glacial terminations) atmospheric $CO_2$ release events with CHC strengthening events occurring within the moving 7-kyr windows (Fig. 6c, Supplementary Fig. 11), suggesting that the relative changes in flow strength appear to be positively correlated to the amplitude of $CO_2$ variations (Fig. 6c).

Ahn and Brook[66] and more recently Barker et al.[11] have observed a relative shoaling in the North Atlantic deep-water formation as well as an increase in IRD deposits in the North Atlantic during phases of rising $CO_2$ (Fig. 5a). An AMOC disturbance induces a change in the meridional heat transport resulting in heat being retained in the Southern Hemisphere, which can then melt sea-ice[65] and/or cause a southward migration of the Intertropical Convergence Zone[67]. This can result in a strengthening and southward displacement of the SWW, enhancing

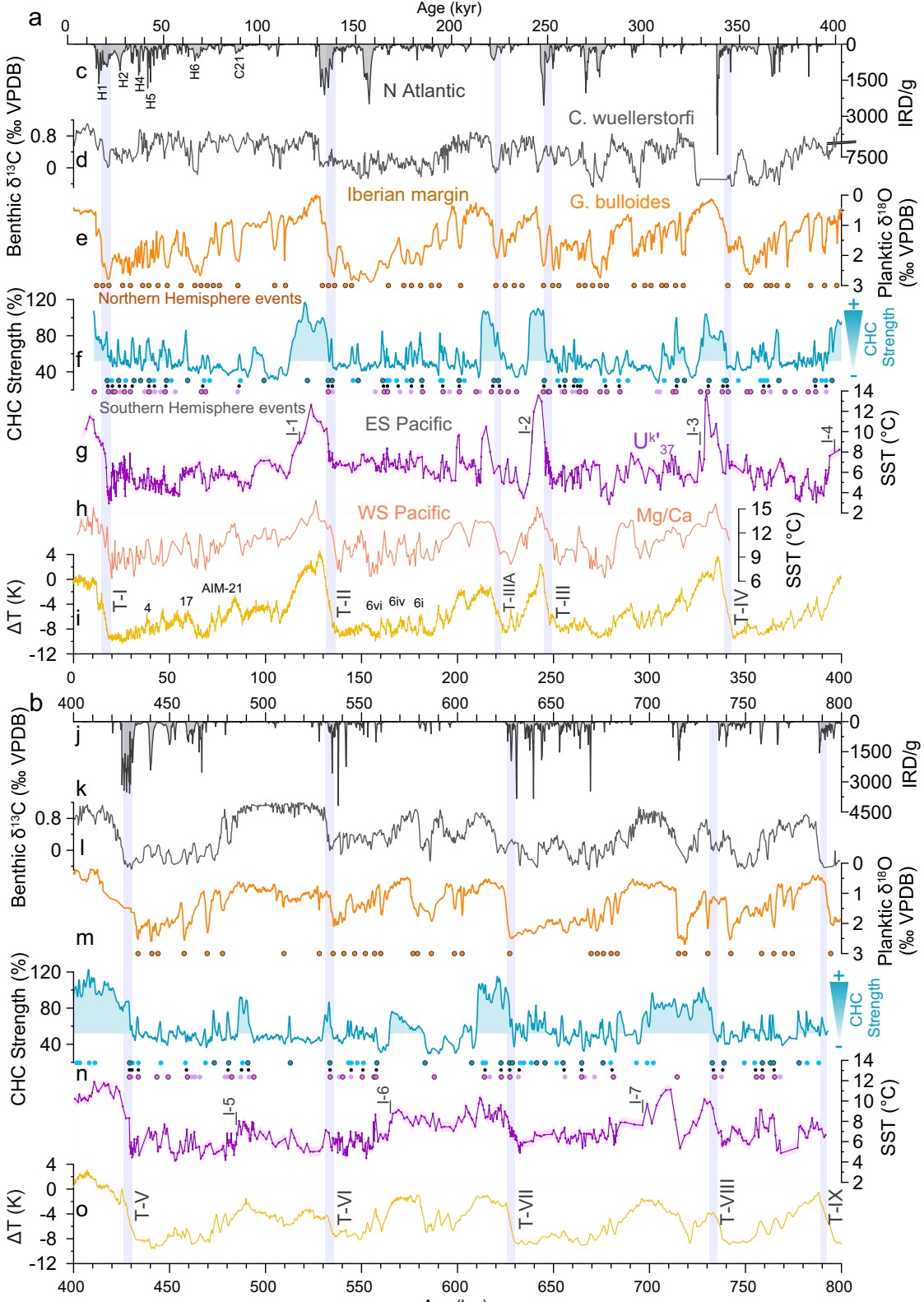

**Fig. 5 | Millennial-scale climate records of the past 800 kyr. a** Climate reconstruction from 0 to 400 kyr. **b** Climate reconstruction from 400 to 800 kyr. **c, j** Ice-rafted debris at ODP Site 983[11]. **d, k** Benthic δ13C from Site U1308 indicating the mixing ratio between northern and southern sourced waters[12]. **e, l** Planktic δ18O from Site U1385[13] taken as proxy for sea surface temperature (SST) changes. **f, m** Cape Horn Current (CHC) strength. **g, n** SST from Site U1542 with the uncertainty envelope (0.5 °C; see methods) (this study). **h** Mg/Ca-derived SST from the Southwest Pacific[16]. **i, o** Antarctic ice core EPICA Dome C temperature record[17] on the AICC2012 age model[73]. Purple, blue, and orange dots respectively represent SST, CHC, and stadial events recorded from Site U1542 and Site U1385. Timing and nomenclature of isotopic stage follow[77]. Vertical purple bars and Roman numerals indicate glacial terminations. H Henrich events, AIM Antarctic Isotopic Maxima, MIS Marine Isotope Stage, T Terminations, I Inceptions.

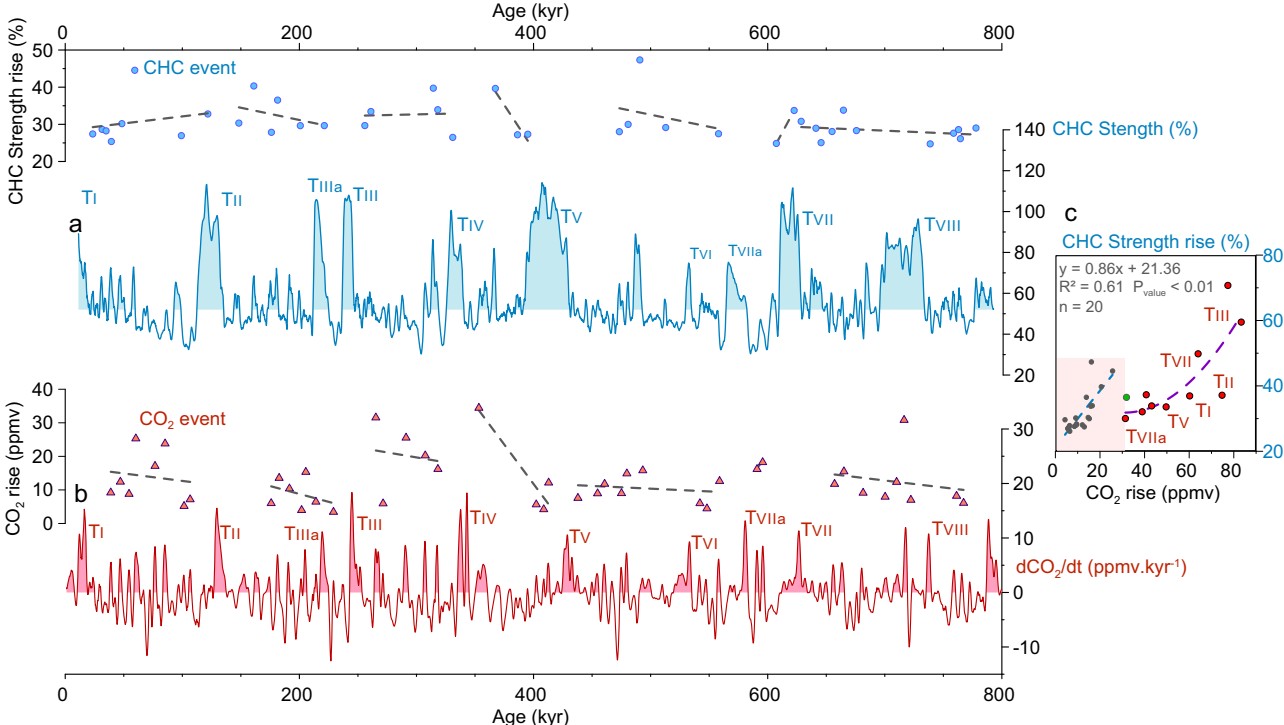

**Fig. 6 | Cape Horn Current and rate of atmospheric $CO_2$ changes during the Pleistocene. a** Amplitude of major millennial-scale Cape Horn Current (CHC) strengthening events (blue dots) and CHC Strength from Site U1542. **b** Amplitude of atmospheric $CO_2$ release events (red triangles) and rate of atmospheric $CO_2$ changes (dCO$_2$/dt) from ref. 11 based on atmospheric $CO_2$ concentrations from the EPICA Dome C ice core[79] (Supplementary Fig. 11). **c** Correlation between the associated amplitude of the CHC events with atmospheric $CO_2$ rise observed in less than 7 kyr (Supplementary Fig. 11). Light red area indicates millennial-scale window.

ACC strength, decreasing Southern Ocean stratification, promoting ventilation, and consequently, increasing atmospheric $CO_2$ levels. The similarities between $CO_2$ changes and CHC strength confirm and extend the role of the ACC fluctuations in enhancing the exchange between surface and deeper water in the Southern Ocean and the corresponding release of $CO_2$[64,65,67]. During each glacial inception, a reduction in SWW wind-driven upwelling, marked by the abrupt drop in the CHC strength record (up to 70% reduction during Inception 6, 3, 2, and 1; Fig. 6a) may have reduced the exchange of water from the deep ocean to the surface, thus contributing to the storage of carbon in the deep ocean and the reduction of atmospheric $CO_2$[62]. Other mechanisms, such as dust-borne iron fertilization[68], and SST and salinity changes[69] also contributed to the continuing and more gradual drawdown of $CO_2$ (Supplementary Fig. 11c) throughout the glacial stage by enhancing the efficiency of the global ocean's physical and biological pump[62,70].

In conclusion, Site U1542, located underneath the palaeoceanographically sensitive CHC, provides unprecedented insights into the millennial-scale CHC variability over the past 790 kyr in the ESP. Coupled CHC strength and SST changes provide compelling evidence that millennial-scale variability previously documented for the last glacial cycle persisted for the last eight glacial cycles. In line with evidence from other Southern Ocean records, this variability is representative of the broader SH. A comparison with NH records indicates that the interhemispheric dynamics observed during the last glacial period have persisted over the Pleistocene. Periods of climate instability in one hemisphere align with those in the other hemisphere, imply that the expected impact of DP throughflow changes on the Atlantic circulation characteristic for the most recent glacial cycle extends to the past 800 kyr. The CHC strength record supports the significant role played by the ACC in promoting inter-basin water mass exchange in the Southern Ocean can influence atmospheric $CO_2$ levels. Notably, these findings align with contemporary observations of a warming and

accelerated Southern Ocean[71] in conjunction with AMOC weakening[72] under anthropogenic forcing of the climate.

## Methods

### Sediment record

We analyzed a Pleistocene sediment record recovered during the International Ocean Discovery Program (IODP) Expedition 383 Site U1542[26]. Positioned in the ESP at 52°42.29'S, 75°35.769'W, IODP Site U1542 is situated ~30 nautical miles west of the entrance to the Strait of Magellan, at a water depth of 1101 m beneath the southward-flowing Cape Horn Current (Fig. 1). The site sits at the upper slope of the Chile continental margin, within a relatively small-scale sediment depocenter ("sediment drift").

The nearly continuous, undisturbed, 249-m-long sedimentary sequence recovered at Site U1542 covers the past 790,000 years with sedimentation rates that exceed 30 cm/kyr. The glacial sedimentary sequence is primarily constituted of siliciclastic sediments with low carbonate contents (~1–12 wt% $CaCO_3$), and biogenic silica contents ranging from 1 to 4 wt%. Interglacials are characterized by sandy foraminiferal ooze (~30–55 wt% $CaCO_3$) deposited during warm interglacial periods[26].

### Southern Chilean Margin composite record

We combined the sedimentary record from Site U1542 with the published records from the nearby located Calypso piston core, MD07-3128. The U1542 'pre-site survey' sediment core MD07-3128 (30.33 m) was recovered in 2007, at 52°39.57'S, 75°33.97'W (1032 m water depth), during the IMAGES (International Marine Past Global Changes Studies) XV-MD159-Pachiderme cruise on board R/V Marion Dufresne and is situated only ~5 nautical miles from U1542. Given the availability of several high-resolution multi-proxy records for MD07-3128[4,33], we chose to incorporate the ~60 kyr MD07-3128 sediment core data into the corresponding section of the Site U1542 sediment record, in order

to build a composite sequence with very high resolution in the last glacial period. To achieve this, we aligned the bottom of MD07-3128 with the corresponding age in Site U1542, using reflectance b* records from both cores and alkenone-derived SST and XRF Zr/Rb serving as controls (Supplementary Fig. 2). The tie point was identified at 28.63 m (MD07-3128 depth) corresponding to 25.47 CCSF-A (m) in U1542.

Due to the observed persistent temperature offset in alkenone-derived SST between the two records, attributed to the use of a different type of gas chromatography column for the MD07-3128 core, we applied a correction of + 1.44°C to the MD07-3128 SST data from the previous data set by Caniupán et al.,[33], (Supplementary Fig. 2). This correction is derived from the repeated measurement of a reference alkenone standard using both chromatography columns employed for the two sedimentary records. We note that this correction does not change the amplitude between G/IG and millennial-scale shifts in the SST records. Furthermore, a discrepancy in XRF-based Zr/Rb values between Site U1542 and MD07-3128 has been observed. Considering that XRF measurements may exhibit variability across different laboratories, we employed here a simple linear regression between both records to assess the drift between devices. The regression equation (Value$_{U1542}$ = 0.6779*Value$_{MD07-3128}$ + 0.1836) was utilized to align MD07-3128 values to the same scale as Site U1542 values.

## Age model
The stratigraphy of MD07-3128 is well-constrained by 13 Accelerated Mass Spectrometry $^{14}$C AMS age from mixed planktonic foraminifera, along with the identification of the Laschamp paleomagnetic excursion[4,33,52]. We use the latest age model by Anderson et al.[52]. For Site U1542 in the time interval 65–800 ka, we graphically correlated glacial terminations and inceptions based on the benthic foraminiferal oxygen isotope (δ$^{18}$O) and XRF-based Ca counts from Site U1542 to the Antarctic ice core EDC temperature record[17] on the AICC2012 age model[73] (Supplementary Fig. 3). As Site U1542 is located at a relatively shallow depth, several periods exhibit strong variability, with a relatively subdued G/IG variability in the benthic record. For instance, MIS 13 and 7 are not evident in the XRF Ca or benthic δ$^{18}$O record. To address this, we added additional tuning points using our SST reconstruction with reference to the Antarctic ice core temperature record[17] (Supplementary Fig. 3).

## Stable oxygen analysis on benthic foraminifera
Foraminiferal stable oxygen (δ$^{18}$O) measurements were performed on samples of each 2 tests of the infaunal benthic foraminifera *Uvigerina peregrina* from core Site U1542. The samples were wet-sieved using a 125 μm mesh, oven-dried at 50°C, and then stored in glass vials. *Uvigerina peregrina* from the sediment fraction larger than 250 μm were handpicked under a stereo microscope every 20 cm. Isotopic analyses were performed on a Thermo Scientific MAT 253 mass spectrometer with an automated Kiel IV Carbonate Preparation Device at AWI. External reproducibility of δ$^{18}$O measurements based on an internal laboratory standard (Solnhofen limestone) measured over a 1-year period together with the samples was better than 0.08‰ for δ$^{18}$O. Isotope data has been converted to the delta notation. The isotope values were calibrated versus IAEA603 and are given in per mil (‰) relative to the V-PDB (Vienna Pee Dee Belemnite) standard.

## Biomarkers analysis
For the determination of alkenones at Site U1542, about 5 g of freeze-dried and homogenized sediment samples were extracted by accelerated solvent extraction (ASE 350, Dionex) with a mixture of dichloromethane and methanol (DCM:MeOH, 9:1, v/v) at the Alfred Wegener Institute Bremerhaven. The resulting total lipid extract was further separated into three fractions through column chromatography with silica gel as the stationary phase. *n*-alkanes were eluted with Hexane (5 ml), alkenones were separated using DCM (5 ml), and

glycerol dialkyls glycerol tetraethers (GDGTs) were eluted with DCM: MeOH (1:1; 4 ml). The first and third fractions (i.e., *n*-alkanes and GDGTs, respectively) were stored for subsequent investigations. Internal standards (squalane, hexatriacontane, C$_{46}$-GDGT) added before extraction served for quantification purposes.

Alkenones were analyzed by gas chromatography on an Agilent 7890 fitted with a flame ionization detector using an Agilent VF-200 ms capillary column (60 m length, 250 μm diameter, 0.25 μm film thickness). The oven temperature was programmed to be held at 50 °C for 2 min, then increased at 20 °C/min to 255 °C, at 3 °C/min to 300 °C, at 10 °C/min until 320 °C and held for 10 min. The identification of alkenones was achieved by comparing the chromatographic retention times of the samples with those of a laboratory *Emiliania huxleyi* culture extract that was routinely used as a working standard to control data quality. The reproducibility of the procedure was evaluated using a homogeneous sediment standard, extracted with every batch of samples. The relative analytical errors were below 0.5 °C in SST estimates. To convert U$^{K'}_{37}$ values (expressed as the ratio of C$_{37:2}$/ (C$_{37:2}$ + C$_{37:3}$)), into an estimation of SST, we applied here the calibration of Prahl et al. (SST = (U$^{K'}_{37}$ −0.039)/(0.034)), widely used in paleotemperature reconstructions[35]. The instrument analytical precision based on replicate analyses of the culture extract was 0.23°C ($n$ = 29).

## Geochemistry
We obtained geochemical data through the high-resolution X-ray fluorescence (XRF) scanning measurements of Site U1542 using an Avaatech (non-destructive) XRF Core Scanner at Texas A&M University. The scanning was performed at intervals of 3 cm (area 10 × 12 mm, down-core x cross-core) across the core in three runs at 10 kV (Tube current 0.16 mA, live time 6 s, no filter), 30 kV (1.25 mA, 6 s, Pd-thick filter) and 50 kV (0.75 mA, 10 s, Cu filter).

## Grain-size measurements and current speed reconstruction
To assess changes in near-bottom flow speed, we employed the sortable silt (SS) proxy. SS is widely used to assess variations in near-bottom flow speed in deep-sea sediments. This sedimentological parameter operates on the principle that a coarser mean size reflects stronger near-bottom flow, through selective deposition and winnowing[36]. As current velocities are very high in our study area, we extended the sortable silt range by including fine sand (SSFS). This proxy exhibits a strong correlation with modern and past variability in the Drake Passage area[22].

For the grain-size analysis, the terrigenous fraction was isolated from 5 g of freeze-dried bulk sediment by treating each sample with 5 ml H$_2$O$_2$ (35%), 5 ml HCl (10%) and 5 ml NaOH (6%) while being heated, to remove organic matter, carbonate, and biogenic silica. The samples were rinsed and centrifuged until reaching a neutral pH between each step. Immediately before measurements, Na$_4$P$_2$O$_7$·10H$_2$O (sodium pyrophosphate) was added to the leached material and the samples were sonicated for 30 s, to avoid aggregations. Grain-size analyses were carried out with a Mastersizer 3000 (Malvern Panalytical) at the Leibniz Institute for Baltic Sea Research Warnemünde. To investigate whether the sediments were subject to significant bottom current sorting, the mean sortable silt plus fine sand grain size (SSFS; geometric mean of the 10–125 μm silt fraction) and the SSFS percentage (SSFS%, defined as the 10–125 μm fraction relative to the <125 μm fraction) were calculated after Wu et al.[22], utilizing the software GRADISTATv9.1[74].

Several studies suggested that changes in the element compositions of fine-grained sediments, particularly the zirconium/rubidium (Zr/Rb) count ratio, hold significant potential as a tracer for grain-size variations of marine sediments, providing valuable insights into current strength[36]. Recently, the logarithmic Zr/Rb ratio derived from XRF core scanning has been utilized as a proxy for reconstructing

millennial-scale variability in near-bottom flow speed in the Drake Passage[4,22,37] and in the Southern Ocean[22]. To apply the (Zr/Rb) proxy to our record, we correlated SSFS and ln(Zr/Rb), obtaining a tight positive linear correlation (SSFS = 25.87 × ln(Zr/Rb) + 9.55, $R^2$ = 0.80, $n$ = 133, Supplementary Fig. 5).

In order to obtain a constant time resolution along the record, we resampled the resulting SSFS record at 100-year intervals, and subsequently smoothed it with a 0.6 kyr running average. The smoothed SSFS record is broadly similar to the resolution of the SST reconstruction. This running mean window has also the advantage of being close to the smoothing window selected by Barker et al.[50], to predict abrupt events, providing a reasonable compromise between noise reduction and signal fidelity. In order to obtain current velocities, we used the SSFS-current speed equation for the Drake Passage region (SSFS = 2.76U + 4.61)[22] (Supplementary Fig. 5). While the standard analytical error of the grain-size analyses to obtain sortable silt values is in the range of ±0.6 μm (at 20 μm, see below), the exact error of the current speed calculations from current meter data is more difficult to assess as only few current meter and grain-size data are available. McCave et al.[41] estimated the standard error to be in the range of ±12.5%.

### Spectral analyses and filtering

To identify periodic components in the spectrum of the U1542 record, spectral analyses were performed using the Blackman-Tukey spectral power estimator, implemented in the Analyseries software. Prior to analysis, linear trends were removed, and values were normalized. The frequency scale was resampled from 0 to 0.1 with a step of 0.0002. A Bartlett window was applied, and the bandwidth was approximately set to 0.005. Prior to analysis, our record was evenly sampled at 200 years. In consideration of the lower resolution of core PS75/034-2, the record was evenly sampled at 1 kyr.

The removal of orbital-timescale variability is achieved by subtracting a 7 kyr smooth. A cut-off at 7 kyr was selected, consistent with several previous studies (5 kyr in Pahnke et al.[16], 7 kyr in Barker et al.[50]). The filtered result emphasizes millennial-scale variability and is thought to remove the background climate evolution on G-IG timescales (Supplementary Fig. 8).

### Characterization of the millennial-scale events

To identify millennial-scale events in the records, we used a thresholding approach, following Barker et al.[50] to predict the occurrence of Dansgaard-Oeschger (DO) events. We identify events using minima in the first-time differential of the 600-year mean signal giving similar results as from the filtered signal (i.e., <7 kyr to exclude long-term insolation-driven signal) of CHC strength and SST record (Supplementary Fig. 8). This approach is thought to avoid any subjectivity in the detection of millennial-scale events[13]. Each event is therefore defined by an abrupt CHC strengthening or SST warming. The empirical choice of the threshold underscored the importance of maintaining a balanced approach. An overly sensitive threshold could indeed prevent the distinction between environmental changes in the record and variations introduced by analytical and calibration uncertainties in the alkenone-SST and bottom-current estimates. On the other hand, a threshold that is too insensitive might omit important events, potentially leading to a biased representation of Pleistocene millennial-scale variability. We selected a threshold intending to effectively distinguish between event magnitudes chosen to ensure that the number of events fell within the range of DO events observed over the last glacial period and in accordance with estimations for the northern hemisphere records. Ultimately the threshold selected was set higher than the standard deviation of the data to only capture climatic events, minimizing the inclusion of noise or non-climatic variations in the sedimentary

record. Subsequently, we identified major events by sorting each event by their respective amplitude, which visually displays two distributions at 24% CHC strength (Supplementary Fig. 8g, h, i). Each amplitude is defined by the difference between the minima before each event and the maxima after each event (Supplementary Fig. 8a, b, c).

## Data availability

All relevant data in this manuscript are available at PANGAEA Data Publisher (https://doi.pangaea.de/10.1594/PANGAEA.972776 and https://doi.pangaea.de/10.1594/PANGAEA.972778). Additional data related to this paper may be requested from the authors.

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

## Acknowledgements

We thank the captain, crew, and scientific party of R/V JOIDES Resolution for their support during International Ocean Discovery Program (IODP) Expedition 383 "Dynamics of Pacific Antarctic Circumpolar Current (DYNAPACC)". We acknowledge funding by the AWI Helmholtz-Zentrum für Polar- und Meeresforschung through the institutional research program "Changing Earth—Sustaining Our Future" through the PhD program INSPIRES. The authors are grateful to Nils Plonka for grain-size measurements and N. Beech who greatly assisted in improving the English of the manuscript. We acknowledge funding by the DFG Priority Program 527 grants SESPOD (AR 367/16-1) and IODP383-DYNAPACC (LA1273/10-1) to V.R. and F.L.

## Author contributions

V.R. and F.L. conceived the study. V.R. conducted the analysis and drafted the manuscript. V.R., N.R., and H.S. conducted alkenones measurements. V.R. and H.W.A. conducted grain-size measurements. L.L.J. and I.M.V. provided the isotope measurements. V.R., F.L., L.L.J. and H.W.A. constructed the age model. V.R., F.L., N.R., H.S., H.W.A., S.B., L.L.-J., A.W., G.K., I.M.V., T.M.L.P., R.T., and G.W. contributed to the scientific discussion, reviewed, and contributed to the text of the manuscript.

## Funding

## Competing interests

The authors declare no competing interests.
