## [Transparent Peer Review file · Nature Communications]

790,000 years of millennial-scale Cape Horn Current variability and interhemispheric linkages

Corresponding Author: Dr Vincent Rigalleau

Version 0:

Reviewer comments:

Reviewer #1

(Remarks to the Author)

Review of "Persistent interhemispheric link of the Antarctic Circumpolar Current millennial-scale oscillations over the last 800 kyr" from Rigalleau et al.

Rigalleau et al. provide new ACC proxy data with an unprecedentedly high resolution to demonstrate a relationship between the ACC and millennial-scale northern hemispheric variability. The manuscript was nice to review: it reads very well, and figures are fancy and pleasant to read as well.

Unfortunately, as I have very limited knowledge in sedimentology, bio/geo chemistry, and quaternary climate, I'm afraid I had to ask the editor to replace me with another reviewer or to add a new one for the next review. I've tried to focus my review on statistical analyses provided by the authors, and I have not found any compromising content for publication in this respect. This is the reason why I only have a list of minor comments the authors must consider, see below.

Thanks again for the nice and interesting read.

Minor comments:

Referencing in general: Blank spaces before superscript referencing numbers must be deleted throughout the text in order to fit the journal format.

L. 26: I suggest using a less personal phrasing for "we still lack [...]", such as "there is still a lack of a clear mechanistic understanding of these variations", or "a deeper understanding of these variations is still lacking".

L. 55: "[...] is the largest current system on Earth." – Such statement needs a reference.

L. 55: "It is driven by atmospheric forcing" – reference needed.

L. 56-57: "bathymetry and ocean density gradients originating from surface" – reference needed.

L. 65-67: "Various model studies" contrasts with the presence of a single reference.

L. 65-67: It would be worth specifying the time scales at which this influences is exerted in models. In this line, what kind of models the authors refer to? EMICs/GCMs/others?

Figure. 1: There is some information missing in the figure caption:

- Indicate that Altimetry-derived ACC fronts are represented with dashed lines.
- Indicate in the caption what yellow points are.
- Indicated over which period the mean annual sea surface temperatures were calculated.

L. 112: Replace "an previously" by "a previously".

L. 161-162: “the millennial scale variability in ACC strength persist”, there is missing “s” at the end of “persists”

Figure 2: Orbitally induced spectral peaks are clear for the provided time-series. Could the authors try providing an additional analysis evidencing the millennial-scale frequency they refer to throughout the paper? So far it is only a visual assessment. A wavelet analysis could be able to provide such information.

L. 180-183: How did these threshold-based events classification was made exactly? From the text it reads that thresholds were somewhat determined arbitrarily (1.6°C and ~10cm/s), where the “~” reads quite surprising as one would expect a threshold to be fixed. I can understand it is difficult to determine a clear approach to determining classification thresholds here, but the authors should consider being more transparent by precisely explaining how these choices were made. In the case thresholds were determined arbitrarily, an obvious subsequent question I would have is: are the following results sensitive to this choice then? Could the authors demonstrate these thresholds were not cherry-picked?

L. 194: I think “1 ka” should be “1 kyr” here? I may be wrong here.

L. Replace “This imply” by “this implies”.

L. 205-207: A reference is needed to support this sentence.

L. 209-210, Figure. 3E,F: It is difficult to assume the linear relationship to be significant here. The R2 most likely is very low (given the distance of points with the regression line), and the sample size is pretty small. I would mention this in the present interpretation and add some conditionality in this statement.

Figure 3: Significance tests for regression line slopes for panels C-F are missing – please provide p-values and sample sizes. Use Student t-tests.

Interpretation of Figure 3: Under the assumption that millennial-scale events occur on a regular basis, one would inevitably obtain more events for longer time windows (thus for longer MIS). In other words, what do we learn exactly from panels C and D?

Figure 4: To my knowledge, figures and their respective caption must be self-consistent in Nat. Comm. This means that acronyms and abbreviations such as “D-O”, “AMOC”, or “EDML” must be given in their full version the first time they are mentioned in the caption.

L. 387: Missing “-“ symbols at “state of the-art”.

L. 423: The acronym for North Atlantic Deep Water (NADW) has not been defined earlier.

L. 442: “SWW wind-driven” reads redundant. Should maybe be SWW-driven?

Figure 6: See my above comment of Fig. 4.

Reviewer #2

(Remarks to the Author)

The manuscript “Persistent interhemispheric link of the Antarctic Circumpolar Current millennial-scale oscillations over the last 800 kyr” by Rigalleau et al., provides new high-resolution current strength and sea surface temperatures (SST) records, spanning the past 790.000 years. The sedimentary archive IODP U1542 is located within the Cap Horn Current (CHC) – northern branch of the ACC. The authors report that their records show persisting millennial-scale climate variability throughout all Late Pleistocene, with stronger ACC flow coinciding with warmer SST and Antarctic warm intervals. Additionally, the Southern Hemisphere changes seems to coincide with North Atlantic millennia-scale fluctuations suggesting a close link between ACC flow and the Atlantic thermohaline circulation. Lastly the observed changes in IODP U1542 also seem to follow atmospheric CO₂ changes and for that, the authors propose a mechanistic link through the Southern Ocean carbon cycle.

Variations in the strength and position of the ACC play an important role in global ocean circulation, ocean carbon cycle and thus Earth’s climate. Understanding the mechanistic links between these processes is therefore of major importance. I would first like to point that the total number of samples analyzed for this study, especially the alkenone-based SST is impressive. Overall, the data are high quality and convincing. I also appreciate the way the data are presented - figures illustrate clearly the datasets. These high-resolution records are of great interest and importance to the scientific community and can be used for interhemispheric comparisons (as authors did) providing important insights into the millennia-scale variations of the ACC and their links to Earth’s climate.

Said that, I unfortunately think that the manuscript, in its current form, does not provide new insights in terms of processes/mechanistic links and the understanding of past ACC variations, compared to what has already been reported by several studies in the same area (e.g., Lamy et al., 2015; Toyos et al., 2020; Wu et al., 2021). Moreover, the new data does not always strongly support the main conclusions. I think that there is more information that can be extracted from these datasets and help in understanding the dynamics of the ACC and their interactions with climate and CO₂ variability during

the past 800 000 years. Below I explain in detail my main concerns.

SST and ACC strength reconstruction from current-controlled deposits.

As IODP U1542 is located close to the Chile continental margin, in the upper slope and thus not in a deep ocean setting, it is expected that sedimentation is largely affected by down-slope processes (turbidites, debris flows, etc), mixed with along-slope processes (bottom currents). Sedimentation rates are five times higher during glacial times compared to interglacial periods. This indicates a major sediment input from land. That could have a great influence in the grain size variations. Using the mean sortable silt grain-size (SS; geometric mean of the 10 – 63 μm silt fraction) versus the sortable silt percentage (SS%, 10–63 μm fraction relative to the <63 μm fraction) can help to investigate whether the sediments are subject to significant bottom current sorting. However, this is not the only test that can be used and given the complex depositional setting, I would recommend to provide more evidence on whether bottom currents play the main control on sediment transport in the upper slope of the Chile continental margin. In line 139: authors argue for only limited ice-rafted debris impact in their sediments. This can be easily checked using their data by plotting the fine sand (wt %) and the coarser “possibly IRD fraction” (>150 μm ; wt %). Additionally, a more detailed investigation on the grain-size and XRF-core scanning datasets can really help. A Principal Component Analysis on the XRF scanning and grain-size data can provide robust insights into the main processes governing the variations in your sedimentary archive. Alternatively, it would be helpful to plot together the SS, SS%, fine sand (wt %), (>150 μm ; wt %) and terrigenous elements from your XRF scanning dataset, e.g., Ti, Al, Zr, Rb. This will illustrate potential similar patterns and help to robustly argue for the main processes controlling sedimentation at Site U1542.

Sensitivity of the Southeast Pacific to orbital and millennial-scale climate oscillations.

Orbital and millennial-scale variations of the ACC strength in the Drake Passage region, as well as in the Chilean Margin and have been previously reported covering different time periods: over the 65 ka in the CHC current (Lamy et al., 2015), over the past 140 ka (Wu et al., 2021), over the past 1.3 Ma (Toyos et al., 2020). Thanks to these works, we have a good view on the variations of the ACC orbital and millennial-scale variations of the ACC/CHC strength in the Drake Passage and Southern Chilean Margin. The aforementioned studies provide similar conclusions and mechanistic processes with the current work, describing stronger ACC in interglacial intervals vs weaker ACC in glacial intervals (Lamy et al., 2015; Toyos et al., 2020; Wu et al., 2021), linking them with low AMOC strength (Wu et al., 2021). I believe that with the new high-resolution records presented in this study the authors have the opportunity and advantage to explore in depth these mechanisms. So, I highly recommend to go deeper in their interpretations.

For example, in Lines 232-234: “At orbital timescales over the past 790 kyr, our reconstructed SST variations at the southern Chilean margin largely follow the atmospheric temperature record of the Antarctic ice core EDC and the open Southeast Pacific SST recorded at PS75/034-2 (Fig. 2C-E).” Looking the figure carefully, I would say that this is not really the case. PS75/034-2 follows largely the EPICA Dome C record, but Site U1542 is not always following the same pattern (e.g., MIS 7-6, MIS 13-12). This is a very interesting observation from these new high-resolution data that I believe needs to be discussed in more detail. I agree with the interpretation of the authors for the higher temperatures recorded at Site U1542 compared to PS75/034-2, but there is more to discuss and extract from the comparison between these records.

The authors suggest that the decoupling between Southern Ocean and Antarctica southern latitudes (Lines 253-259), resulted in an increase of the meridional thermal gradient, increased moisture to the poles and continue ice growth, but as they mention later this feature is probably not a prerequisite for the ice sheet growth. This part needs further explanation.

Lines 262-264: “Enhanced flow speeds coincide with warmer periods, while reduced flow speeds coincide with colder periods, implying a close coupling between near-bottom current velocity and SST (Fig. S4)”. The correlation the authors show in Fig.S4 did not suggest a close coupling ($R^2= 0.5$). I would say that this close coupling they mention in the text is not actually the case. There are indeed times, where peaks of ACC strength coincide with high SST and warm intervals in Antarctic climate, but there are many times that this is not the case. For the example, the last 100 ka, ACC show a long-term increasing trend whereas SST data from U1542 and more clearly from MD921-20 show a long-term cooling trend coinciding with Antarctic ice core EDC. This is observed in other time intervals as well. This is a very important observation that needs to be discussed.

In lines 268-288, the authors propose that latitudinal migrations of the Southern Ocean frontal system during glacial / interglacial times as a primary driver of changes recorded at Site U1542 (located north of the SAF), similar to the proposed mechanism in PS97/85 (located in the polar front). Northward migration of the Southern Ocean frontal system during glacial times should bring SAF closer to Site U1524, compared to interglacial times. Given that stronger ACC flow appears within /near the SAF. Why don't we see higher ACC flow speeds during glacial times at Site U1542 and vice versa for the interglacial times? In other words, do the authors expect to see the same trends in ACC strength variations in all records: PS97/085-3, PS97/093-2, PS75/034-2 and Site U1542, unrelated to their locations within the ACC frontal system? Why?

I would recommend to include a figure showing ACC strength and SST variations from all sites discussed in the manuscript (U1542, PS75/034-2, PS97/093-2 and PS97/085-3). This figure will help to clearly identify changes across the whole ACC frontal system in the Drake Passage region.

A persistent interhemispheric teleconnection

I generally found this section difficult to follow as the main message seems to be hidden in the comparison between the

different dataset.

Lines 302-303 "Millennial-scale climate events at Site U1542 are found to be contemporaneous with AIM events" A closer look to the figure does not show this. There are peaks in ACC strength that does not coincide with AIM events. Moreover, in lines 311-314, authors report that intervals characterized by stronger ACC coincide with the weakening of the AMOC (during Heinrich Stadials). I would argue that this is not what the figure 4 shows. I suggest that the authors modify the text, so it will describe more precisely the comparison between the different datasets and clearly state the main message of this section.

Role of ACC in CO₂ changes

The authors associate a total 31 major ACC strengthening events with CO₂ rise, 15 of them present a CO₂ rise before an ACC strengthening and 15 events present a CO₂ rise after an ACC strengthening, suggesting a link between ACC strengthening events with CO₂ rise. They propose that strengthening and southward displacement of the SWW, enhancing ACC strength, decreasing Southern Ocean stratification promoting ventilation, resulting in increasing atmospheric CO₂ levels. However, what are the mechanistic processes that occur during the events where CO₂ rise before an ACC strengthening? Are they the same?

Line 405-407: "Moreover, the relative changes in flow strength appear to be positively correlated to the amplitude of CO₂ variations (Fig. S13)". However, the correlation between ACC strength and CO₂ rise shown in Fig. S13 is rather weak ($R^2=0.3$).

As mentioned in a previous comment, a figure showing ACC strength, SST variations and productivity proxies XRF-Ca from all sites discussed in the manuscript (U1542, PS75/034-2, PS97/093-2 and PS97/085-3) can help to unravel changes across the whole ACC frontal system (e.g., latitudinal migrations, bottom strength and productivity changes) that can be then linked with CO₂ changes.

Minor comments

Fig. 1: Please add the location of MD07-3128

Fig. 6: Please add AIM events

Line 311: Wu et al 2021, change it to reference number

Lines 326-329: This part can be moved to methods

Lines 410-411: Please cite figure 6 and add AIM events in the figure.

I am sorry for not rating this paper higher. I strongly appreciate the efforts of the authors to generate such high-resolution records, which are of great value.

Reviewer #3

(Remarks to the Author)

In this paper the authors analyze a new core from underneath the Cape Horn Current (CHC) system that goes back 790 kyr. They find persistent millennial scaled variability in the Antarctic Circumpolar Current (ACC) strength and Sea Surface Temperatures, and relate this to millennial variability in the North Atlantic suggesting that this link is present over the past 790 kyr. Lastly, they also suggest a role of this variability in the carbon cycle in the Southern Ocean with global consequences. I think the paper is in principle very interesting and I applaud the authors for the amount of work done. I also believe that getting a better understanding of the influence of Southern Ocean dynamics on the ocean, the carbon cycle and the entire climate system is very relevant for future climate change since it is a large source of uncertainty.

However, I have some major concerns about some of the aspects of this study. To address these concerns it might be necessary to make significant revisions that can be quite a bit of work and might also change the scope of this paper. It might therefore be better to reject/withdraw the paper and submit it to a different journal or resubmit here. I leave that decision to the editor and/or authors.

My first main concern is that it is claimed that the CHC is part of the ACC and it is assumed that CHC dynamics captured in the core provide an accurate proxy for ACC dynamics. I am not convinced this assumption is valid. Figure 1 actually shows that the core used in this study is north of the Northern Boundary and therefore not in the ACC. Also two papers cited (Chaigneau and Pizarro, 2005; Park et al., 2019) state/imply that the CHC is not part of the ACC. I think it matters for the significance of this paper and the overall conclusions of this paper whether this assumption is valid, which in my opinion it is not valid. I suggest that the authors provide a sound argumentation why this assumption is valid.

My second main concern is that it is difficult to check the conclusions with the figures provided and how I analyze some figures I would conclude that there does not seem to be a clear relation between some of the variables. First of all, the figures with time axes often have a length of 800 kyr while variability at the 1 kyr scale is analyzed. I suggest besides providing the full length series, also provide zoom in of e.g. ~50 kyr to support your conclusions. Secondly, for me Figure S6 shows that there is no(!) relation between events in the Northern Hemisphere and the Southern Hemisphere because the graphs look very different in my opinion. In the main text Figure S6 is used to argue that there is a clear relation. I suggest that when considering the relations between the different variables either a well discussed statistical test is used and/or a sound physics based mechanism is provided.

My third main concern is closely related to what I describe above. I think there is a lot of variations in timing and amplitude in

the events of the different variables and this makes me doubt whether the events are necessarily connected. Events in the variables can be leading, lagging, coincide or not coincide at all without, in my opinion, a good explanation why these events are connected even though they are so different in these aspects. Also the time window used to detect events in multiple variables is a bit large considering that typical ocean timescales are at 1000 – 2000 year. I suggest providing a better argumentation (e.g. statistical tests and/or physics based mechanisms) as to why and how events between different variables are connected. Improving figures displaying the events can already help a lot here.

Next, I have also some more specific comments/suggestions for the authors:

General:

1. In the text and in the title the word 'oscillation' is used frequently. In my opinion, an oscillation has a more or less stable frequency and I don't think that is the case in your records. I suggest using for example variability instead of oscillation in the title and throughout the text.
2. I don't see anything about measurement uncertainties in the text and in the figures. Have I missed this? This should be included in my opinion.

Text:

1. Also based on previous comments, I think the title does not accurately capture the content of the paper in my opinion. I think that oscillations could be replaced with variability, the ACC with CHC, and 800,000 with 790,000 since that is the length of your core. Also, as mentioned previously I have my doubts about the interhemispheric link based on this study.
2. Line 55-57: I think this sentence can be clarified a bit. It's not clear to me if the atmospheric forcing is just the Southern Westerly Winds or also temperature and freshwater forcing. It is also not clear whether the salinity changes also relate to the ocean density gradients because of the comma before 'and' at the end of the sentence.
3. Line 109 – 111: Can you refer to a figure here?
4. Line 111 – 113: I'm not sure if I understand this. Is there a bias in the measurements? Is this bias compensated for? Why not if it's not? I suggest clarifying this phrase.
5. Line 197: 'amplified', I think you mean here that events are more likely to occur in these ranges, so I suggest changing the text to reflect this.
6. Line 199: you make a reference to ice sheet volume in Fig. 2, but there is no ice sheet volume in Fig. 2 which is a bit confusing. I suggest being more accurate in this statement.
7. Lines 279 – 288: I don't understand where to look for the amplification in Fig. 4. How does this amplification look in the data? I also don't understand how this supports the idea of SWW shifts as a primary driver. How I would reason (tell me if I'm correct in this): when SWW shifts southward, the water flowing into the CHC becomes colder, when the SWW shift northward, the water becomes warmer because of the latitude the source waters come from. But I don't understand now where to look in the data for the SST response in U1542. I suggest clarifying the text and specifically the link with Fig. 4.
8. Line 382 – 388: Here you relate your results to results from ESMs. I do not agree that these results can be compared. You look at completely different timescales. The timescales assessed here are 1 – 10 ka, while the ESM studies are on centennial time scales. The processes are therefore, in my opinion, not translatable. I do think that you can mention it in the text, but I suggest being more nuanced here.

Figures:

1. Figure 1: This corresponds to my main concern 1. There are arrows drawn north of the NB line that are called the ACC. Looking at the paper where the data is from, they call the NB line the actual northern boundary of the ACC. I suggest changing the name to SPC as in Fig. S1 or motivating very clearly why you consider this to be the ACC.
2. Figure 1: No source of the SST data is given, and it is not clear what period the SSTs represent. I suggest adding both in the caption.
3. Figure 3: This figure misses clear labels and units for the axes. It is therefore difficult to interpret this figure. On the left 'ACC strength' and 'SST' look like labels, but it took me a couple of looks to understand they're not.
4. Figure S3: Caption does not clearly explain the subplots. In particular, LSR (local sea level rise?) is not mentioned.
5. Figure S7: I think it would be useful to be able to better compare the timing of the events by plotting the events of the different variables closer to each other. The time between 2 events could for example be shown by changing the marker size.
6. Figure S9: Shouldn't n be 103 in the bottom subplot?

The data collection is outside the scope of my expertise.

Version 1:

Reviewer comments:

Reviewer #2

(Remarks to the Author)

The manuscript titled " Persistent interhemispheric link of the millennial-scale Antarctic Circumpolar Current variability over the last 790 kyr" from Rigalleau et al., presents a high-resolution current strength and sea surface temperature records covering the past 790,000 years from the Chilean Cape Horn Current, off the Chilean coast.

The authors present evidence of persisting millennial-scale climate variability throughout the last eight glacial periods, characterized by stronger current flow and warmer sea surface temperature, coinciding with Antarctic warm intervals. They

interpret these changes as being linked to the Antarctic Circumpolar Current (ACC) and suggest a link to North Atlantic, proposing persisting interhemispheric dynamics over the last 790 kyr. Lastly, they associate the variability in current strength with atmospheric CO₂ fluctuations, suggesting a role of this variability in influencing the Southern Ocean carbon cycle.

This is the second time I have reviewed this paper, and overall, the results of the study are interesting, with an impressive amount of work and analyses. The data are of high quality, and the figures clearly illustrate the datasets. This work could make a significant contribution to our understanding of the Southern Ocean climate, ocean and carbon cycle dynamics, and its broader connections to Earth's climate system.

While the manuscript presents valuable insights, I believe it is not yet ready for publication. There are still major concerns that need to be addressed, particularly in clarifying interpretation of the results and improving the flow and readability of the text. I have outlined my concerns and specific recommendations below to help strengthen the manuscript.

My first main concern remains whether the grain-size changes at U1542 can be linked to changes in the ACC strength. This is due to the location of the site which is situated very close to the Chilean coast, in the upper slope and thus its sedimentation is likely primarily driven by downslope processes related to continental sediment inputs from meltwater transport and sea-level changes, with current reworking playing a lesser role. You mention that the expansion of Patagonian Ice Sheet that reached the continental shelf edge at glacial maxima may explain the interhemispheric differences you observe between your records and others (e.g., 364-367 "the interhemispheric disparity in several cycles..."). This raises the question of whether the observed changes in sediment composition, grain-size, are truly driven by variations in bottom currents or rather by downslope processes. If the latter is the case, the grain-size changes may not serve as reliable indicators of changes in current strength.

The moderate correlation ($R^2 = 0.43$) between the mean SS and the weight percentage of the SS (Supplementary Fig. 4e) may also suggest that current reworking is not the primary control driving grain size changes within the silt fraction.

Additionally, out of the 94 grain size (SS) measurements, fewer than 10 samples were measured from the high peaks in XRF Zr/Rb, with only one sample measured for each major peak (Supplementary Figure 4b). This limited sampling could bias the statistical analyses, as well as the correlation between Zr and mean SS and the calibration of the XRF-Zr/Rb using discrete SS measurements. To provide strong evidence that mean SS and XRF-Zr/Rb can be used as proxies for near bottom current strength in such a dynamic environment, I recommend that the authors provide additional grain size measurements at these prominent XRF Zr/Rb peaks to improve the robustness of their analysis.

Moreover, there are notable differences in the timing and amplitude of ACC current strength changes over the past 800 kyr between U1542 and records from the Drake Passage and Central South Pacific (Supplementary figure 5). The latter sites are strategically positioned within the main ACC pathway and exhibit similar changes, while U1542 does not follow this pattern. Could the proximity of U1542 to land have influenced its grain size patterns, given the variable sediment sources and sedimentary/depositional mechanisms?

My second concern is the "persistent" connections proposed between U1542 records and those from the Northern Hemisphere and Antarctica (EPICA). Given the significant discrepancies in timing, duration, and amplitudes of events, it's unclear whether a persistent interhemispheric link over the last 790kyr truly exists.

While I appreciate the effort in presenting such extensive high-resolution datasets, I find the explanations provided for these differences to be insufficiently robust. Much of the text is focused on reporting these discrepancies, without adequately explaining them, often offering only a single sentence to explain these differences. Below I provide some examples. I believe that refining the text and clarifying the main message would greatly improve the paper's readability. As it stands, the current version is difficult to follow.

-Lines 206-207: "This decoupling between the SH mid-latitudes and Antarctica likely resulted in an increase of the meridional thermal gradient at these periods"

-Lines 216-218: "This partly different behaviour of the bottom currents within subantarctic ACC/CHC and the SST might indicate a different response to atmospheric (e.g., SWW) and oceanic forcing (e.g., see-saw).

-Lines 277-278: "These differences might be due to varying millennial-scale sensitivities and thresholds for SST and ACC strength changes, related to various atmospheric and oceanic forcings (see below)". However, the authors do not explain later these" various atmospheric and oceanic forcings".

My last concern is regarding the mechanism proposed for the changes recorded at Site U1542. Between lines: 228-253, the authors suggest that shifts in the Southern Westerly Winds (SWW) are the primary driver of the observed changes at Site U1542. Later this mechanism is used to support the role of ACC in CO₂ exchanges, for examples in lines 468-470: "... strengthening and southward displacement of the SWW, enhancing ACC strength, decreasing Southern Ocean stratification, promoting ventilation, and consequently, increasing atmospheric CO₂ levels.

However, in their response to reviewer #2, the authors state that "the observed ACC strength variations are consistent and indicate that frontal shifts are not the main factor in determining ACC strength in our study area and the South Pacific in general. I may have overlooked something, but I find myself unclear about the specific mechanism proposed as the driver of

the observed changes in their record.

Other comments

The authors should thoroughly check references in both the text and reference list. For instance, in line 146, citations like (Davies et al. 2020, Hagemann 2024) need to be converted to numerical format. Additionally, in line 807, please note that Hagemann et al. 2022 refers to a preprint, the peer-reviewed version was published in 2023.

Line 159: "Fig 2c". I believe the authors refer to Fig. 2b.

Line 223: Please define the abbreviation "PIS"

Reviewer #3

(Remarks to the Author)

The revised version of the manuscript has improved and some of my concerns have been adequately addressed. However, I still have some concerns about some of the analysis and assumptions made in this study and some small points to clarify the manuscript a bit more.

Before I address my concerns, I have a tip for future submissions (not only for this paper). I like to clarify that it is not meant as criticism and that I did not let this influence my review of the revised manuscript. Some references in the text to figures and references to line numbers in the response are not correct. I also feel like there are more typos in the revised version. Firstly, the incorrect references make it more difficult for me to review the paper because I have to do more work to see where the revised text is, and what figure is actually meant. Secondly, it makes it feel like the revised version was a bit rushed and I think that is a shame, because it is obvious you have spent a lot of time on this study. For the future, I suggest being more thorough on the last check before submission.

My main concern is still how the measurements of the Cape Horn Current (CHC) are framed as being representative of the Antarctic Circumpolar Current (ACC). I am not satisfied with the arguments given in the response. To address the arguments in the response:

(1) The authors refer to the study of Chaigneau and Pizarro (2005; CP05) where an example of a drifter (black drifter in Fig. S1) is provided that leaves the ACC around 100W and returns to the ACC through the CHC. Based on CP05 I would say there are probably a few more drifters that follow a similar trajectory. However, a few drifters do not necessarily reflect the mean flow in this region. Compared to a more recent paper (Strub et al., 2019) I would rather say they are probably not representing the mean flow accurately. I therefore believe that the quality of this paper is insufficient to argue that the CHC is a northern branch of the ACC.

(2) The authors also refer to the study of Zheng et al. (2023). However, I believe the citation to be inaccurate. In Zheng et al. (2023) I do not think they say that the CHC is part of the ACC. What they do say is that the CHC is part of the ACC system. A subtle difference, but in my view a very important one.

(3) Other paleo papers are cited where the CHC is also referred to as being part of the ACC. I did not look thoroughly at these papers, but in my opinion, if they argue that the CHC is part of the ACC, then I think those claims are inaccurate.

My main two concerns here are:

(1) The source of the waters in the CHC. In the manuscript CP05 is used to argue that the waters in the CHC are of ACC origin. However, the sea surface temperatures (SSTs) in Fig. 1 show in my opinion that the water is more likely to be of subtropical origin, which I think is also in line with what Strub et al. (2019) show. Yes, there will be leakage and mixing from the ACC (CP05 show this in my opinion), but based on the SSTs I would argue that most of the waters are of subtropical origin.

(2) From Zheng et al. (2023) we can see that the CHC is at best 10% of the volume transport through the Drake Passage. Therefore, I find it difficult to believe that variability in the CHC is representative of ACC variability.

I think the main conclusions of this paper are based on the assumption that you are measuring ACC variability in the CHC sediment core. I think this assumption is false and I am not convinced by the arguments provided. The variability could be representative of Southern Hemispheric variability, but that is not how the paper is framed. For me to recommend this paper for publication a major revision on how the paper is framed is necessary where the assumption that ACC variability is measured in the CHC sediment core is dropped.

I do want to note that I appreciate that in the text the measurements are referred to as CHC/ACC strength. I think this is well done. However, it is not done consistently throughout the full text and for all figures.

Major points:

- Line 69: I feel like sources 24, 25 and 26 are wrongly cited here. I must admit I only skimmed through the papers but the general message I think of all three papers is that Southern Ocean warming affects AMOC stability. None of them mention millennial scale variability in the Drake Passage throughflow or the Agulhas current (is Agulhas leakage meant here by the way?) as triggers for Dansgaard-Oeschger events as is suggested by the main text. Why are these papers cited here/what message should these citations convey?

- As I understand it correctly now, there is no relation between the millennial scale SST variability and the millennial scale CHC variability? (lines 266 – 311). Do I understand this correctly?

- Like my first review on the relation between SST events and CHC/ACC events, Fig. 6 shows to me that there is not necessarily a relation between CO₂ events and CHC/ACC events. Also here there are many events only occurring in one of the two variables, sometimes the CHC/ACC event is leading, and sometimes the CO₂ event is leading. I suggest presenting the CHC/ACC – CO₂ events in a similar fashion as the CHC/ACC – SST events.

- Line 492: '... our findings shed light on the impact of the DP throughflow on Atlantic circulation on millennial timescales...'. Can you explain to me what is meant with this statement? Because I did not fully get that from this study.

- In view of open science, I encourage the authors to also publish the code used to make the figures and perform the analysis with.

Minor points:

- Line 71: 'abrupt changes' does not sit well with me. Abrupt changes with respect to what? I suggest changing this sentence to something like: '... understanding of millennial scale dynamics in the Southern Ocean is ...'.
- Line 127: '... the major part deviates into the southward branch, the CHC.' This needs a reference.
- Line 148: Use capitals for Supplementary Figure 4.
- Line 159: 2c should be 2b I think.
- Line 196: 2f should be 2g I think.
- Line 209: G/I should be G/IG I think.
- Line 242: 'millennial-scale changes'. Is it possible to mention in what direction these changes are? If it is possible to be specific here, then I think that is better.
- Line 244: I still do not understand this sentence about the amplification of millennial-scale variability.
- Line 323: 3f should be 3d I think.
- Line 363: 620 kyr should be 621 kyr I think.
- Line 406: From lines 266 – 311 I understood there is not really a relation between SSTs and ACC strength millennial scale variability. Here it is stated there is a relation. Can you explain this a bit more?
- Line 433: 6c should be 6d I think.
- Line 439: 6c should be 6d I think.
- Line 440: I suggest changing 'parameters' to 'processes'.
- Line 513: Here Cape-Horn current is used instead of Cape Horn current. I suggest being consistent over the entire manuscript.

Figures:

- Figure 1: I still disagree with calling the arrow above the Northern Boundary (NB) around 48 degrees the ACC. It falls outside of the definition of the ACC, so it should not be called the ACC in my opinion. I know that in Lamy et al. (2015) they also call the ocean circulation at this latitude the ACC, but also that is, in my opinion, not correct. I suggest calling the arrow the South Pacific Current (SPC) as in Strub et al. (2019).
- Figure 1: The northward current here is named the PCC. In the main text this current is called the Humboldt current (line 127). I suggest being consistent in naming this current.
- Figure 1: there is a typo in the caption: 'Cape Horn Current'
- Figure 2f: why is not a linear fit used?
- Figure 2: Typo in the caption: Antarctic circumpolar current Strength.
- Figure 4: I think subfigures (c) – (f) can be described a bit more in the caption.
- Figure S5: Is it possible to include the locations of the cores in Figure S6i?
- Figure S6: subfigure (i) is not labelled.
- Figure S6: The fronts in subfigure (i) are not mentioned in the caption.
- Figure S8: In the caption it is not explained what the purple, blue and orange markers represent.
- Figure S8 and S11: the use of the colors for the markers is confusing because they have the same color as the CHC/ACC, SST, and planktic d18O lines. I suggest using different colors for the markers.
- Figures 4, 5, 6 and S8: the same color (or at least very similar) is used for the planktic d18O record and the dCO₂/dt records. It might be nice to use a different color for the dCO₂/dt records for clarity.

References:

- Strub, P. T., James, C., Montecino, V., Ruttant, J. A., and Blanco, J. L.: Ocean circulation along the southern Chile transition region (38–46S): Mean, seasonal and interannual variability, with a focus on 2014–2016, *Prog. Oceanogr.*, 172, 159–198, <https://doi.org/10.1016/j.pocean.2019.01.004>, 2019.

Version 2:

Reviewer comments:

Reviewer #2

(Remarks to the Author)

The revised version of the manuscript 790,000 years of millennial-scale Cape Horn Current variability and interhemispheric linkages by Rigalleau et al., has been significantly improved and my concerns have been adequately addressed. Just a minor comment: please define ESP in line 343. I commend the authors on their extensive efforts and recommend their work for publication.

Reviewer #3

(Remarks to the Author)

I thank the authors for the revision. My concerns have been addressed well. I have some minor comments that should be relatively easy to address.

1. Line 78: As I understand it from the sources is that the CHC originates solely from the bifurcation of the SPC into the CHC and the Humboldt Current, meaning I would not mention the ACC here. This is also mentioned in line 272.
2. Lines 214-217: This statement needs a reference.
3. Lines 217-220: This sentence confuses me a bit. The first part is quite clear, though I suggest rewording the sentence a bit; e.g. '... total water transport including wind-driven, barotropic and eddy-induced transport.' I'm not sure what is meant in the second part. As it is worded now, it suggests to me that there are processes active now, that weren't active in the past.

However, I don't think that is what is meant here. I suppose what is meant here is that it is impossible to determine the contribution of for example these three transport mechanisms on the total transport based on the proxies that are used. I suggest clarifying this sentence.

4. Line 222: What 'low-latitude forcings' are meant here?

Reviewer #1 (Remarks to the Author):

(comments by the reviewer are printed in black, and our point-by-point response in blue)

Review of “Persistent interhemispheric link of the Antarctic Circumpolar Current millennial-scale oscillations over the last 800 kyr” from Rigalleau et al.

Rigalleau et al. provide new ACC proxy data with an unprecedentedly high resolution to demonstrate a relationship between the ACC and millennial-scale northern hemispheric variability. The manuscript was nice to review: it reads very well, and figures are fancy and pleasant to read as well.

Unfortunately, as I have very limited knowledge in sedimentology, bio/geo chemistry, and quaternary climate, I’m afraid I had to ask the editor to replace me with another reviewer or to add a new one for the next review. I’ve tried to focus my review on statistical analyses provided by the authors, and I have not found any compromising content for publication in this respect. This is the reason why I only have a list of minor comments the authors must consider, see below. Thanks again for the nice and interesting read.

We thank Referee #1 for the very positive assessment of our paper.

Minor comments:

Referencing in general: Blank spaces before superscript referencing numbers must be deleted throughout the text in order to fit the journal format.

Thank you, we are sorry for this mistake and have deleted the blank spaces in the revised manuscript.

L. 26: I suggest using a less personal phrasing for “we still lack [...]”, such as “there is still a lack of a clear mechanistic understanding of these variations”, or “a deeper understanding of these variations is still lacking”.

Thank you, we agree and rephrased this sentence accordingly.

L. 55: “[...] is the largest current system on Earth.” – Such statement needs a reference.

L. 55: “It is driven by atmospheric forcing” – reference needed.

L. 56-57: “bathymetry and ocean density gradients originating from surface” – reference needed.

This general background information regarding the Antarctic Circumpolar Current refers to the review paper by Rintoul 2018 (Nature). We clarified this point in the revised paper.

L. 65-67: “Various model studies” contrasts with the presence of a single reference.

We agree to this being misleading and deleted “various”. We only cite one paper as an example of a benchmark paleomodelling study that for the first time showed the influence of the Drake Passage throughflow and the Agulhas Leakage on the Atlantic Meridional Overturning circulation. We are aware

that there are other, more recent modelling studies, but prefer to refer to Knorr & Lohmann, 2003 (Nature) in the introduction of the paper.

L. 65-67: It would be worth specifying the time scales at which this influences is exerted in models. In this line, what kind of models the authors refer to? EMICs/GCMs/others?

We agree that a more detailed discussion would be warranted but our paper focuses on the proxy records. There are numerous model studies discussing interhemispheric climate pattern during the last glacial with regard to the bipolar seesaw concept (e.g. Stocker and Johnson 2003, Paleoclimatology).

Figure. 1: There is some information missing in the figure caption:- Indicate that Altimetry-derived ACC fronts are represented with dashed lines. - Indicate in the caption what yellow points are. - Indicated over which period the mean annual sea surface temperatures were calculated.

We agree and added the missing information to the figure caption following reviewers 1 and 3.

L. 112: Replace “an previously” by “a previously”.

Thank you, corrected.

L. 161-162: “the millennial scale variability in ACC strength persist”, there is missing “s” at the end of “persists”

Thank you, corrected.

Figure 2: Orbitally induced spectral peaks are clear for the provided time-series. Could the authors try providing an additional analysis evidencing the millennial-scale frequency they refer to throughout the paper? So far it is only a visual assessment. A wavelet analysis could be able to provide such information.

We agree with Reviewer #1 and added a wavelet analysis (as a new Supplementary figure 10) The wavelet analysis clearly reveals the three major orbital frequencies (precession, obliquity, and the ~100-kyr eccentricity cycle) over the past ~800 kyr in both SST and ACC strength records. Superimposed on orbital variability, we observe substantial spectral power in the millennial band (i.e., 1-10 kyr). In the wavelet spectrum, these changes are evident throughout the record.

L. 180-183: How did these threshold-based events classification was made exactly? From the text it reads that thresholds were somewhat determined arbitrarily (1.6°C and ~10cm/s), where the “~” reads quite surprising as one would expect a threshold to be fixed. I can understand it is difficult to determine a clear approach to determining classification thresholds here, but the authors should consider being more transparent by precisely explaining how these choices were made. In the case thresholds were determined arbitrarily, an obvious subsequent question I would have is: are the following results sensitive to this choice then? Could the authors demonstrate these thresholds were not cherry-picked?

The threshold chosen is empirical, as explained in the methods. We chose it in order to identify almost all of the canonical D-O events during the last glacial cycle, similar to the approach of Barker et al.,

2011 (Science). The procedure of separating major events from all events is explained in Supplementary figure 8. By sorting each event for the respective amplitude, we visually observe for each record two distributions. The velocity is derived from the visual separation (28% ACC strength) between the distributions and the subsequent translation to velocity. We now better clarified our approach in supplementary figure 8g and corrected the main text (methods) accordingly (lines 725-727).

L. 194: I think “1 ka” should be “1 kyr” here? I may be wrong here.

Thank you, this is correct, it should be kyr. We have corrected the text accordingly.

L. Replace “This imply” by “this implies”.

Thank you, corrected.

L. 205-207: A reference is needed to support this sentence. “This imply that the frequency of events is rather constant, even though the magnitude varies with background climate.”

Thank you for this comment. This sentence does not refer to a specific reference but is based on the assessment of the new records from Site U1542. Following a major Reviewer #3 we decided to modify our approach and now refer to complete glacial cycles for our interhemispheric comparisons (lines 366-367 and 391-404, and updated Fig. 4).

L. 209-210, Figure. 3E,F: It is difficult to assume the linear relationship to be significant here. The R2 most likely is very low (given the distance of points with the regression line), and the sample size is pretty small. I would mention this in the present interpretation and add some conditionality in this statement.

Figure 3: Significance tests for regression line slopes for panels C-F are missing – please provide p-values and sample sizes. Use Student t-tests.

Interpretation of Figure 3: Under the assumption that millennial-scale events occur on a regular basis, one would inevitably obtain more events for longer time windows (thus for longer MIS). In other words, what do we learn exactly from panels C and D?

We agree with Reviewer #1 that these plots do not provide new insights. We therefore removed the panels Fig. 3c to f. However, we keep the two panels with the overall distribution of the events in the Supplementary figure 9.

Figure 4: To my knowledge, figures and their respective caption must be self-consistent in Nat. Comm. This means that acronyms and abbreviations such as “D-O”, “AMOC”, or “EDML” must be given in their full version the first time they are mentioned in the caption.

We apologize for overlooking this point. We corrected the figure captions accordingly.

L. 387: Missing “-“ symbols at “state of the-art”.

Thank you, this has been corrected.

L. 423: The acronym for North Atlantic Deep Water (NADW) has not been defined earlier.

Thank you, we use the full name as it is only used once.

L. 442: “SWW wind-driven” reads redundant. Should maybe be SWW-driven?

We agree and deleted “wind”.

Figure 6: See my above comment of Fig. 4.

We thank the reviewer for the careful revisions and corrected the respective text parts accordingly.

Reviewer #2 (Remarks to the Author):

(comments by the reviewer are printed in black, our point-by-point response in blue)

The manuscript “Persistent interhemispheric link of the Antarctic Circumpolar Current millennial-scale oscillations over the last 800 kyr” by Rigalleau et al., provides new high-resolution current strength and sea surface temperatures (SST) records, spanning the past 790,000 years. The sedimentary archive IODP U1542 is located within the Cap Horn Current (CHC) – northern branch of the ACC. The authors report that their records show persisting millennial-scale climate variability throughout all Late Pleistocene, with stronger ACC flow coinciding with warmer SST and Antarctic warm intervals. Additionally, the Southern Hemisphere changes seem to coincide with North Atlantic millennial-scale fluctuations suggesting a close link between ACC flow and the Atlantic thermohaline circulation. Lastly the observed changes in IODP U1542 also seem to follow atmospheric CO₂ changes and for that, the authors propose a mechanistic link through the Southern Ocean carbon cycle.

Variations in the strength and position of the ACC play an important role in global ocean circulation, ocean carbon cycle and thus Earth’s climate. Understanding the mechanistic links between these processes is therefore of major importance. I would first like to point that the total number of samples analyzed for this study, especially the alkenone-based SST is impressive. Overall, the data are high quality and convincing. I also appreciate the way the data are presented - figures illustrate clearly the datasets. These high-resolution records are of great interest and importance to the scientific community and can be used for interhemispheric comparisons (as authors did) providing important insights into the millennial-scale variations of the ACC and their links to Earth’s climate.

We thank Referee #2 for the positive assessment of our paper.

Said that, I unfortunately think that the manuscript, in its current form, does not provide new insights in terms of processes/mechanistic links and the understanding of past ACC variations, compared to what has already been reported by several studies in the same area (e.g., Lamy et al., 2015; Toyos et al., 2020; Wu et al., 2021). Moreover, the new data does not always strongly support the main conclusions. I think that there is more information that can be extracted from these datasets and help in understanding the dynamics of the ACC and their interactions with climate and CO₂ variability during the past 800,000 years. Below I explain in detail my main concerns.

We address the reviewer’s concerns further down in the rebuttal letter.

SST and ACC strength reconstruction from current-controlled deposits.

As IODP U1542 is located close to the Chile continental margin, in the upper slope and thus not in a deep ocean setting, it is expected that sedimentation is largely affected by down-slope processes (turbidites, debris flows, etc), mixed with along-slope processes (bottom currents). Sedimentation rates are five times higher during glacial times compared to interglacial periods. This indicates a major sediment input from land. That could have a great influence in the grain size variations. Using the mean sortable silt grain-size (SS; geometric mean of the 10 – 63 μm silt fraction) versus the sortable silt percentage (SS%, 10–63 μm fraction relative to the <63 μm fraction) can help to investigate whether the sediments are subject to significant bottom current sorting. However, this is not the only test that can be used and given the complex depositional setting, I would recommend to provide more evidence

on whether bottom currents play the main control on sediment transport in the upper slope of the Chile continental margin. In line 139: authors argue for only limited ice-rafted debris impact in their sediments. This can be easily checked using their data by plotting the fine sand (wt %) and the coarser “possibly IRD fraction” (>150 µm; wt %). Additionally, a more detailed investigation on the grain-size and XRF-core scanning datasets can really help. A Principal Component Analysis on the XRF scanning and grain-size data can provide robust insights into the main processes governing the variations in your sedimentary archive. Alternatively, it would be helpful to plot together the SS, SS%, fine sand (wt %), (>150 µm; wt %) and terrigenous elements from your XRF scanning dataset, e.g., Ti, Al, Zr, Rb. This will illustrate potential similar patterns and help to robustly argue for the main processes controlling sedimentation at Site U1542.

We thank the reviewer for bringing this to our attention and completely agree to discuss in more detail sedimentological and geochemical data supporting primarily current controlled processes at Site U1542. We added Supplementary figure 4 highlighting the correlation between different sediment parameters and changed the main text accordingly. These new results clearly support the following discussion and consolidate the whole study (lines 145-160).

Figure 1. Scatter plot of grain size variations and terrigenous elements from XRF scanning datasets and Principal Component Analysis (PCA) performed at Site U1542.

We also followed the advice of Reviewer #2 and performed a Principal Component Analysis (PCA) as suggested (see figure above). The lower part of the figure shows the loadings for the PCA for the Site U1542 XRF and grain size data. Each loading is coloured by its typically associated origin according to Rothwell & Croudace (2015; Developments in Paleoenvironmental Research). Stronger loadings indicate that the element is more affiliated with that component. The PCA data shows that the sediments are primarily of turbiditic/terrestrial sedimentological deposits, subsequently current sorted.

We found that generally PC analyses do not give new insights into the sedimentation process at Site U1542, required for our study. We think that such more detailed sedimentological analyses and discussions are beyond the scope of this manuscript.

Sensitivity of the Southeast Pacific to orbital and millennial-scale climate oscillations.

Orbital and millennial-scale variations of the ACC strength in the Drake Passage region, as well as in the Chilean Margin and have been previously reported covering different time periods: over the 65 ka in the CHC current (Lamy et al., 2015), over the past 140 ka (Wu et al., 2021), over the past 1.3 Ma (Toyos et al., 2020). Thanks to these works, we have a good view on the variations of the ACC orbital and millennial-scale variations of the ACC/CHC strength in the Drake Passage and Southern Chilean Margin. The aforementioned studies provide similar conclusions and mechanistic processes with the current work, describing stronger ACC in interglacial intervals vs weaker ACC in glacial intervals (Lamy et al., 2015; Toyos et al., 2020; Wu et al., 2021), linking them with low AMOC strength (Wu et al., 2021). I believe that with the new high-resolution records presented in this study the authors have the opportunity and advantage to explore in depth these mechanisms. So, I highly recommend to go deeper in their interpretations.

For example, in Lines 232-234: “At orbital timescales over the past 790 kyr, our reconstructed SST variations at the southern Chilean margin largely follow the atmospheric temperature record of the Antarctic ice core EDC and the open Southeast Pacific SST recorded at PS75/034-2 (Fig. 2C-E).” Looking the figure carefully, I would say that this is not really the case. PS75/034-2 follows largely the EPICA Dome C record, but Site U1542 is not always following the same pattern (e.g., MIS 7-6, MIS 13-12). This is a very interesting observation from these new high-resolution data that I believe needs to be discussed in more detail. I agree with the interpretation of the authors for the higher temperatures recorded at Site U1542 compared to PS75/034-2, but there is more to discuss and extract from the comparison between these records.

We agree that a more detailed comparison between our high-resolution U1542 SST record and the more offshore-located, lower-resolution SST record from core PS75/34-2 would improve our interpretations. The original publication of the PS75/34-2 SST record favours the usage of the Uk-based SST data including the C_{37:4} alkenones. Ho et al 2012 (Paleoceanography) argued that the Uk₃₇ based SST record is more similar to the EDC ice-core temperature variations, particularly before the Mid Brunhes Transition (MBT). However, in our study, we applied the Uk₃₇-based SST calculation which is usually used in paleoceanography including the subantarctic South Pacific (Jaeschke et al., 2017). We changed Fig. 2 accordingly.

We also agree that the mentioned studies discuss ACC changes overall in terms of similar climate mechanisms as discussed in our study. This strengthens our interpretations as our new data are consistent with the previous records from the Drake Passage and at the southern Chilean margin. However, our study provides two major advances: (1) We extend the previously shown millennial-scale variations of northern ACC strength changes at our site (Lamy et al., 2015) and in the central Drake Passage (Wu et al., 2021) to a substantially longer time period. This allows us to discuss millennial-scale climate pattern over the past ~800 kyr, i.e. the entire time period of the EDC ice-core. (2) For the first time, at Site U1542 we provide a reconstruction resolving Southern Hemisphere millennial-scale SST changes over this long time period. Both ACC strength and SST records can thus be thoroughly compared to Northern Hemisphere records. In our manuscript, we compare our Site U1542 millennial-scale pattern to North Atlantic temperature, IRD, and AMOC records allowing us to discuss interhemispheric climate linkages with unprecedented length.

The Uk₃₇-based records from our Site U1542 and PS75/34-2 show very similar long-term (orbital-scale) variability. Both records reveal subdued glacial/interglacial amplitudes before the MBT. SSTs at U1542 are overall warmer by 2-3°C consistent with the impact of the relatively warmer Cape Horn Current at the site U1542 SST record. Using the Uk₃₇-based SST record, the differences at orbital scale are smaller and potentially due to age model uncertainties (e.g., MIS13 to 15). In addition, SST changes at precessional time scales appear more prominent at Site U1542 (e.g. MIS7). We observe large amplitude sub-orbital-scale variability in our record compared to PS75/34-2. In addition to the much higher temporal resolution, the sensitivity of the Cape Horn Current to sub-orbital climate change could have been higher. Furthermore, the glacial/interglacial amplitudes are lower in PS75/34-2.

We refined the discussion on the sensitivity of the eastern South Pacific chapter accordingly. We further added a new supplementary showing additional key SST records from the Southern Hemisphere (Lines 192-212 and Supplementary figure 6).

The authors suggest that the decoupling between Southern Ocean and Antarctica southern latitudes (Lines 253-259), resulted in an increase of the meridional thermal gradient, increased moisture to the poles and continue ice growth, but as they mention later this feature is probably not a prerequisite for the ice sheet growth. This part needs further explanation.

We agree with the reviewer that this would need further explanation. However, we think that a more thorough discussion on whether the difference of trend would probably be beyond the scope of our paper and therefore removed this sentence.

Lines 262-264: “Enhanced flow speeds coincide with warmer periods, while reduced flow speeds coincide with colder periods, implying a close coupling between near-bottom current velocity and SST (Fig. S4)”. The correlation the authors show in Fig.S4 did not suggest a close coupling ($R^2= 0.5$). I would say that this close coupling they mention in the text is not actually the case. There are indeed times, where peaks of ACC strength coincide with high SST and warm intervals in Antarctic climate, but there are many times that this is not the case. For the example, the last 100 ka, ACC show a long-term increasing trend whereas SST data from U1542 and more clearly from MD921-20 show a long-term cooling trend coinciding with Antarctic ice core EDC. This is observed in other time intervals as well. This is a very important observation that needs to be discussed.

We agree that at first glance the correlation between ACC strength and SST is not very high. However, the R^2 of 0.5 is based on the 1125 SST data points correlated with the much higher resolved XRF data ($n=7322$) (Fig. 2f). As the XRF data represent only a limited area at the surface of the sediment core and the SST are derived from 1 cm slices, some error is introduced. Nevertheless, the reviewer is right in pointing to a closer look at the differences between ACC strength and SST (e.g., the long-term trends in MIS2-5). We think that in particular, the trends might indicate a partly different behavior of the bottom currents within subantarctic ACC/CHC and the SST responding in parts to different atmospheric (e.g. Southern Westerlies) and oceanic forcings (lines 218-243).

In lines 268-288, the authors propose that latitudinal migrations of the Southern Ocean frontal system during glacial / interglacial times as a primary driver of changes recorded at Site U1542 (located north of the SAF), similar to the proposed mechanism in PS97/85 (located in the polar front). Northward migration of the Southern Ocean frontal system during glacial times should bring SAF closer to Site

U1524, compared to interglacial times. Given that stronger ACC flow appears within /near the SAF. Why don't we see higher ACC flow speeds during glacial times at Site U1542 and vice versa for the interglacial times? In other words, do the authors expect to see the same trends in ACC strength variations in all records: PS97/085-3, PS97/093-2, PS75/034-2 and Site U1542, unrelated to their locations within the ACC frontal system? Why?

We agree with reviewer #2 that latitudinal migrations of the Southern Ocean frontal system might influence the ACC strength at the mentioned sites. However, all sites with different locations regarding the modern fronts register the same glacial/interglacial pattern with reduced ACC strength during glacials and stronger ACC during interglacials. This pattern is consistent with ACC reconstructions in the Central and Southwestern Pacific Southern Ocean (Lamy et al., 2024; Nature) showing the same glacial interglacial reduction/strengthening of the ACC from the Subantarctic Zone, across the Polar Frontal Zone and the even south of the Polar Front, though glacial/amplitudes decrease southwards. Therefore, the observed ACC strength variations are consistent and indicate that frontal shifts are not the main factor in determining ACC strength in our study area and the South Pacific in general. Moreover, even with a substantial northward shift of the frontal system, the Subantarctic Front most likely would have remained south of U1542.

I would recommend to include a figure showing ACC strength and SST variations from all sites discussed in the manuscript (U1542, PS75/034-2, PS97/093-2 and PS97/085-3). This figure will help to clearly identify changes across the whole ACC frontal system in the Drake Passage region.

We agree to the reviewer and now provide two new supplementary figures showing ACC strength and SST variations at the relevant sites (Supplementary figure 5 (for ACC strength) and supplementary figure 6 for SST changes). There is unfortunately no SST record for PS97/093-2 and PS97/085-3 available and an ACC strength record for site PS75/34 has not been published (we are showing the unpublished record below).

Supplementary figure 5. ACC strength changes over the past 800 kyr in the subantarctic Pacific. (A) Reconstructed relative ACC strength variations (compared with Holocene mean values) in the Drake Passage (unpublished data), (B) (Toyos et al. 2020, P&P; Lamy et al. 2024, Nature), (C) on high resolution near the Polar Front (Wu et al. 2021, Nat.comm.), (D) on the Chilean margin (this study) and (E) in the central subantarctic Pacific (Lamy et al. 2024, Nature). Latitude, location related to oceanic fronts and water depth are indicated by identical color to the record.

A persistent interhemispheric teleconnection

I generally found this section difficult to follow as the main message seems to be hidden in the comparison between the different dataset.

Lines 302-303 “Millennial-scale climate events at Site U1542 are found to be contemporaneous with AIM events” A closer look to the figure does not show this. There are peaks in ACC strength that does not coincide with AIM events. Moreover, in lines 311-314, authors report that intervals characterized by stronger ACC coincide with the weakening of the AMOC (during Heinrich Stadials). I would argue that this is not what the figure 4 shows. I suggest that the authors modify the text, so it will describe

more precise the comparison between the different datasets and clearly state the main message of this section.

Following a similar concern of Reviewer #3, we changed our approach now displayed in Fig. 4. Our interhemispheric millennial-scale analyses are now based on glacial cycles (and not fixed 10-kyr windows). We assume that glacial terminations provide the best time marker in the records and minimize the impact of age-model uncertainties. We modify the text regarding the figure displaying the last glacial period (new Fig. 3), which provides an overview of the bipolar seesaw concept, with warming and strengthening in the SH associated with cooling in the NH. (lines 246-247, 356-358).

Role of ACC in CO₂ changes

The authors associate a total 31 major ACC strengthening events with CO₂ rise, 15 of them present a CO₂ rise before an ACC strengthening and 15 events present a CO₂ rise after an ACC strengthening, suggesting a link between ACC strengthening events with CO₂ rise. They propose that strengthening and southward displacement of the SWW, enhancing ACC strength, decreasing Southern Ocean stratification promoting ventilation, resulting in increasing atmospheric CO₂ levels. However, what are the mechanistic processes that occur during the events where CO₂ rise before an ACC strengthening? Are they the same?

This is an important question. Unfortunately, it is difficult to constrain the exact timing of millennial-scale ACC maxima and CO₂ rise in more detail. We do not have a strictly independent age model on millennial-scales allowing to discuss potential leads and lags of CO₂ variations to ACC strength changes. Besides age model uncertainties, there are also non-Southern Ocean related processes affecting atmospheric CO₂. Therefore, we removed the sentence associated and only focus on the associated events in less than 7 kyr, regardless of the exact timing.

Line 405-407: “Moreover, the relative changes in flow strength appear to be positively correlated to the amplitude of CO₂ variations (Fig. S13)”. However, the correlation between ACC strength and CO₂ rise shown in Fig. S13 is rather weak ($R^2=0.3$).

The relatively weak correlation can mainly be explained by one single millennial event at 263.2 ka, characterized by an ACC strengthening of 36.5% and a corresponding CO₂ rise of 31.9 ppmv. The amplitude of this event is high and nearly in the order of changes across glacial terminations, with higher amplitudes in CO₂ compared to the ACC strengthening. By removing this event, r^2 reaches 0.5, which supports the correlation between ACC strength and CO₂ rise.

As mentioned in a previous comment, a figure showing ACC strength, SST variations and productivity proxies XRF-Ca from all sites discussed in the manuscript (U1542, PS75/034-2, PS97/093-2 and PS97/085-3) can help to unravel changes across the whole ACC frontal system (e.g., latitudinal migrations, bottom strength and productivity changes) that can be then linked with CO₂ changes.

We agree to the reviewer and now provide two new supplementary figures (Supplementary figure 5 and 6) showing ACC strength and SST variations at the relevant sites (see also additional figure for the reviewer including the unpublished ACC strength record from PS75/34. The eastern South Pacific and Drake Passage sites have not all been studied for paleoproductivity changes (only PS97/93; Toyos et

al., 2022, CLP). The PS97/93 records suggest a strong influence of micro-nutrient input from the Patagonian ice sheet during glacials. Though with the same sign, these changes are less directly related to frontal movements and shifts of the opal belt as documented in the central and southwestern Pacific Southern Ocean (Lamy et al., 2024, Nature). We agree that these changes are overall linked to CO₂ variations but a more detailed discussion of the productivity records is complex and beyond the scope of this manuscript.

Minor comments

Fig. 1: Please add the location of MD07-3128

Considering the scale of this map, it is unfortunately not possible to graphically distinguish the locations of core MD07-3128 and Site U1542 as they are only two nautical miles away from each other. We added to the figure caption that MD07-3128 is at the same location as U1542.

Fig. 6: Please add AIM events

We added “AIM” events in figure 6.

Line 311: Wu et al 2021, change it to reference number

Thank you, corrected.

Lines 326-329: This part can be moved to methods

We removed this sentence from the main text and moved it to the methods.

Lines 410-411: Please cite figure 6 and add AIM events in the figure.

We cite the respective figure and added “AIM” numbers.

I am sorry for not rating this paper higher. I strongly appreciate the efforts of the authors to generate such high-resolution records, which are of great value.

We thank Reviewer #2 for this final statement and hope that our revisions address the reviewer’s concerns and improve our manuscript accordingly.

Reviewer #3 (Remarks to the Author):

(comments by the reviewer are printed in black, our point-by-point response in blue)

In this paper the authors analyze a new core from underneath the Cape Horn Current (CHC) system that goes back 790 kyr. They find persistent millennial scaled variability in the Antarctic Circumpolar Current (ACC) strength and Sea Surface Temperatures, and relate this to millennial variability in the North Atlantic suggesting that this link is present over the past 790 kyr. Lastly, they also suggest a role of this variability in the carbon cycle in the Southern Ocean with global consequences. I think the paper is in principle very interesting and I applaud the authors for the amount of work done. I also believe that getting a better understanding of the influence of Southern Ocean dynamics on the ocean, the carbon cycle and the entire climate system is very relevant for future climate change since it is a large source of uncertainty.

Thank you, we appreciate this statement.

However, I have some major concerns about some of the aspects of this study. To address these concerns it might be necessary to make significant revisions that can be quite a bit of work and might also change the scope of this paper. It might therefore be better to reject/withdraw the paper and submit it to a different journal or resubmit here. I leave that decision to the editor and/or authors

My first main concern is that it is claimed that the CHC is part of the ACC and it is assumed that CHC dynamics captured in the core provide an accurate proxy for ACC dynamics. I am not convinced this assumption is valid. Figure 1 actually shows that the core used in this study is north of the Northern Boundary and therefore not in the ACC. Also two papers cited (Chaigneau and Pizarro, 2005; Park et al., 2019) state/imply that the CHC is not part of the ACC. I think it matters for the significance of this paper and the overall conclusions of this paper whether this assumption is valid, which in my opinion it is not valid. I suggest that the authors provide a sound argumentation why this assumption is valid.

We thank Reviewer #3 for pointing us to discuss the oceanographic setting of IODP Site 1542. The northern Boundary (NB) of the ACC frontal system is defined by altimetry and is strongly related to bathymetry features (Park et al. 2019). Therefore, the location of the NB in the Southeast Pacific close to the continent is likely to be affected by the Chilean margin, which would explain why the coastal CHC is located north of the NB and, a priori, why the CHC and Site U1542 do not belong to the ACC. Therefore, our interpretation that the CHC forms part of the ACC is primarily based on previous oceanographic studies in the region (Chaigneau and Pizarro 2005), Supplementary figure 1 (modified from Chaigneau and Pizarro 2005) shows examples of surface buoy trajectories over the bathymetry. When buoys reach the coastal region at $\sim 44^{\circ}\text{S}$, the “black drifter” is advected southward into the coastal CHC and returns to the ACC two months later (Chaigneau and Pizarro 2005; paragraph 6). Therefore, this study implies that the CHC is fully associated with the ACC and can be characterized as a northern, subantarctic branch of the ACC. Furthermore, a recent study from Zheng et al., 2023 focuses on the CHC. Based on satellite altimetry (1993–2021, 400 m deep), as well as 2 ocean models, one free-running and one data-assimilating, this study clearly indicates/assimilates the CHC as a component of the ACC. This study further describes the CHC as “a component that acts as an inter-basin conduit, drawing relatively cold and fresh Subantarctic Surface Water from the South Pacific, and injecting it into the South Atlantic.”. This recent study was not cited in the first version of the manuscript, and we include it now to better support this assumption. Together, these two modern studies of the CHC, one

from observation and one from satellite observation and model data strongly support that the CHC can be considered a component of the overall ACC system. The incorporation of the CHC into the ACC has been the baseline of several surface sediment studies (e.g., Vollmar et al. 2022; Wu et al. 2019). The only paleo record of the CHC (Lamy et al. 2015) stated that the CHC is advected through the DP as well during glacial period. The similar glacial reduction observed at the CHC (MD07-3128) and in the northern DP (MR0806) further implies that the CHC is connected to the subantarctic area of the DP. Therefore, the study of the CHC appears to be relevant in quantifying the northern ACC strength changes in the Drake Passage. This extended explanation is part of the Supplementary Fig. 1 caption.

My second main concern is that it is difficult to check the conclusions with the figures provided and how I analyze some figures I would conclude that there does not seem to be a clear relation between some of the variables. First of all, the figures with time axes often have a length of 800 kyr while variability at the 1 kyr scale is analyzed. I suggest besides providing the full length series, also provide zoom in of e.g. ~50 kyr to support your conclusions. Secondly, for me Figure S6 shows that there is no(!) relation between events in the Northern Hemisphere and the Southern Hemisphere because the graphs look very different in my opinion. In the main text Figure S6 is used to argue that there is a clear relation. I suggest that when considering the relations between the different variables either a well discussed statistical test is used and/or a sound physics based mechanism is provided.

We thank the reviewer and improved the respective calculations and figures. In the revised Supplementary figure 6, we changed the time window to compare different records. Due to age model uncertainties, the 10 kyr may be too arbitrary for the interhemispheric comparison, and one event may be put in one box or another depending of both the age model and box definition. Therefore, we changed the windows to full glacial periods which appears to be more realistic regarding age model uncertainties, as terminations represent robust tie points for tuning. In this way, we are confident which cycle belongs to each millennial-scale event. Furthermore, given the length of the comparison, eight glacial cycles seem to be a reasonable number of glacial cycles for interhemispheric comparisons. We used the glacial terminations given by the Lisiecki & Raymo (LR04) marine isotope stack. Fig.4 replaces the 10kyr windows and we changed the text accordingly. Following the reviewer, we now also provide individual figures for each glacial cycle (Supplementary figure 12a-f). With these revisions, we provide better statistical correlations and graphical support for our conclusions. The main text has been changed accordingly (lines 391-404).

My third main concern is closely related to what I describe above. I think there is a lot of variations in timing and amplitude in the events of the different variables and this makes me doubt whether the events are necessarily connected. Events in the variables can be leading, lagging, coincide or not coincide at all without, in my opinion, a good explanation why these events are connected even though they are so different in these aspects. Also the time window used to detect events in multiple variables is a bit large considering that typical ocean timescales are at 1000 – 2000 year. I suggest providing a better argumentation (e.g. statistical tests and/or physics based mechanisms) as to why and how events between different variables are connected. Improving figures displaying the events can already help a lot here.

We agree with the reviewer that the age uncertainties of our records over the past ~790 kyr is too large for a detailed interhemispheric comparison of synchronicity or lead/lag relationships of individual

millennial-scale events. Such comparison is for example possible, albeit difficult, for methane-synchronized Greenland and Antarctic ice-core records or ideally for radiocarbon-dated marine records over the past ~40 kyr. Such records formed the base for the bipolar seesaw concept of antiphased millennial-scale temperature changes in both hemispheres (e.g., Stocker & Johnson, 1998, *Paleoceanography*). This conceptual model based on the findings from the paleoclimate proxy records and paleoclimate modelling, is the “physics-based mechanism” for extending these concepts to a longer time scale, i.e. eight glacial/interglacial cycles. There have been several studies over the last glacial cycle showing that the southern (Antarctic) timing of millennial-scale changes extended to the northern ACC both in the South Pacific and South Atlantic (e.g. Lamy et al., 2004, *Science*; Canipuan et al., 2011, *Paleoceanography*; Lamy et al., 2015, *PNAS*; Barker & Diz, 2014, *Paleoceanography*; Barker et al 2009, *Nature*; and others).

Therefore, we assume that the bipolar pattern extends to the ACC, at least in the Southeast Pacific/Drake Passage region. However, we do not expect that all individual events correlate in detail and we therefore compare rather their numbers and amplitudes in each glacial to the North Atlantic records. To our knowledge, this is the first study of millennial-scale proxy records in the southern hemisphere extending back to ~790 kyr.

Next, I have also some more specific comments/suggestions for the authors:

General:

1. In the text and in the title the word ‘oscillation’ is used frequently. In my opinion, an oscillation has a more or less stable frequency and I don’t think that is the case in your records. I suggest using for example variability instead of oscillation in the title and throughout the text.

We agree, and changed the text and the title accordingly.

2. I don’t see anything about measurement uncertainties in the text and in the figures. Have I missed this? This should be included in my opinion.

We thank Reviewer #3 for this concern. The analytical uncertainties regarding alkenones-derived SST are given in the material and methods (0.23°C), and has been subsequently taken into consideration in the measurement of the amplitude of the events. The reproducibility of the procedure was evaluated using a homogeneous sediment standard, extracted with every batch of samples. The relative analytical errors were below 0.5°C in SST estimates (we added this text to the methods (Lines 637-640). The uncertainties of the ACC strength are relatively small as the analytical error of the XRF Core Scanner for both Zr and Rb is small. We have not measured these elements wet-chemically but the signal-to-noise ratio for both elements has been previously shown to be excellent as those “heavy elements are precisely detected by X-ray fluorescence, even at low concentrations (Wu et al. 2020, G-cubed). The uncertainty of the grain-size measurements is also low and does not impact the final result, as it serves only for calibration, the XRF $\ln(\text{Zr}/\text{Rb})$ being used for subsequent analyses. Furthermore, the linear equation used for the calculation of sortable silt is sufficiently defined. we a text into the methods (Lines 685-689

Text:

1. Also based on previous comments, I think the title does not accurately capture the content of the paper in my opinion. I think that oscillations could be replaced with variability, the ACC with CHC,

and 800,000 with 790,000 since that is the length of your core. Also, as mentioned previously I have my doubts about the interhemispheric link based on this study.

Thank you. We have revised the title accordingly.

2. Line 55-57: I think this sentence can be clarified a bit. It's not clear to me if the atmospheric forcing is just the Southern Westerly Winds or also temperature and freshwater forcing. It is also not clear whether the salinity changes also relate to the ocean density gradients because of the comma before 'and' at the end of the sentence.

Thank you for pointing out the mistaken use of the comma. We have revised the text accordingly (lines 55 to 57).

3. Line 109 – 111: Can you refer to a figure here?

We now refer to figure 2c.

4. Line 111 – 113: I'm not sure if I understand this. Is there a bias in the measurements? Is this bias compensated for? Why not if it's not? I suggest clarifying this phrase.

No, we do not think that there is an analytical bias in the measurements. The late Holocene value is above the modern annual mean, mainly because alkenone SSTs in the region are seasonally biased, i.e. towards summer temperatures (e.g. Hagemann et al., 2023, CLP).

5. Line 197: 'amplified', I think you mean here that events are more likely to occur in these ranges, so I suggest changing the text to reflect this.

We apologize for the erroneous use of "amplified" and changed the sentence accordingly (lines 314-317)

6. Line 199: you make a reference to ice sheet volume in Fig. 2, but there is no ice sheet volume in Fig. 2 which is a bit confusing. I suggest being more accurate in this statement.

We thank the reviewer and reworded the text accordingly.

7. Lines 279 – 288: I don't understand where to look for the amplification in Fig. 4. How does this amplification look in the data? I also don't understand how this supports the idea of SWW shifts as a primary driver. How I would reason (tell me if I'm correct in this): when SWW shifts southward, the water flowing into the CHC becomes colder, when the SWW shift northward, the water becomes warmer because of the latitude the source waters come from. But I don't understand now where to look in the data for the SST response in U1542. I suggest clarifying the text and specifically the link with Fig. 4.

We thank the reviewer for bringing up this important issue. We are referring mainly to the high amplitudes of SST changes along Chilean margin SST records, both on millennial-scale and orbital time-scales. This has been shown in a number of previous studies. Compared to e.g. Antarctic ice-core

records, SST changes across the last glacial termination and millennial-scale events were relatively sharp which has been suggested to indicate a stronger atmospheric linkage to the Northern Hemisphere involving shifts of the Intertropical Convergence Zone and the Southern Westerlies superimposed by SST changes through ocean circulation (including the “see-saw”) (e.g., Anderson and Carr, 2010; Science, Lamy et al., 2007; EPSL). In this scenario, the westerlies shifts southward during NH coolings, resulting in enhanced advection of relatively warm Cape Horn Current water. The contrary occurs during SH coolings. We think these concepts explain the high amplitude in our SST record reasonably.

8. Line 382 – 388: Here you relate your results to results from ESMs. I do not agree that these results can be compared. You look at completely different timescales. The timescales assessed here are 1 – 10 ka, while the ESM studies are on centennial time scales. The processes are therefore, in my opinion, not translatable. I do think that you can mention it in the text, but I suggest being more nuanced here.

We find the comparison with the impacts of anthropogenic climate change from observations and ESMs reasonable because the magnitude of change, in terms of temperature rise, for example, is comparable despite the different time scales. We find that scepticism over whether the same processes are occurring on such different timescales is reasonable, but we do not think that this precludes it as a discussion point. Rather, it is an interesting opportunity for further research and we find the transdisciplinary comparison relevant for a journal such as *Nature Communication*. However, we recognize that this part can be more nuanced and we have added an acknowledgement of the distinct time scales and highlighted the need for further research at line 454.

Figures:

1. Figure 1: This corresponds to my main concern 1. There are arrows drawn north of the NB line that are called the ACC. Looking at the paper where the data is from, they call the NB line the actual northern boundary of the ACC. I suggest changing the name to SPC as in Fig. S1 or motivating very clearly why you consider this to be the ACC.

Please see our response regarding concern 1

2. Figure 1: No source of the SST data is given, and it is not clear what period the SSTs represent. I suggest adding both in the caption.

We now give the source “observations from all months of all years” for the period 2005-2017, as described by (Locarnini et al. 2018), citation also cited in the figure caption.

3. Figure 3: This figure misses clear labels and units for the axes. It is therefore difficult to interpret this figure. On the left ‘ACC strength’ and ‘SST’ look like labels, but it took me a couple of looks to understand they’re not

This figure has been removed following reviewer’s concerns.

4. Figure S3: Caption does not clearly explain the subplots. In particular, LSR (local sea level rise?) is not mentioned.

LSR is the acronym for Linear sedimentation rate and is now mentioned.

5. Figure S7: I think it would be useful to be able to better compare the timing of the events by plotting the events of the different variables closer to each other. The time between 2 events could for example be shown by changing the marker size.

The figure has been removed and replaced by Supplementary figure 10 (evolutive spectral analysis) which better shows the frequency of millennial-scale variability.

6. Figure S9: Shouldn't n be 103 in the bottom subplot?

Has been corrected (the figure is now Supplementary figure 8).

REVIEWER COMMENTS

Reviewer #2 (Remarks to the Author):

The manuscript titled " Persistent interhemispheric link of the millennial-scale Antarctic Circumpolar Current variability over the last 790 kyr" from Rigalleau et al., presents a high-resolution current strength and sea surface temperature records covering the past 790,000 years from the Chilean Cape Horn Current, off the Chilean coast.

The authors present evidence of persisting millennial-scale climate variability throughout the last eight glacial periods, characterized by stronger current flow and warmer sea surface temperature, coinciding with Antarctic warm intervals. They interpret these changes as being linked to the Antarctic Circumpolar Current (ACC) and suggest a link to North Atlantic, proposing persisting interhemispheric dynamics over the last 790 kyr. Lastly, they associate the variability in current strength with atmospheric CO₂ fluctuations, suggesting a role of this variability in influencing the Southern Ocean carbon cycle.

This is the second time I have reviewed this paper, and overall, the results of the study are interesting, with an impressive amount of work and analyses. The data are of high quality, and the figures clearly illustrate the datasets. This work could make a significant contribution to our understanding of the Southern Ocean climate, ocean and carbon cycle dynamics, and its broader connections to Earth's climate system.

We thank Reviewer #2 for the positive assessment of our paper.

While the manuscript presents valuable insights, I believe it is not yet ready for publication. There are still major concerns that need to be addressed, particularly in clarifying interpretation of the results and improving the flow and readability of the text. I have outlined my concerns and specific recommendations below to help strengthen the manuscript.

My first main concern remains whether the grain-size changes at U1542 can be linked to changes in the ACC strength. This is due to the location of the site which is situated very close to the Chilean coast, in the upper slope and thus its sedimentation is likely primarily driven by downslope processes related to continental sediment inputs from meltwater transport and sea-level changes, with current reworking playing a lesser role. You mention that the expansion of Patagonian Ice Sheet that reached the continental shelf edge at glacial maxima may explain the interhemispheric differences you observe between your records and others (e.g., 364-367 "the interhemispheric disparity in several cycles..."). This raises the question of whether the observed changes in sediment composition, grain-size, are truly driven by variations in bottom currents or rather by downslope processes. If the latter is the case, the grain-size changes may not serve as reliable indicators of changes in current strength.

We agree with the reviewer that downslope processes might play a role in peak glacial periods, as we mentioned in the manuscript. Given the new grain-size analyses (see next point below), we assume that current strength is, however, the major factor for most of the record, as indicated by the overall excellent correlations between the different grain-size parameters (see below). Downslope processes may have some importance during glacials, when finer grain-sizes might be influenced by glacial sediment supply. Therefore, we have extracted all samples with SSFS < 30 μm (primarily representing glacial intervals) and plotted the mean SS versus the weight percentage of SS separately (Supplementary Fig. 4f). The comparably good correlation between mean SS and percentage of SS ($R^2 = 0.59$) suggests an only moderate influence of terrestrial sources due to current reworking, even during glacial periods. These data are consistent with the Patagonian Ice Sheet reaching the outer Magellan Strait only during peak glaciations as shown, for example, with the high sedimentation rates during the last glacial (*c.f.* Supplementary Fig. 3) (e.g. Lamy et al., 2015; Hagemann et al., 2024). At least during most parts of intermediate and interglacial periods, downslope transport of terrigenous sediments was less important, as a large amount of sediment was trapped within the fjords and never reaches the shelf break. During

these intervals, sorting by currents gained more importance and can, in our opinion, be used as current strength indicators.

The moderate correlation ($R^2 = 0.43$) between the mean SS and the weight percentage of the SS (Supplementary Fig. 4e) may also suggest that current reworking is not the primary control driving grain size changes within the silt fraction.

We address the reviewer's concerns in the following discussion.

Additionally, out of the 94 grain size (SS) measurements, fewer than 10 samples were measured from the high peaks in XRF Zr/Rb, with only one sample measured for each major peak (Supplementary Figure 4b). This limited sampling could bias the statistical analyses, as well as the correlation between Zr and mean SS and the calibration of the XRF-Zr/Rb using discrete SS measurements. To provide strong evidence that mean SS and XRF-Zr/Rb can be used as proxies for near bottom current strength in such a dynamic environment, I recommend that the authors provide additional grain size measurements at these prominent XRF Zr/Rb peaks to improve the robustness of their analysis.

We thank the reviewer #2 for his/her suggestions to improve the robustness of the grain-size analyses. We have measured 39 additional samples from the coarser part of the record, i.e. interglacial sections in the high peaks of XRF Zr/Rb. These new measurements substantially improved the correlation coefficients if one extends the traditional sortable silt range to 10-125 μm (sortable silt plus fine sand (SSFS)). This approach has been successfully applied for other strong bottom current regimes as in the central Drake Passage (Wu et al., 2021) and fine sand has been included in grain-size based current reconstructions in the site survey core (Lamy et al., 2015).

Using SS/FS yields significant correlations between mean SSFS grain-size and the weight percentage of the SSFS (Supplementary Fig. 4b; $R^2 = 0.91$). Using the new SS/FS data, we now use the current velocity calibration from the Drake Passage (Wu et al, 2021, *Nature Comm.*). While the absolute flow speed changes with this new calibration, the relative changes in strength does not, and therefore the results of our subsequent analyses remain unaffected.

Moreover, there are notable differences in the timing and amplitude of ACC current strength changes over the past 800 kyr between U1542 and records from the Drake Passage and Central South Pacific (Supplementary figure 5). The latter sites are strategically positioned within the main ACC pathway and exhibit similar changes, while U1542 does not follow this pattern. Could the proximity of U1542 to land have influenced its grain size patterns, given the variable sediment sources and sedimentary/depositional mechanisms?

While the general glacial/interglacial current strength variation patterns are similar (enhanced currents during interglacials versus reduced glacial current strength), we agree that some detailed pattern between the upper continental slope IODP Site U1542 and the deep ocean, lower-resolution sites are partly diverging. This applies to MIS 5, but also to most of the other peak interglacials, which stand out more prominently, with more abrupt increases in current strength from the overall glacial background. We speculate that the shallower water depth and the particular strength of the Cape Horn Current (CHC) might contain thresholds above which the current strength strongly increases in a nonlinear mode.

As mentioned above, the current sensitivity of some of the peak glacials may be lower compared to interglacials, most likely related to the proximal ice sheet and sea level. Nevertheless, we observe strong millennial-scale current strength peaks throughout the record.

We added discussion on this important point and the uniqueness of the CHC, partly diverging from the ACC changes in the main text and supplements (lines 212-237). The relation between CHC and ACC is discussed below in relation to various valid comments of Reviewer #3.

My second concern is the “persistent” connections proposed between U1542 records and those from the Northern Hemisphere and Antarctica (EPICA). Given the significant discrepancies in timing, duration, and amplitudes of events, it’s unclear whether a persistent interhemispheric link over the last 790kyr truly exists.

Beyond the last glacial cycle and the radiocarbon range, age model uncertainties prevent direct one-to-one comparisons between different records. Concepts like the bipolar seesaw further complicate interhemispheric comparisons. Consequently, we do not address the precise “timing” in the comparison between Sites U1542 and U1385 beyond the last glacial period.

We focused on comparing events within a single glacial cycle, separated by terminations, as they provide the most reliable markers for graphical correlation across records. Comparing the number and amplitude of events for each cycle offers a more robust approach to mitigate age model uncertainties and minimize subjective interpretation. Using the threshold method further helps to distinguish events from potential background signals, and enable to compare different proxies such as SST, current reconstructions, and planktic oxygen isotopes. Enabled by these two long proxy records (U1542 and U1835) allow for the long-term comparison of millennial-scale climate variability, the novelty of this study lies in the comparison of event amplitudes (Figure 4c-d) and frequencies (Figure 4e-f), which suggests that interhemispheric links as observed during the last glacial period (Epica Community Members, Nature 2004) persisted back over the past 790 kyr. The moderate to good positive linear correlations (Figures 4 c-f) indicates that periods of high variability in one hemisphere are reflected in the other.

We compared CHC/ACC strengthening events to atmospheric CO₂ rise events in a different manner, as both records follows a Southern Hemisphere pattern. Site U1542 and the Eastern South Pacific follow an "Antarctic timing," and both CO₂ and CHC events variability can be considered contemporaneous on millennial timescales. In Figure 6, only major CHC events are compared with 2-kyr average CO₂ variability, minimizing interpretative uncertainty. The one-to-one CHC to CO₂ comparison is thought to be speculative, but demonstrates that the magnitude of millennial-scale events and terminations differs, suggesting that factors other than ACC reacceleration also contribute to increasing the atmospheric CO₂ levels during terminations.

While I appreciate the effort in presenting such extensive high-resolution datasets, I find the explanations provided for these differences to be insufficiently robust. Much of the text is focused on reporting these discrepancies, without adequately explaining them, often offering only a single sentence to explain these differences. Below I provide some examples. I believe that refining the text and clarifying the main message would greatly improve the paper’s readability. As it stands, the current version is difficult to follow.

We thank Reviewer #2 for asking us to improve these text sections. We carefully rewrote these parts and tried to more adequately explain the reasons for discrepancies discovered during our data analyses.

-Lines 206-207: “This decoupling between the SH mid-latitudes and Antarctica likely resulted in an increase of the meridional thermal gradient at these periods”

We thank the reviewer to pointing to this sentence. We removed it, as it may have been too speculative and difficult to assess with the existing records.

-Lines 216-218: “This partly different behaviour of the bottom currents within subantarctic ACC/CHC and the SST might indicate a different response to atmospheric (e.g., SWW) and oceanic forcing (e.g., see-saw).

We rephrased and clarified this part (lines 301-308).

-Lines 277-278: “These differences might be due to varying millennial-scale sensitivities and thresholds for SST and ACC strength changes, related to various atmospheric and oceanic forcings (see below)”. However, the authors do not explain later these” various atmospheric and oceanic forcings”.

We added a more detailed section on the CHC based on more recent oceanographic studies. These new studies discuss the various forcings of the CHC, including the ACC and low-latitude atmosphere-ocean mechanisms presently controlling the ACC. We would like to refer to our new text section (lines 212-237) and the text in lines 301-308.

My last concern is regarding the mechanism proposed for the changes recorded at Site U1542. Between lines: 228-253, the authors suggest that shifts in the Southern Westerly Winds (SWW) are the primary driver of the observed changes at Site U1542. Later this mechanism is used to support the role of ACC in CO₂ exchanges, for examples in lines 468-470: “...strengthening and southward displacement of the SWW, enhancing ACC strength, decreasing Southern Ocean stratification, promoting ventilation, and consequently, increasing atmospheric CO₂ levels.

However, in their response to reviewer #2, the authors state that “the observed ACC strength variations are consistent and indicate that frontal shifts are not the main factor in determining ACC strength in our study area and the South Pacific in general. I may have overlooked something, but I find myself unclear about the specific mechanism proposed as the driver of the observed changes in their record.

Similar to the reviewer’s previous point, we would like to refer to the now improved explanation of oceanographic processes and forcings of the CHC.

Other comments

The authors should thoroughly check references in both the text and reference list. For instance, in line 146, citations like (Davies et al. 2020, Hagemann 2024) need to be converted to numerical format. Additionally, in line 807, please note that Hagemann et al. 2022 refers to a preprint, the peer-reviewed version was published in 2023.

Thank you for the careful revisions, we checked the reference list accordingly.

Line 159: “Fig 2c”. I believe the authors refer to Fig. 2b.

Thank you, done

Line 223: Please define the abbreviation “PIS”

We thank the reviewer for the careful revisions and corrected the respective text parts accordingly. PIS is now defined above (line 135)

Reviewer #3 (Remarks to the Author):

The revised version of the manuscript has improved and some of my concerns have been adequately addressed. However, I still have some concerns about some of the analysis and assumptions made in this study and some small points to clarify the manuscript a bit more.

We thank Reviewer #3 for the positive assessment of our revised manuscript.

Before I address my concerns, I have a tip for future submissions (not only for this paper). I like to clarify that it is not meant as criticism and that I did not let this influence my review of the revised manuscript. Some references in the text to figures and references to line numbers in the response are not correct. I also feel like there are more typos in the revised version. Firstly, the incorrect references make it more difficult for me to review the paper because I have to do more work to see where the revised text is, and what figure is actually meant. Secondly, it makes it feel like the revised version was a bit rushed and I think that is a shame, because it is obvious you have spent a lot of time on this study. For the future, I suggest being more thorough on the last check before submission.

Thank you for your communicating this point to us. We are sorry for the typos and other mistakes. We have now checked formatting and layout more carefully.

My main concern is still how the measurements of the Cape Horn Current (CHC) are framed as being representative of the Antarctic Circumpolar Current (ACC). I am not satisfied with the arguments given in the response. To address the arguments in the response:

(1) The authors refer to the study of Chaigneau and Pizzaro (2005; CP05) where an example of a drifter (black drifter in Fig. S1) is provided that leaves the ACC around 100W and returns to the ACC through the CHC. Based on CP05 I would say there are probably a few more drifters that follow a similar trajectory. However, a few drifters do not necessarily reflect the mean flow in this region. Compared to a more recent paper (Strub et al., 2019) I would rather say they are probably not representing the mean flow accurately. I therefore believe that the quality of this paper is insufficient to argue that the CHC is a northern branch of the ACC.

We agree that the CP05 study based on drifters is not optimal to infer longer-term mean circulation dynamics, which would be most relevant for past current strength reconstructions on geological time-scales. We therefore also included satellite altimetry data from Zheng et al. (2023) in the previous version of the manuscript. However, the reviewer correctly noted that we should likewise consider the comprehensive satellite altimetry and model study by Strub et al. (2019). While Strub et al. (2019) primarily focus on the coastal ocean mainly further north (38°S-46°S), their map with geostrophic velocities (Strub et al., Fig. 1) also extends south to the CHC area to ca. 55°S. This map thus includes the CHC area and, in accordance with the drifter data, shows the strong current along within the CHC before entering the Drake Passage. This is likewise consistent with an additional recent remote sensing study in the area by Saldias et al. (2024; *Remote Sensing*).

The CP05 study is probably not conclusive in linking the CHC to the subantarctic ACC as required to characterize the CHC as a northern branch of the ACC. We now reframed our paper accordingly towards common forcing mechanisms of the CHC and the subantarctic ACC which we discuss below referring to the reviewer's points (2) and (3).

(2) The authors also refer to the study of Zheng et al. (2023). However, I believe the citation to be inaccurate. In Zheng et al. (2023) I do not think they say that the CHC is part of the ACC. What they do say is that the CHC is part of the ACC system. A subtle difference, but in my view a very important one.

(3) Other paleo papers are cited where the CHC is also referred to as being part of the ACC. I did not

look thoroughly at these papers, but in my opinion, if they argue that the CHC is part of the ACC, then I think those claims are inaccurate.

We comment on points (2) and (3) together. We agree with Reviewer #3's view that the CHC cannot be strictly considered as being a part of the ACC. We therefore adapted the wording that the CHC is part of the ACC system (lines 84-85). As such, the CHC is located in vicinity of the Northern Boundary (NB) of the ACC (after Park et al., 2019) and is in this sense connected to the northernmost ACC.

In point (3), the reviewer mentions paleo studies involving the CHC. We assume that this mainly refers to Lamy et al., 2015 (*PNAS*) based on sediment-core MD07-3128 located close to our IODP Site U1542. Lamy et al. (2015), for the first time reconstructed CHC current strength on geological time-scales. At that time little has been published on the CHC in the modern oceanographic literature. One such study is Boisvert (1969) describing the prominent poleward flow of CHC reaching to deep water depth along the continental margin (i.e. the water depth of core MD07/3128 and Site1542)

The original CHC view by Lamy et al., (2015) involved interpreting the deep reaching CHC as an important component of the "return" flow to the ACC (also described in Well et al., 2003 (*Deep-Sea Research*). The subsequent drifter study (CP05) then provided further indications of the CHC being part of the ACC "system" as the larger Southeast Pacific circulation is described in there.

The more recent remote sensing-based studies by Strub et al. (2019), Zheng et al. (2023), and Salidas et al (2024) provide a more detailed view on the CHC. Salidas et al (2024) and Zheng et al (2023) suggest that the CHC is mainly driven by the barotropic component of the pressure gradient through sea-level. A major implication for paleoceanographic current reconstructions in general (of e.g. the ACC system) is the adjustment time-scale for changing the density structure of major currents. For the open ocean ACC in the central South Pacific, sortable silt changes generally correspond to the total water transport including wind, baroclinic, and eddy-induced, as even high-resolution core scanning (Zr/Rb data) records still average across centuries. Lamy et al. (2024, *Nature*). Though adjustment times of the CHC may be shorter and sedimentation-rates are higher (and thus our temporal resolution of the records), we still assume that these basic assumptions are also valid for the Chilean margin.

Though beyond the resolution of our paleo records, the pronounced seasonal changes in the CHC and the northern adjacent area to beyond 40°S, described by Salidas et al. (2024), provide evidence for the involvement of large-scale atmospheric and oceanic circulation changes. During austral winter and fall, the southward meridional transport, characteristic for the CHC, extends significantly further north and retreats southward during austral spring/summer. These large seasonal changes are connected to the seasonal migration of the South Pacific Gyre and are today related to the Southern Annual Mode (e.g., Qu et al., 2019; *J. Geophys. Res. Ocean*) and thus the Southern Westerly Wind belt. For paleoceanographic time-scales, these seasonal changes imply that a northward extension of subantarctic ACC waters (probably connected to a stronger South Pacific Current (SPC)), would be consistent with a relative northward migration of the Chilean bifurcation during cold periods. Conversely, relatively warm periods would be characterised by a poleward movement of the ACC, SPC and westerly wind circulation. These mechanisms could serve as analogues both for orbital-scale (glacial-interglacial) time-scales and the short-term millennial variations discussed in our manuscript.

Based on these studies (including the PhD study by Qi Zheng (2023) providing additional more detailed information), we changed the overall framing and now discuss our data as reflecting CHC variability controlled by deep ocean forcings (i.e. ACC) and lower latitude processes. We changed and extended the text in several parts (mainly lines 77-85, lines 213-238, lines 302-309).

My main two concerns here are:

(1) The source of the waters in the CHC. In the manuscript CP05 is used to argue that the waters in the CHC are of ACC origin. However, the sea surface temperatures (SSTs) in Fig. 1 show in my opinion that the water is more likely to be of subtropical origin, which I think is also in line with what Strub et al. (2019) show. Yes, there will be leakage and mixing from the ACC (CP05 show this in my opinion), but based on the SSTs I would argue that most of the waters are of subtropical origin.

Yes, it is possible that the SSTs at our CHC site are influenced by low latitude sources. Following the location of the altimetry-defined Northern Boundary (of the ACC, which separates it from the adjacent subtropical gyres and the SPC), we agree that the surface characteristics of the CHC are probably influenced by both subantarctic water masses approaching South America south of the “bifurcation” as also shown by Strub et al. (2019), and subtropical sources from regions north of the NB and the bifurcation. For our bottom current speed reconstruction at ~1000 m water depth, we likewise assume subantarctic water masses with Antarctic Intermediate Water formed in the Southeast Pacific relatively nearby. In addition, we might have contributions of deep slope water originating substantially further north of the SST and wind bifurcation around 40-45°S (Well et al., 2003; *Deep-Sea Research*). We added a paragraph to the text (lines 222-238).

(2) From Zheng et al. (2023) we can see that the CHC is at best 10% of the volume transport through the Drake Passage. Therefore, I find it difficult to believe that variability in the CHC is representative of ACC variability.

Though we think that 10% of the world’s largest current is still quite significant on a global scale, we agree that our wording “representative of ACC variability” might mislead some readers. We clarified that the CHC variability at geological time-scales would plausibly parallel variations of the northernmost ACC, i.e. primarily in the subantarctic zone in the South Pacific and not on a global scale. Our records show that the millennial-scale variations of CHC strength parallel millennial-scale variation of ACC strength in the Drake Passage close to the Polar Front (Fig. 3). Both records show strengthened currents during the major Antarctic warm events (Antarctic Isotope Maxima - AIMs) of the last glacial period. These observations imply that millennial-scale variations of the CHC were similar to those within the ACC south to the Polar Frontal Zone, at least in the Drake Passage.

I think the main conclusions of this paper are based on the assumption that you are measuring ACC variability in the CHC sediment core. I think this assumption is false and I am not convinced by the arguments provided. The variability could be representative of Southern Hemispheric variability, but that is not how the paper is framed. For me to recommend this paper for publication a major revision on how the paper is framed is necessary where the assumption that ACC variability is measured in the CHC sediment core is dropped.

I do want to note that I appreciate that in the text the measurements are referred to as CHC/ACC strength. I think this is well done. However, it is not done consistently throughout the full text and for all figures.

As discussed in more detail above, we agree with this general statement of Reviewer #3. We therefore, changed the overall framing and now clearly separate variability of the CHC from that of the subantarctic ACC. However, both might, at least partly, reflect similar atmospheric and oceanic forcing mechanisms. Therefore, we keep our comparisons of CHC variations to more pelagic ACC records and discuss major differences and their implications in more detail. We rewrote the relevant parts of the main text, figure captions, and supplementary information.

Major points:

- Line 69: I feel like sources 24, 25 and 26 are wrongly cited here. I must admit I only skimmed through the papers but the general message I think of all three papers is that Southern Ocean warming affects AMOC stability. None of them mention millennial scale variability in the Drake Passage throughflow or the Agulhas current (is Agulhas leakage meant here by the way?) as triggers for Dansgaard-Oeschger events as is suggested by the main text. Why are these papers cited here/what message should these citations convey?

We agree that this sentence does not relate to the references 24, 25, 26 (Buizert et al., 2015, Oka et al., 2021, Knorr and Lohmann, 2003). As mentioned by the reviewer, these studies discuss Southern Ocean warming affecting AMOC stability. We now adapted our sentence to clearly mention the role of Southern Ocean temperature affecting AMOC, separate from the link between Agulhas leakage (Beal et al., 2015) and the Drake Passage throughflow (Wu et al., 2021), which potentially affect AMOC. We replaced Agulhas current by Agulhas leakage (lines 64, 66).

- As I understand it correctly now, there is no relation between the millennial scale SST variability and the millennial scale CHC variability? (lines 266 – 311). Do I understand this correctly?

We are sorry for the potential misunderstanding here. We rewrote this part and now state that the CHC strength events are not in all cases corresponding to SST warmings. In light of Reviewer #3's valid point that at least part of the SST signal might originate from the low latitudes (today), and the CHC strength is primarily a deeper water signal with different thresholds of change, these deviations would be easier to explain. We changed the text accordingly (line 302-309).

- Like my first review on the relation between SST events and CHC/ACC events, Fig. 6 shows to me that there is not necessarily a relation between CO₂ events and CHC/ACC events. Also here there are many events only occurring in one of the two variables, sometimes the CHC/ACC event is leading, and sometimes the CO₂ event is leading. I suggest presenting the CHC/ACC – CO₂ events in a similar fashion as the CHC/ACC – SST events.

Although we acknowledge that comparing interhemispheric variability by glacial cycle is more reliable to address age model uncertainties, we directly compared CHC/ACC strengthening events to CO₂ rise events, as both records are located in the Southern Hemisphere.

Similar to a concern from Reviewer #2, while interhemispheric comparison poses greater challenge (e.g., bipolar seesaw), Site U1542 and the Eastern South Pacific follow an "Antarctic timing," with both CO₂ and CHC events considered contemporaneous on geological timescales. In this comparison we highlight that the magnitude of millennial-scale events and glacials terminations differs, particularly for to CO₂ rise. This suggests that factors other than ACC reacceleration also contribute to increase the atmospheric CO₂ levels during terminations. Following Reviewer #3 suggestion, we now present in figure 6 the CHC/ACC– CO₂ events in a similar fashion as the CHC/ACC – SST events. We keep the previous figure as a new supplementary figure to detail the correlation of millennial-scale events that are appearing in both record in less than 7 kyr (corresponding to the suborbital limit), as only major CHC events are compared with 2-kyr average CO₂ variability, which minimizes interpretative uncertainty. Although more speculative, this one-to-one CHC to CO₂ comparison suggests a correlation at millennial timescales.

- Line 492: '... our findings shed light on the impact of the DP throughflow on Atlantic circulation on millennial timescales...'. Can you explain to me what is meant with this statement? Because I did not fully get that from this study.

We rephrased this sentence to better clarify that we provide evidence that the last glacial CHC changes paralleled variations in Drake Passage throughflow and were linked to changes in Atlantic circulation (i.e., Wu et al., 2020 *Nat. comm.*), most likely extending back to ~790 kyr based on our new data.

- In view of open science, I encourage the authors to also publish the code used to make the figures and perform the analysis with.

We used published scientific software such as Analyseries (Paillard et al., 1996), and PAST (Hammer and Harper 2001, Pal Elec). The maps are realised with Ocean Data View (Schlitzer 2022, <https://odv.awi.de/>). Other figures are realised using the commercial software Grapher. We added this information to the methods.

Minor points:

- Line 71: 'abrupt changes' does not sit well with me. Abrupt changes with respect to what? I suggest changing this sentence to something like: '... understanding of millennial scale dynamics in the Southern Ocean is ...'.

Thank you, done

- Line 127: '... the major part deviates into the southward branch, the CHC.' This needs a reference.

Thank you, done

- Line 148: Use capitals for Supplementary Figure 4.

We now give a more complete equation for each step, thank you

- Line 159: 2c should be 2b I think.

Thank you, done

- Line 196: 2f should be 2g I think.

Thank you, done

- Line 209: G/I should be G/IG I think.

Thank you, done

- Line 242: 'millennial-scale changes'. Is it possible to mention in what direction these changes are? If it is possible to be specific here, then I think that is better.

We specified, mentioning here the millennial-scale strengthening in ACC strength

- Line 244: I still do not understand this sentence about the amplification of millennial-scale variability.

The amplification refers to the previous sentence and to the higher amplitude of the millennial-scale changes in more coastal oceanic settings (i.e. CHC) and likewise because of the higher sedimentation-rates. In contrast, more pelagic sediment records are either affected by smoothing through bioturbation or generally limited resolution to resolve millennial-scale climate change. We rephrased the sentence accordingly (lines 268-269).

- Line 323: 3f should be 3d I think.

Thank you, done

- Line 363: 620 kyr should be 621 kyr I think.

Thank you for your precision, done

- Line 406: From lines 266 – 311 I understood there is not really a relation between SSTs and ACC strength millennial scale variability. Here it is stated there is a relation. Can you explain this a bit more?

We rephrased to clarify this sentence (lines 485–487) which links our interhemispheric comparison (i.e., AMOC, SST and CHC) to recent modern observations. We think that this part is relevant to discuss our results with modern observations.

- Line 433: 6c should be 6d I think.

Thank you, done

- Line 439: 6c should be 6d I think.

Thank you, done

- Line 440: I suggest changing ‘parameters’ to ‘processes’.

We agree that ‘processes’ is a better fit in the sentence

- Line 513: Here Cape-Horn current is used instead of Cape Horn current. I suggest being consistent over the entire manuscript.

Thank you, done

Figures:

- Figure 1: I still disagree with calling the arrow above the Northern Boundary (NB) around 48 degrees the ACC. It falls outside of the definition of the ACC, so it should not be called the ACC in my opinion. I know that in Lamy et al. (2015) they also call the ocean circulation at this latitude the ACC, but also that is, in my opinion, not correct. I suggest calling the arrow the South Pacific Current (SPC) as in Strub et al. (2019).

We changed figures 1 and supplementary figure 1 in agreement with the oceanographic study of Strub et al., 2019.

- Figure 1: The northward current here is named the PCC. In the main text this current is called the Humboldt current (line 127). I suggest being consistent in naming this current.

Thank you, we now are consistent using Humboldt Current, commonly used in the literature.

- Figure 1: there is a typo in the caption: ‘Cape Horn Current’

Thank you, corrected

- Figure 2f: why is not a linear fit used?

During interglacials, the CHC and the DP throughflow reach a plateau at a maximum of 120% of interglacial values. In contrast, central South Pacific records exceed 160% of Holocene values during

MIS 11. This indicates that Drake Passage throughflow strength reaches a maximum during interglacials and the current strength cannot increase, whereas temperatures are not constrained by such a plateau. This difference of behaviour in warmer periods likely explains why a polynomial fit performs better than a linear fit.

- Figure 2: Typo in the caption: Antarctic circumpolar current Strength.

Thank you, corrected

- Figure 4: I think subfigures (c) – (f) can be described a bit more in the caption.

- Figure S5: Is it possible to include the locations of the cores in Figure S6i?

- Figure S6: subfigure (i) is not labelled.

- Figure S6: The fronts in subfigure (i) are not mentioned in the caption.

We modified figures S5 and S6, both including now a labelled map displaying the locations of the cores and the major oceanic fronts.

- Figure S8: In the caption it is not explained what the purple, blue and orange markers represent.

Thank you, corrected

- Figure S8 and S11: the use of the colors for the markers is confusing because they have the same color as the CHC/ACC, SST, and planktic d18O lines. I suggest using different colors for the markers.

We hope the new colors of the markers will be less confusing.

- Figures 4, 5, 6 and S8: the same color (or at least very similar) is used for the planktic d18O record and the dCO₂/dt records. It might be nice to use a different color for the dCO₂/dt records for clarity.

We agree and hope that the new color better distinguish the dCO₂/dt records

References:

- Strub, P. T., James, C., Montecino, V., Rutllant, J. A., and Blanco, J. L.: Ocean circulation along the southern Chile transition region (38–46S): Mean, seasonal and interannual variability, with a focus on 2014–2016, *Prog. Oceanogr.*, 172, 159–198, <https://doi.org/10.1016/j.pocean.2019.01.004>, 2019.

(comments by the reviewer are printed in black, and our point-by-point response in blue)

Reviewer Reports on the First Revision

Reviewer #1 (Remarks to the Author):

Review of “Persistent interhemispheric link of the Antarctic Circumpolar Current millennial-scale oscillations over the last 800 kyr” from Rigalleau et al.

Rigalleau et al. provide new ACC proxy data with an unprecedentedly high resolution to demonstrate a relationship between the ACC and millennial-scale northern hemispheric variability. The manuscript was nice to review: it reads very well, and figures are fancy and pleasant to read as well.

Unfortunately, as I have very limited knowledge in sedimentology, bio/geo chemistry, and quaternary climate, I’m afraid I had to ask the editor to replace me with another reviewer or to add a new one for the next review. I’ve tried to focus my review on statistical analyses provided by the authors, and I have not found any compromising content for publication in this respect. This is the reason why I only have a list of minor comments the authors must consider, see below. Thanks again for the nice and interesting read.

We thank Referee #1 for the very positive assessment of our paper.

Minor comments:

Referencing in general: Blank spaces before superscript referencing numbers must be deleted throughout the text in order to fit the journal format.

Thank you, we are sorry for this mistake and have deleted the blank spaces in the revised manuscript.

L. 26: I suggest using a less personal phrasing for “we still lack [...]”, such as “there is still a lack of a clear mechanistic understanding of these variations”, or “a deeper understanding of these variations is still lacking”.

Thank you, we agree and rephrased this sentence accordingly.

L. 55: “[...] is the largest current system on Earth.” – Such statement needs a reference.

L. 55: “It is driven by atmospheric forcing” – reference needed.

L. 56-57: “bathymetry and ocean density gradients originating from surface” – reference needed.

This general background information regarding the Antarctic Circumpolar Current refers to the review paper by Rintoul 2018 (Nature). We clarified this point in the revised paper.

L. 65-67: “Various model studies” contrasts with the presence of a single reference.

We agree to this being misleading and deleted “various”. We only cite one paper as an example of a benchmark paleomodelling study that for the first time showed the influence of the Drake Passage throughflow and the Agulhas Leakage on the Atlantic Meridional Overturning circulation. We are aware that there are other, more recent modelling studies, but prefer to refer to Knorr & Lohmann, 2003 (Nature) in the introduction of the paper.

L. 65-67: It would be worth specifying the time scales at which this influences is exerted in models. In this line, what kind of models the authors refer to? EMICs/GCMs/others?

We agree that a more detailed discussion would be warranted but our paper focuses on the proxy records. There are numerous model studies discussing interhemispheric climate pattern during the last glacial with regard to the bipolar seesaw concept (e.g. Stocker and Johnson 2003, Paleoceanography).

Figure. 1: There is some information missing in the figure caption:- Indicate that Altimetry-derived ACC fronts are represented with dashed lines. - Indicate in the caption what yellow points are. - Indicated over which period the mean annual sea surface temperatures were calculated.

We agree and added the missing information to the figure caption following reviewers 1 and 3.

L. 112: Replace “an previously” by “a previously”.

Thank you, corrected.

L. 161-162: “the millennial scale variability in ACC strength persist”, there is missing “s” at the end of “persists”

Thank you, corrected.

Figure 2: Orbitally induced spectral peaks are clear for the provided time-series. Could the authors try providing an additional analysis evidencing the millennial-scale frequency they refer to throughout the paper? So far it is only a visual assessment. A wavelet analysis could be able to provide such information.

We agree with Reviewer #1 and added a wavelet analysis (as a new Supplementary figure 10) The wavelet analysis clearly reveals the three major orbital frequencies (precession, obliquity, and the ~100-kyr eccentricity cycle) over the past ~800 kyr in both SST and ACC strength records. Superimposed on orbital variability, we observe substantial spectral power in the millennial band (i.e., 1-10 kyr). In the wavelet spectrum, these changes are evident throughout the record.

L. 180-183: How did these threshold-based events classification was made exactly? From the text it reads that thresholds were somewhat determined arbitrarily (1.6°C and ~10cm/s), where the “~” reads quite surprising as one would expect a threshold to be fixed. I can understand it is difficult to determine a clear approach to determining classification thresholds here, but the authors should consider being more transparent by precisely explaining how these choices were made. In the case thresholds were determined arbitrarily, an obvious subsequent question I would have is: are the following results sensitive to this choice then? Could the authors demonstrate these thresholds were not cherry-picked?

The threshold chosen is empirical, as explained in the methods. We chose it in order to identify almost all of the canonical D-O events during the last glacial cycle, similar to the approach of Barker et al., 2011 (Science). The procedure of separating major events from all events is explained in Supplementary figure 8. By sorting each event for the respective amplitude, we visually observe for each record two distributions. The velocity is derived from the visual separation (28% ACC strength) between the distributions and the subsequent translation to velocity. We now better clarified our approach in supplementary figure 8g and corrected the main text (methods) accordingly (lines 725-727).

L. 194: I think “1 ka” should be “1 kyr” here? I may be wrong here.

Thank you, this is correct, it should be kyr. We have corrected the text accordingly.

L. Replace “This imply” by “this implies”.

Thank you, corrected.

L. 205-207: A reference is needed to support this sentence. “This imply that the frequency of events is rather constant, even though the magnitude varies with background climate.”

Thank you for this comment. This sentence does not refer to a specific reference but is based on the assessment of the new records from Site U1542. Following a major Reviewer #3 we decided to modify our approach and now refer to complete glacial cycles for our interhemispheric comparisons (lines 366-367 and 391-404, and updated Fig. 4).

L. 209-210, Figure. 3E,F: It is difficult to assume the linear relationship to be significant here. The R^2 most likely is very low (given the distance of points with the regression line), and the sample size is pretty small. I would mention this in the present interpretation and add some conditionality in this statement.

Figure 3: Significance tests for regression line slopes for panels C-F are missing – please provide p-values and sample sizes. Use Student t-tests.

Interpretation of Figure 3: Under the assumption that millennial-scale events occur on a regular basis, one would inevitably obtain more events for longer time windows (thus for longer MIS). In other words, what do we learn exactly from panels C and D?

We agree with Reviewer #1 that these plots do not provide new insights. We therefore removed the panels Fig. 3c to f. However, we keep the two panels with the overall distribution of the events in the Supplementary figure 9.

Figure 4: To my knowledge, figures and their respective caption must be self-consistent in Nat. Comm. This means that acronyms and abbreviations such as “D-O”, “AMOC”, or “EDML” must be given in their full version the first time they are mentioned in the caption.

We apologize for overlooking this point. We corrected the figure captions accordingly.

L. 387: Missing “-“ symbols at “state of the-art”.

Thank you, this has been corrected.

L. 423: The acronym for North Atlantic Deep Water (NADW) has not been defined earlier.

Thank you, we use the full name as it is only used once.

L. 442: “SWW wind-driven” reads redundant. Should maybe be SWW-driven?

We agree and deleted “wind”.

Figure 6: See my above comment of Fig. 4.

We thank the reviewer for the careful revisions and corrected the respective text parts accordingly.

Reviewer #2 (Remarks to the Author):

(comments by the reviewer are printed in black, our point-by-point response in blue)

The manuscript “Persistent interhemispheric link of the Antarctic Circumpolar Current millennial-scale oscillations over the last 800 kyr” by Rigalleau et al., provides new high-resolution current strength and sea surface temperatures (SST) records, spanning the past 790,000 years. The sedimentary archive IODP U1542 is located within the Cap Horn Current (CHC) – northern branch of the ACC. The authors report that their records show persisting millennial-scale climate variability throughout all Late Pleistocene, with stronger ACC flow coinciding with warmer SST and Antarctic warm intervals. Additionally, the Southern Hemisphere changes seem to coincide with North Atlantic millennial-scale fluctuations suggesting a close link between ACC flow and the Atlantic thermohaline circulation. Lastly the observed changes in IODP U1542 also seem to follow atmospheric CO₂ changes and for that, the authors propose a mechanistic link through the Southern Ocean carbon cycle.

Variations in the strength and position of the ACC play an important role in global ocean circulation, ocean carbon cycle and thus Earth’s climate. Understanding the mechanistic links between these processes is therefore of major importance. I would first like to point that the total number of samples analyzed for this study, especially the alkenone-based SST is impressive. Overall, the data are high quality and convincing. I also appreciate the way the data are presented - figures illustrate clearly the datasets. These high-resolution records are of great interest and importance to the scientific community and can be used for interhemispheric comparisons (as authors did) providing important insights into the millennial-scale variations of the ACC and their links to Earth’s climate.

We thank Referee #2 for the positive assessment of our paper.

Said that, I unfortunately think that the manuscript, in its current form, does not provide new insights in terms of processes/mechanistic links and the understanding of past ACC variations, compared to what has already been reported by several studies in the same area (e.g., Lamy et al., 2015; Toyos et al., 2020; Wu et al., 2021). Moreover, the new data does not always strongly support the main conclusions. I think that there is more information that can be extracted from these datasets and help in understanding the dynamics of the ACC and their interactions with climate and CO₂ variability during the past 800,000 years. Below I explain in detail my main concerns.

We address the reviewer’s concerns further down in the rebuttal letter.

SST and ACC strength reconstruction from current-controlled deposits.

As IODP U1542 is located close to the Chile continental margin, in the upper slope and thus not in a deep ocean setting, it is expected that sedimentation is largely affected by down-slope processes (turbidites, debris flows, etc), mixed with along-slope processes (bottom currents). Sedimentation rates are five times higher during glacial times compared to interglacial periods. This indicates a major sediment input from land. That could have a great influence in the grain size variations. Using the mean sortable silt grain-size (SS; geometric mean of the 10 – 63 μm silt fraction) versus the sortable silt percentage (SS%, 10–63 μm fraction relative to the <63 μm fraction) can help to investigate whether the sediments are subject to significant bottom current sorting. However, this is not the only test that can be used and given the complex depositional setting, I would recommend to provide more evidence

on whether bottom currents play the main control on sediment transport in the upper slope of the Chile continental margin. In line 139: authors argue for only limited ice-rafted debris impact in their sediments. This can be easily checked using their data by plotting the fine sand (wt %) and the coarser “possibly IRD fraction” (>150 µm; wt %). Additionally, a more detailed investigation on the grain-size and XRF-core scanning datasets can really help. A Principal Component Analysis on the XRF scanning and grain-size data can provide robust insights into the main processes governing the variations in your sedimentary archive. Alternatively, it would be helpful to plot together the SS, SS%, fine sand (wt %), (>150 µm; wt %) and terrigenous elements from your XRF scanning dataset, e.g., Ti, Al, Zr, Rb. This will illustrate potential similar patterns and help to robustly argue for the main processes controlling sedimentation at Site U1542.

We thank the reviewer for bringing this to our attention and completely agree to discuss in more detail sedimentological and geochemical data supporting primarily current controlled processes at Site U1542. We added Supplementary figure 4 highlighting the correlation between different sediment parameters and changed the main text accordingly. These new results clearly support the following discussion and consolidate the whole study (lines 145-160).

Figure 1. Scatter plot of grain size variations and terrigenous elements from XRF scanning datasets and Principal Component Analysis (PCA) performed at Site U1542.

We also followed the advice of Reviewer #2 and performed a Principal Component Analysis (PCA) as suggested (see figure above). The lower part of the figure shows the loadings for the PCA for the Site U1542 XRF and grain size data. Each loading is coloured by its typically associated origin according to Rothwell & Croudace (2015; Developments in Paleoenvironmental Research). Stronger loadings indicate that the element is more affiliated with that component. The PCA data shows that the sediments are primarily of turbiditic/terrestrial sedimentological deposits, subsequently current sorted.

We found that generally PC analyses do not give new insights into the sedimentation process at Site U1542, required for our study. We think that such more detailed sedimentological analyses and discussions are beyond the scope of this manuscript.

Sensitivity of the Southeast Pacific to orbital and millennial-scale climate oscillations.

Orbital and millennial-scale variations of the ACC strength in the Drake Passage region, as well as in the Chilean Margin and have been previously reported covering different time periods: over the 65 ka in the CHC current (Lamy et al., 2015), over the past 140 ka (Wu et al., 2021), over the past 1.3 Ma (Toyos et al., 2020). Thanks to these works, we have a good view on the variations of the ACC orbital and millennial-scale variations of the ACC/CHC strength in the Drake Passage and Southern Chilean Margin. The aforementioned studies provide similar conclusions and mechanistic processes with the current work, describing stronger ACC in interglacial intervals vs weaker ACC in glacial intervals (Lamy et al., 2015; Toyos et al., 2020; Wu et al., 2021), linking them with low AMOC strength (Wu et al., 2021). I believe that with the new high-resolution records presented in this study the authors have the opportunity and advantage to explore in depth these mechanisms. So, I highly recommend to go deeper in their interpretations.

For example, in Lines 232-234: “At orbital timescales over the past 790 kyr, our reconstructed SST variations at the southern Chilean margin largely follow the atmospheric temperature record of the Antarctic ice core EDC and the open Southeast Pacific SST recorded at PS75/034-2 (Fig. 2C-E).” Looking the figure carefully, I would say that this is not really the case. PS75/034-2 follows largely the EPICA Dome C record, but Site U1542 is not always following the same pattern (e.g., MIS 7-6, MIS 13-12). This is a very interesting observation from these new high-resolution data that I believe needs to be discussed in more detail. I agree with the interpretation of the authors for the higher temperatures recorded at Site U1542 compared to PS75/034-2, but there is more to discuss and extract from the comparison between these records.

We agree that a more detailed comparison between our high-resolution U1542 SST record and the more offshore-located, lower-resolution SST record from core PS75/34-2 would improve our interpretations. The original publication of the PS75/34-2 SST record favours the usage of the Uk-based SST data including the C_{37:4} alkenones. Ho et al 2012 (Paleoceanography) argued that the Uk₃₇ based SST record is more similar to the EDC ice-core temperature variations, particularly before the Mid Brunhes Transition (MBT). However, in our study, we applied the Uk₃₇-based SST calculation which is usually used in paleoceanography including the subantarctic South Pacific (Jaeschke et al., 2017). We changed Fig. 2 accordingly.

We also agree that the mentioned studies discuss ACC changes overall in terms of similar climate mechanisms as discussed in our study. This strengthens our interpretations as our new data are consistent with the previous records from the Drake Passage and at the southern Chilean margin. However, our study provides two major advances: (1) We extend the previously shown millennial-scale variations of northern ACC strength changes at our site (Lamy et al., 2015) and in the central Drake Passage (Wu et al., 2021) to a substantially longer time period. This allows us to discuss millennial-scale climate pattern over the past ~800 kyr, i.e. the entire time period of the EDC ice-core. (2) For the first time, at Site U1542 we provide a reconstruction resolving Southern Hemisphere millennial-scale SST changes over this long time period. Both ACC strength and SST records can thus be thoroughly compared to Northern Hemisphere records. In our manuscript, we compare our Site U1542 millennial-scale pattern to North Atlantic temperature, IRD, and AMOC records allowing us to discuss interhemispheric climate linkages with unprecedented length.

The Uk₃₇-based records from our Site U1542 and PS75/34-2 show very similar long-term (orbital-scale) variability. Both records reveal subdued glacial/interglacial amplitudes before the MBT. SSTs at U1542 are overall warmer by 2-3°C consistent with the impact of the relatively warmer Cape Horn Current at the site U1542 SST record. Using the Uk₃₇-based SST record, the differences at orbital scale are smaller and potentially due to age model uncertainties (e.g., MIS13 to 15). In addition, SST changes at precessional time scales appear more prominent at Site U1542 (e.g. MIS7). We observe large amplitude sub-orbital-scale variability in our record compared to PS75/34-2. In addition to the much higher temporal resolution, the sensitivity of the Cape Horn Current to sub-orbital climate change could have been higher. Furthermore, the glacial/interglacial amplitudes are lower in PS75/34-2.

We refined the discussion on the sensitivity of the eastern South Pacific chapter accordingly. We further added a new supplementary showing additional key SST records from the Southern Hemisphere (Lines 192-212 and Supplementary figure 6).

The authors suggest that the decoupling between Southern Ocean and Antarctica southern latitudes (Lines 253-259), resulted in an increase of the meridional thermal gradient, increased moisture to the poles and continue ice growth, but as they mention later this feature is probably not a prerequisite for the ice sheet growth. This part needs further explanation.

We agree with the reviewer that this would need further explanation. However, we think that a more thorough discussion on whether the difference of trend would probably be beyond the scope of our paper and therefore removed this sentence.

Lines 262-264: “Enhanced flow speeds coincide with warmer periods, while reduced flow speeds coincide with colder periods, implying a close coupling between near-bottom current velocity and SST (Fig. S4)”. The correlation the authors show in Fig.S4 did not suggest a close coupling ($R^2= 0.5$). I would say that this close coupling they mention in the text is not actually the case. There are indeed times, where peaks of ACC strength coincide with high SST and warm intervals in Antarctic climate, but there are many times that this is not the case. For the example, the last 100 ka, ACC show a long-term increasing trend whereas SST data from U1542 and more clearly from MD921-20 show a long-term cooling trend coinciding with Antarctic ice core EDC. This is observed in other time intervals as well. This is a very important observation that needs to be discussed.

We agree that at first glance the correlation between ACC strength and SST is not very high. However, the R^2 of 0.5 is based on the 1125 SST data points correlated with the much higher resolved XRF data ($n=7322$) (Fig. 2f). As the XRF data represent only a limited area at the surface of the sediment core and the SST are derived from 1 cm slices, some error is introduced. Nevertheless, the reviewer is right in pointing to a closer look at the differences between ACC strength and SST (e.g., the long-term trends in MIS2-5). We think that in particular, the trends might indicate a partly different behavior of the bottom currents within subantarctic ACC/CHC and the SST responding in parts to different atmospheric (e.g. Southern Westerlies) and oceanic forcings (lines 218-243).

In lines 268-288, the authors propose that latitudinal migrations of the Southern Ocean frontal system during glacial / interglacial times as a primary driver of changes recorded at Site U1542 (located north of the SAF), similar to the proposed mechanism in PS97/85 (located in the polar front). Northward migration of the Southern Ocean frontal system during glacial times should bring SAF closer to Site

U1524, compared to interglacial times. Given that stronger ACC flow appears within /near the SAF. Why don't we see higher ACC flow speeds during glacial times at Site U1542 and vice versa for the interglacial times? In other words, do the authors expect to see the same trends in ACC strength variations in all records: PS97/085-3, PS97/093-2, PS75/034-2 and Site U1542, unrelated to their locations within the ACC frontal system? Why?

We agree with reviewer #2 that latitudinal migrations of the Southern Ocean frontal system might influence the ACC strength at the mentioned sites. However, all sites with different locations regarding the modern fronts register the same glacial/interglacial pattern with reduced ACC strength during glacials and stronger ACC during interglacials. This pattern is consistent with ACC reconstructions in the Central and Southwestern Pacific Southern Ocean (Lamy et al., 2024; Nature) showing the same glacial interglacial reduction/strengthening of the ACC from the Subantarctic Zone, across the Polar Frontal Zone and the even south of the Polar Front, though glacial/amplitudes decrease southwards. Therefore, the observed ACC strength variations are consistent and indicate that frontal shifts are not the main factor in determining ACC strength in our study area and the South Pacific in general. Moreover, even with a substantial northward shift of the frontal system, the Subantarctic Front most likely would have remained south of U1542.

I would recommend to include a figure showing ACC strength and SST variations from all sites discussed in the manuscript (U1542, PS75/034-2, PS97/093-2 and PS97/085-3). This figure will help to clearly identify changes across the whole ACC frontal system in the Drake Passage region.

We agree to the reviewer and now provide two new supplementary figures showing ACC strength and SST variations at the relevant sites (Supplementary figure 5 (for ACC strength) and supplementary figure 6 for SST changes). There is unfortunately no SST record for PS97/093-2 and PS97/085-3 available and an ACC strength record for site PS75/34 has not been published (we are showing the unpublished record below).

Supplementary figure 5. ACC strength changes over the past 800 kyr in the subantarctic Pacific. (A) Reconstructed relative ACC strength variations (compared with Holocene mean values) in the Drake Passage (unpublished data), (B) (Toyos et al. 2020, P&P; Lamy et al. 2024, Nature), (C) on high resolution near the Polar Front (Wu et al. 2021, Nat.comm.), (D) on the Chilean margin (this study) and (E) in the central subantarctic Pacific (Lamy et al. 2024, Nature). Latitude, location related to oceanic fronts and water depth are indicated by identical color to the record.

A persistent interhemispheric teleconnection

I generally found this section difficult to follow as the main message seems to be hidden in the comparison between the different dataset.

Lines 302-303 “Millennial-scale climate events at Site U1542 are found to be contemporaneous with AIM events” A closer look to the figure does not show this. There are peaks in ACC strength that does not coincide with AIM events. Moreover, in lines 311-314, authors report that intervals characterized by stronger ACC coincide with the weakening of the AMOC (during Heinrich Stadials). I would argue that this is not what the figure 4 shows. I suggest that the authors modify the text, so it will describe

more precise the comparison between the different datasets and clearly state the main message of this section.

Following a similar concern of Reviewer #3, we changed our approach now displayed in Fig. 4. Our interhemispheric millennial-scale analyses are now based on glacial cycles (and not fixed 10-kyr windows). We assume that glacial terminations provide the best time marker in the records and minimize the impact of age-model uncertainties. We modify the text regarding the figure displaying the last glacial period (new Fig. 3), which provides an overview of the bipolar seesaw concept, with warming and strengthening in the SH associated with cooling in the NH. (lines 246-247, 356-358).

Role of ACC in CO₂ changes

The authors associate a total 31 major ACC strengthening events with CO₂ rise, 15 of them present a CO₂ rise before an ACC strengthening and 15 events present a CO₂ rise after an ACC strengthening, suggesting a link between ACC strengthening events with CO₂ rise. They propose that strengthening and southward displacement of the SWW, enhancing ACC strength, decreasing Southern Ocean stratification promoting ventilation, resulting in increasing atmospheric CO₂ levels. However, what are the mechanistic processes that occur during the events where CO₂ rise before an ACC strengthening? Are they the same?

This is an important question. Unfortunately, it is difficult to constrain the exact timing of millennial-scale ACC maxima and CO₂ rise in more detail. We do not have a strictly independent age model on millennial-scales allowing to discuss potential leads and lags of CO₂ variations to ACC strength changes. Besides age model uncertainties, there are also non-Southern Ocean related processes affecting atmospheric CO₂. Therefore, we removed the sentence associated and only focus on the associated events in less than 7 kyr, regardless of the exact timing.

Line 405-407: “Moreover, the relative changes in flow strength appear to be positively correlated to the amplitude of CO₂ variations (Fig. S13)”. However, the correlation between ACC strength and CO₂ rise shown in Fig. S13 is rather weak ($R^2=0.3$).

The relatively weak correlation can mainly be explained by one single millennial event at 263.2 ka, characterized by an ACC strengthening of 36.5% and a corresponding CO₂ rise of 31.9 ppmv. The amplitude of this event is high and nearly in the order of changes across glacial terminations, with higher amplitudes in CO₂ compared to the ACC strengthening. By removing this event, r^2 reaches 0.5, which supports the correlation between ACC strength and CO₂ rise.

As mentioned in a previous comment, a figure showing ACC strength, SST variations and productivity proxies XRF-Ca from all sites discussed in the manuscript (U1542, PS75/034-2, PS97/093-2 and PS97/085-3) can help to unravel changes across the whole ACC frontal system (e.g., latitudinal migrations, bottom strength and productivity changes) that can be then linked with CO₂ changes.

We agree to the reviewer and now provide two new supplementary figures (Supplementary figure 5 and 6) showing ACC strength and SST variations at the relevant sites (see also additional figure for the reviewer including the unpublished ACC strength record from PS75/34. The eastern South Pacific and Drake Passage sites have not all been studied for paleoproductivity changes (only PS97/93; Toyos et

al., 2022, CLP). The PS97/93 records suggest a strong influence of micro-nutrient input from the Patagonian ice sheet during glacials. Though with the same sign, these changes are less directly related to frontal movements and shifts of the opal belt as documented in the central and southwestern Pacific Southern Ocean (Lamy et al., 2024, Nature). We agree that these changes are overall linked to CO₂ variations but a more detailed discussion of the productivity records is complex and beyond the scope of this manuscript.

Minor comments

Fig. 1: Please add the location of MD07-3128

Considering the scale of this map, it is unfortunately not possible to graphically distinguish the locations of core MD07-3128 and Site U1542 as they are only two nautical miles away from each other. We added to the figure caption that MD07-3128 is at the same location as U1542.

Fig. 6: Please add AIM events

We added “AIM” events in figure 6.

Line 311: Wu et al 2021, change it to reference number

Thank you, corrected.

Lines 326-329: This part can be moved to methods

We removed this sentence from the main text and moved it to the methods.

Lines 410-411: Please cite figure 6 and add AIM events in the figure.

We cite the respective figure and added “AIM” numbers.

I am sorry for not rating this paper higher. I strongly appreciate the efforts of the authors to generate such high-resolution records, which are of great value.

We thank Reviewer #2 for this final statement and hope that our revisions address the reviewer’s concerns and improve our manuscript accordingly.

Reviewer #3 (Remarks to the Author):

(comments by the reviewer are printed in black, our point-by-point response in blue)

In this paper the authors analyze a new core from underneath the Cape Horn Current (CHC) system that goes back 790 kyr. They find persistent millennial scaled variability in the Antarctic Circumpolar Current (ACC) strength and Sea Surface Temperatures, and relate this to millennial variability in the North Atlantic suggesting that this link is present over the past 790 kyr. Lastly, they also suggest a role of this variability in the carbon cycle in the Southern Ocean with global consequences. I think the paper is in principle very interesting and I applaud the authors for the amount of work done. I also believe that getting a better understanding of the influence of Southern Ocean dynamics on the ocean, the carbon cycle and the entire climate system is very relevant for future climate change since it is a large source of uncertainty.

Thank you, we appreciate this statement.

However, I have some major concerns about some of the aspects of this study. To address these concerns it might be necessary to make significant revisions that can be quite a bit of work and might also change the scope of this paper. It might therefore be better to reject/withdraw the paper and submit it to a different journal or resubmit here. I leave that decision to the editor and/or authors

My first main concern is that it is claimed that the CHC is part of the ACC and it is assumed that CHC dynamics captured in the core provide an accurate proxy for ACC dynamics. I am not convinced this assumption is valid. Figure 1 actually shows that the core used in this study is north of the Northern Boundary and therefore not in the ACC. Also two papers cited (Chaigneau and Pizarro, 2005; Park et al., 2019) state/imply that the CHC is not part of the ACC. I think it matters for the significance of this paper and the overall conclusions of this paper whether this assumption is valid, which in my opinion it is not valid. I suggest that the authors provide a sound argumentation why this assumption is valid.

We thank Reviewer #3 for pointing us to discuss the oceanographic setting of IODP Site 1542. The northern Boundary (NB) of the ACC frontal system is defined by altimetry and is strongly related to bathymetry features (Park et al. 2019). Therefore, the location of the NB in the Southeast Pacific close to the continent is likely to be affected by the Chilean margin, which would explain why the coastal CHC is located north of the NB and, a priori, why the CHC and Site U1542 do not belong to the ACC. Therefore, our interpretation that the CHC forms part of the ACC is primarily based on previous oceanographic studies in the region (Chaigneau and Pizarro 2005), Supplementary figure 1 (modified from Chaigneau and Pizarro 2005) shows examples of surface buoy trajectories over the bathymetry. When buoys reach the coastal region at $\sim 44^{\circ}\text{S}$, the “black drifter” is advected southward into the coastal CHC and returns to the ACC two months later (Chaigneau and Pizarro 2005; paragraph 6). Therefore, this study implies that the CHC is fully associated with the ACC and can be characterized as a northern, subantarctic branch of the ACC. Furthermore, a recent study from Zheng et al., 2023 focuses on the CHC. Based on satellite altimetry (1993–2021, 400 m deep), as well as 2 ocean models, one free-running and one data-assimilating, this study clearly indicates/assimilates the CHC as a component of the ACC. This study further describes the CHC as “a component that acts as an inter-basin conduit, drawing relatively cold and fresh Subantarctic Surface Water from the South Pacific, and injecting it into the South Atlantic.”. This recent study was not cited in the first version of the manuscript, and we include it now to better support this assumption. Together, these two modern studies of the CHC, one

from observation and one from satellite observation and model data strongly support that the CHC can be considered a component of the overall ACC system. The incorporation of the CHC into the ACC has been the baseline of several surface sediment studies (e.g., Vollmar et al. 2022; Wu et al. 2019). The only paleo record of the CHC (Lamy et al. 2015) stated that the CHC is advected through the DP as well during glacial period. The similar glacial reduction observed at the CHC (MD07-3128) and in the northern DP (MR0806) further implies that the CHC is connected to the subantarctic area of the DP. Therefore, the study of the CHC appears to be relevant in quantifying the northern ACC strength changes in the Drake Passage. This extended explanation is part of the Supplementary Fig. 1 caption.

My second main concern is that it is difficult to check the conclusions with the figures provided and how I analyze some figures I would conclude that there does not seem to be a clear relation between some of the variables. First of all, the figures with time axes often have a length of 800 kyr while variability at the 1 kyr scale is analyzed. I suggest besides providing the full length series, also provide zoom in of e.g. ~50 kyr to support your conclusions. Secondly, for me Figure S6 shows that there is no(!) relation between events in the Northern Hemisphere and the Southern Hemisphere because the graphs look very different in my opinion. In the main text Figure S6 is used to argue that there is a clear relation. I suggest that when considering the relations between the different variables either a well discussed statistical test is used and/or a sound physics based mechanism is provided.

We thank the reviewer and improved the respective calculations and figures. In the revised Supplementary figure 6, we changed the time window to compare different records. Due to age model uncertainties, the 10 kyr may be too arbitrary for the interhemispheric comparison, and one event may be put in one box or another depending of both the age model and box definition. Therefore, we changed the windows to full glacial periods which appears to be more realistic regarding age model uncertainties, as terminations represent robust tie points for tuning. In this way, we are confident which cycle belongs to each millennial-scale event. Furthermore, given the length of the comparison, eight glacial cycles seem to be a reasonable number of glacial cycles for interhemispheric comparisons. We used the glacial terminations given by the Lisiecki & Raymo (LR04) marine isotope stack. Fig.4 replaces the 10kyr windows and we changed the text accordingly. Following the reviewer, we now also provide individual figures for each glacial cycle (Supplementary figure 12a-f). With these revisions, we provide better statistical correlations and graphical support for our conclusions. The main text has been changed accordingly (lines 391-404).

My third main concern is closely related to what I describe above. I think there is a lot of variations in timing and amplitude in the events of the different variables and this makes me doubt whether the events are necessarily connected. Events in the variables can be leading, lagging, coincide or not coincide at all without, in my opinion, a good explanation why these events are connected even though they are so different in these aspects. Also the time window used to detect events in multiple variables is a bit large considering that typical ocean timescales are at 1000 – 2000 year. I suggest providing a better argumentation (e.g. statistical tests and/or physics based mechanisms) as to why and how events between different variables are connected. Improving figures displaying the events can already help a lot here.

We agree with the reviewer that the age uncertainties of our records over the past ~790 kyr is too large for a detailed interhemispheric comparison of synchronicity or lead/lag relationships of individual

millennial-scale events. Such comparison is for example possible, albeit difficult, for methane-synchronized Greenland and Antarctic ice-core records or ideally for radiocarbon-dated marine records over the past ~40 kyr. Such records formed the base for the bipolar seesaw concept of antiphased millennial-scale temperature changes in both hemispheres (e.g., Stocker & Johnson, 1998, *Paleoceanography*). This conceptual model based on the findings from the paleoclimate proxy records and paleoclimate modelling, is the “physics-based mechanism” for extending these concepts to a longer time scale, i.e. eight glacial/interglacial cycles. There have been several studies over the last glacial cycle showing that the southern (Antarctic) timing of millennial-scale changes extended to the northern ACC both in the South Pacific and South Atlantic (e.g. Lamy et al., 2004, *Science*; Canipuan et al., 2011, *Paleoceanography*; Lamy et al., 2015, *PNAS*; Barker & Diz, 2014, *Paleoceanography*; Barker et al 2009, *Nature*; and others).

Therefore, we assume that the bipolar pattern extends to the ACC, at least in the Southeast Pacific/Drake Passage region. However, we do not expect that all individual events correlate in detail and we therefore compare rather their numbers and amplitudes in each glacial to the North Atlantic records. To our knowledge, this is the first study of millennial-scale proxy records in the southern hemisphere extending back to ~790 kyr.

Next, I have also some more specific comments/suggestions for the authors:

General:

1. In the text and in the title the word ‘oscillation’ is used frequently. In my opinion, an oscillation has a more or less stable frequency and I don’t think that is the case in your records. I suggest using for example variability instead of oscillation in the title and throughout the text.

We agree, and changed the text and the title accordingly.

2. I don’t see anything about measurement uncertainties in the text and in the figures. Have I missed this? This should be included in my opinion.

We thank Reviewer #3 for this concern. The analytical uncertainties regarding alkenones-derived SST are given in the material and methods (0.23°C), and has been subsequently taken into consideration in the measurement of the amplitude of the events. The reproducibility of the procedure was evaluated using a homogeneous sediment standard, extracted with every batch of samples. The relative analytical errors were below 0.5°C in SST estimates (we added this text to the methods (Lines 637-640). The uncertainties of the ACC strength are relatively small as the analytical error of the XRF Core Scanner for both Zr and Rb is small. We have not measured these elements wet-chemically but the signal-to-noise ratio for both elements has been previously shown to be excellent as those “heavy elements are precisely detected by X-ray fluorescence, even at low concentrations (Wu et al. 2020, G-cubed). The uncertainty of the grain-size measurements is also low and does not impact the final result, as it serves only for calibration, the XRF $\ln(\text{Zr}/\text{Rb})$ being used for subsequent analyses. Furthermore, the linear equation used for the calculation of sortable silt is sufficiently defined. we a text into the methods (Lines 685-689

Text:

1. Also based on previous comments, I think the title does not accurately capture the content of the paper in my opinion. I think that oscillations could be replaced with variability, the ACC with CHC,

and 800,000 with 790,000 since that is the length of your core. Also, as mentioned previously I have my doubts about the interhemispheric link based on this study.

Thank you. We have revised the title accordingly.

2. Line 55-57: I think this sentence can be clarified a bit. It's not clear to me if the atmospheric forcing is just the Southern Westerly Winds or also temperature and freshwater forcing. It is also not clear whether the salinity changes also relate to the ocean density gradients because of the comma before 'and' at the end of the sentence.

Thank you for pointing out the mistaken use of the comma. We have revised the text accordingly (lines 55 to 57).

3. Line 109 – 111: Can you refer to a figure here?

We now refer to figure 2c.

4. Line 111 – 113: I'm not sure if I understand this. Is there a bias in the measurements? Is this bias compensated for? Why not if it's not? I suggest clarifying this phrase.

No, we do not think that there is an analytical bias in the measurements. The late Holocene value is above the modern annual mean, mainly because alkenone SSTs in the region are seasonally biased, i.e. towards summer temperatures (e.g. Hagemann et al., 2023, CLP).

5. Line 197: 'amplified', I think you mean here that events are more likely to occur in these ranges, so I suggest changing the text to reflect this.

We apologize for the erroneous use of "amplified" and changed the sentence accordingly (lines 314-317)

6. Line 199: you make a reference to ice sheet volume in Fig. 2, but there is no ice sheet volume in Fig. 2 which is a bit confusing. I suggest being more accurate in this statement.

We thank the reviewer and reworded the text accordingly.

7. Lines 279 – 288: I don't understand where to look for the amplification in Fig. 4. How does this amplification look in the data? I also don't understand how this supports the idea of SWW shifts as a primary driver. How I would reason (tell me if I'm correct in this): when SWW shifts southward, the water flowing into the CHC becomes colder, when the SWW shift northward, the water becomes warmer because of the latitude the source waters come from. But I don't understand now where to look in the data for the SST response in U1542. I suggest clarifying the text and specifically the link with Fig. 4.

We thank the reviewer for bringing up this important issue. We are referring mainly to the high amplitudes of SST changes along Chilean margin SST records, both on millennial-scale and orbital time-scales. This has been shown in a number of previous studies. Compared to e.g. Antarctic ice-core

records, SST changes across the last glacial termination and millennial-scale events were relatively sharp which has been suggested to indicate a stronger atmospheric linkage to the Northern Hemisphere involving shifts of the Intertropical Convergence Zone and the Southern Westerlies superimposed by SST changes through ocean circulation (including the “see-saw”) (e.g., Anderson and Carr, 2010; Science, Lamy et al., 2007; EPSL). In this scenario, the westerlies shifts southward during NH coolings, resulting in enhanced advection of relatively warm Cape Horn Current water. The contrary occurs during SH coolings. We think these concepts explain the high amplitude in our SST record reasonably.

8. Line 382 – 388: Here you relate your results to results from ESMs. I do not agree that these results can be compared. You look at completely different timescales. The timescales assessed here are 1 – 10 ka, while the ESM studies are on centennial time scales. The processes are therefore, in my opinion, not translatable. I do think that you can mention it in the text, but I suggest being more nuanced here.

We find the comparison with the impacts of anthropogenic climate change from observations and ESMs reasonable because the magnitude of change, in terms of temperature rise, for example, is comparable despite the different time scales. We find that scepticism over whether the same processes are occurring on such different timescales is reasonable, but we do not think that this precludes it as a discussion point. Rather, it is an interesting opportunity for further research and we find the transdisciplinary comparison relevant for a journal such as *Nature Communication*. However, we recognize that this part can be more nuanced and we have added an acknowledgement of the distinct time scales and highlighted the need for further research at line 454.

Figures:

1. Figure 1: This corresponds to my main concern 1. There are arrows drawn north of the NB line that are called the ACC. Looking at the paper where the data is from, they call the NB line the actual northern boundary of the ACC. I suggest changing the name to SPC as in Fig. S1 or motivating very clearly why you consider this to be the ACC.

Please see our response regarding concern 1

2. Figure 1: No source of the SST data is given, and it is not clear what period the SSTs represent. I suggest adding both in the caption.

We now give the source “observations from all months of all years” for the period 2005-2017, as described by (Locarnini et al. 2018), citation also cited in the figure caption.

3. Figure 3: This figure misses clear labels and units for the axes. It is therefore difficult to interpret this figure. On the left ‘ACC strength’ and ‘SST’ look like labels, but it took me a couple of looks to understand they’re not

This figure has been removed following reviewer’s concerns.

4. Figure S3: Caption does not clearly explain the subplots. In particular, LSR (local sea level rise?) is not mentioned.

LSR is the acronym for Linear sedimentation rate and is now mentioned.

5. Figure S7: I think it would be useful to be able to better compare the timing of the events by plotting the events of the different variables closer to each other. The time between 2 events could for example be shown by changing the marker size.

The figure has been removed and replaced by Supplementary figure 10 (evolutive spectral analysis) which better shows the frequency of millennial-scale variability.

6. Figure S9: Shouldn't n be 103 in the bottom subplot?

Has been corrected (the figure is now Supplementary figure 8).

Reviewer Reports on the Second Revision

Reviewer #2 (Remarks to the Author):

The manuscript titled " Persistent interhemispheric link of the millennial-scale Antarctic Circumpolar Current variability over the last 790 kyr" from Rigalleau et al., presents a high-resolution current strength and sea surface temperature records covering the past 790,000 years from the Chilean Cape Horn Current, off the Chilean coast.

The authors present evidence of persisting millennial-scale climate variability throughout the last eight glacial periods, characterized by stronger current flow and warmer sea surface temperature, coinciding with Antarctic warm intervals. They interpret these changes as being linked to the Antarctic Circumpolar Current (ACC) and suggest a link to North Atlantic, proposing persisting interhemispheric dynamics over the last 790 kyr. Lastly, they associate the variability in current strength with atmospheric CO₂ fluctuations, suggesting a role of this variability in influencing the Southern Ocean carbon cycle.

This is the second time I have reviewed this paper, and overall, the results of the study are interesting, with an impressive amount of work and analyses. The data are of high quality, and the figures clearly illustrate the datasets. This work could make a significant contribution to our understanding of the Southern Ocean climate, ocean and carbon cycle dynamics, and its broader connections to Earth's climate system.

We thank Reviewer #2 for the positive assessment of our paper.

While the manuscript presents valuable insights, I believe it is not yet ready for publication. There are still major concerns that need to be addressed, particularly in clarifying interpretation of the results and improving the flow and readability of the text. I have outlined my concerns and specific recommendations below to help strengthen the manuscript.

My first main concern remains whether the grain-size changes at U1542 can be linked to changes in the ACC strength. This is due to the location of the site which is situated very close to the Chilean coast, in the upper slope and thus its sedimentation is likely primarily driven by downslope processes related to continental sediment inputs from meltwater transport and sea-level changes, with current reworking playing a lesser role. You mention that the expansion of Patagonian Ice Sheet that reached the continental shelf edge at glacial maxima may explain the interhemispheric differences you observe between your records and others (e.g., 364-367 "the interhemispheric disparity in several cycles..."). This raises the question of whether the observed changes in sediment composition, grain-size, are truly driven by variations in bottom currents or rather by downslope processes. If the latter is the case, the grain-size changes may not serve as reliable indicators of changes in current strength.

We agree with the reviewer that downslope processes might play a role in peak glacial periods, as we mentioned in the manuscript. Given the new grain-size analyses (see next point below), we assume that current strength is, however, the major factor for most of the record, as indicated by the overall excellent correlations between the different grain-size parameters (see below). Downslope processes may have some importance during glacials, when finer grain-sizes might be influenced by glacial sediment supply. Therefore, we have extracted all samples with SSFS < 30 μm (primarily representing glacial intervals) and plotted the mean SS versus the weight percentage of SS separately (Supplementary Fig. 4f). The comparably good correlation between mean SS and percentage of SS ($R^2 = 0.59$) suggests an only moderate influence of terrestrial sources due to current reworking, even during glacial periods. These data are consistent with the Patagonian Ice Sheet reaching the outer Magellan Strait only during peak glaciations as shown, for example, with the high sedimentation rates during the last glacial (*c.f.* Supplementary Fig. 3) (e.g. Lamy et al., 2015; Hagemann et al., 2024). At least during most parts of intermediate and interglacial periods, downslope transport of terrigenous sediments was less important, as a large amount of sediment was trapped within the fjords and never reaches the shelf break. During

these intervals, sorting by currents gained more importance and can, in our opinion, be used as current strength indicators.

The moderate correlation ($R^2 = 0.43$) between the mean SS and the weight percentage of the SS (Supplementary Fig. 4e) may also suggest that current reworking is not the primary control driving grain size changes within the silt fraction.

We address the reviewer's concerns in the following discussion.

Additionally, out of the 94 grain size (SS) measurements, fewer than 10 samples were measured from the high peaks in XRF Zr/Rb, with only one sample measured for each major peak (Supplementary Figure 4b). This limited sampling could bias the statistical analyses, as well as the correlation between Zr and mean SS and the calibration of the XRF-Zr/Rb using discrete SS measurements. To provide strong evidence that mean SS and XRF-Zr/Rb can be used as proxies for near bottom current strength in such a dynamic environment, I recommend that the authors provide additional grain size measurements at these prominent XRF Zr/Rb peaks to improve the robustness of their analysis.

We thank the reviewer #2 for his/her suggestions to improve the robustness of the grain-size analyses. We have measured 39 additional samples from the coarser part of the record, i.e. interglacial sections in the high peaks of XRF Zr/Rb. These new measurements substantially improved the correlation coefficients if one extends the traditional sortable silt range to 10-125 μm (sortable silt plus fine sand (SSFS)). This approach has been successfully applied for other strong bottom current regimes as in the central Drake Passage (Wu et al., 2021) and fine sand has been included in grain-size based current reconstructions in the site survey core (Lamy et al., 2015).

Using SS/FS yields significant correlations between mean SSFS grain-size and the weight percentage of the SSFS (Supplementary Fig. 4b; $R^2 = 0.91$). Using the new SS/FS data, we now use the current velocity calibration from the Drake Passage (Wu et al, 2021, *Nature Comm.*). While the absolute flow speed changes with this new calibration, the relative changes in strength does not, and therefore the results of our subsequent analyses remain unaffected.

Moreover, there are notable differences in the timing and amplitude of ACC current strength changes over the past 800 kyr between U1542 and records from the Drake Passage and Central South Pacific (Supplementary figure 5). The latter sites are strategically positioned within the main ACC pathway and exhibit similar changes, while U1542 does not follow this pattern. Could the proximity of U1542 to land have influenced its grain size patterns, given the variable sediment sources and sedimentary/depositional mechanisms?

While the general glacial/interglacial current strength variation patterns are similar (enhanced currents during interglacials versus reduced glacial current strength), we agree that some detailed pattern between the upper continental slope IODP Site U1542 and the deep ocean, lower-resolution sites are partly diverging. This applies to MIS 5, but also to most of the other peak interglacials, which stand out more prominently, with more abrupt increases in current strength from the overall glacial background. We speculate that the shallower water depth and the particular strength of the Cape Horn Current (CHC) might contain thresholds above which the current strength strongly increases in a nonlinear mode.

As mentioned above, the current sensitivity of some of the peak glacials may be lower compared to interglacials, most likely related to the proximal ice sheet and sea level. Nevertheless, we observe strong millennial-scale current strength peaks throughout the record.

We added discussion on this important point and the uniqueness of the CHC, partly diverging from the ACC changes in the main text and supplements (lines 212-237). The relation between CHC and ACC is discussed below in relation to various valid comments of Reviewer #3.

My second concern is the “persistent” connections proposed between U1542 records and those from the Northern Hemisphere and Antarctica (EPICA). Given the significant discrepancies in timing, duration, and amplitudes of events, it’s unclear whether a persistent interhemispheric link over the last 790kyr truly exists.

Beyond the last glacial cycle and the radiocarbon range, age model uncertainties prevent direct one-to-one comparisons between different records. Concepts like the bipolar seesaw further complicate interhemispheric comparisons. Consequently, we do not address the precise “timing” in the comparison between Sites U1542 and U1385 beyond the last glacial period.

We focused on comparing events within a single glacial cycle, separated by terminations, as they provide the most reliable markers for graphical correlation across records. Comparing the number and amplitude of events for each cycle offers a more robust approach to mitigate age model uncertainties and minimize subjective interpretation. Using the threshold method further helps to distinguish events from potential background signals, and enable to compare different proxies such as SST, current reconstructions, and planktic oxygen isotopes. Enabled by these two long proxy records (U1542 and U1835) allow for the long-term comparison of millennial-scale climate variability, the novelty of this study lies in the comparison of event amplitudes (Figure 4c-d) and frequencies (Figure 4e-f), which suggests that interhemispheric links as observed during the last glacial period (Epica Community Members, Nature 2004) persisted back over the past 790 kyr. The moderate to good positive linear correlations (Figures 4 c-f) indicates that periods of high variability in one hemisphere are reflected in the other.

We compared CHC/ACC strengthening events to atmospheric CO₂ rise events in a different manner, as both records follows a Southern Hemisphere pattern. Site U1542 and the Eastern South Pacific follow an "Antarctic timing," and both CO₂ and CHC events variability can be considered contemporaneous on millennial timescales. In Figure 6, only major CHC events are compared with 2-kyr average CO₂ variability, minimizing interpretative uncertainty. The one-to-one CHC to CO₂ comparison is thought to be speculative, but demonstrates that the magnitude of millennial-scale events and terminations differs, suggesting that factors other than ACC reacceleration also contribute to increasing the atmospheric CO₂ levels during terminations.

While I appreciate the effort in presenting such extensive high-resolution datasets, I find the explanations provided for these differences to be insufficiently robust. Much of the text is focused on reporting these discrepancies, without adequately explaining them, often offering only a single sentence to explain these differences. Below I provide some examples. I believe that refining the text and clarifying the main message would greatly improve the paper’s readability. As it stands, the current version is difficult to follow.

We thank Reviewer #2 for asking us to improve these text sections. We carefully rewrote these parts and tried to more adequately explain the reasons for discrepancies discovered during our data analyses.

-Lines 206-207: “This decoupling between the SH mid-latitudes and Antarctica likely resulted in an increase of the meridional thermal gradient at these periods”

We thank the reviewer to pointing to this sentence. We removed it, as it may have been too speculative and difficult to assess with the existing records.

-Lines 216-218: “This partly different behaviour of the bottom currents within subantarctic ACC/CHC and the SST might indicate a different response to atmospheric (e.g., SWW) and oceanic forcing (e.g., see-saw).

We rephrased and clarified this part (lines 301-308).

-Lines 277-278: “These differences might be due to varying millennial-scale sensitivities and thresholds for SST and ACC strength changes, related to various atmospheric and oceanic forcings (see below)”. However, the authors do not explain later these” various atmospheric and oceanic forcings”.

We added a more detailed section on the CHC based on more recent oceanographic studies. These new studies discuss the various forcings of the CHC, including the ACC and low-latitude atmosphere-ocean mechanisms presently controlling the ACC. We would like to refer to our new text section (lines 212-237) and the text in lines 301-308.

My last concern is regarding the mechanism proposed for the changes recorded at Site U1542. Between lines: 228-253, the authors suggest that shifts in the Southern Westerly Winds (SWW) are the primary driver of the observed changes at Site U1542. Later this mechanism is used to support the role of ACC in CO₂ exchanges, for examples in lines 468-470: “...strengthening and southward displacement of the SWW, enhancing ACC strength, decreasing Southern Ocean stratification, promoting ventilation, and consequently, increasing atmospheric CO₂ levels.

However, in their response to reviewer #2, the authors state that “the observed ACC strength variations are consistent and indicate that frontal shifts are not the main factor in determining ACC strength in our study area and the South Pacific in general. I may have overlooked something, but I find myself unclear about the specific mechanism proposed as the driver of the observed changes in their record.

Similar to the reviewer’s previous point, we would like to refer to the now improved explanation of oceanographic processes and forcings of the CHC.

Other comments

The authors should thoroughly check references in both the text and reference list. For instance, in line 146, citations like (Davies et al. 2020, Hagemann 2024) need to be converted to numerical format. Additionally, in line 807, please note that Hagemann et al. 2022 refers to a preprint, the peer-reviewed version was published in 2023.

Thank you for the careful revisions, we checked the reference list accordingly.

Line 159: “Fig 2c”. I believe the authors refer to Fig. 2b.

Thank you, done

Line 223: Please define the abbreviation “PIS”

We thank the reviewer for the careful revisions and corrected the respective text parts accordingly. PIS is now defined above (line 135)

Reviewer #3 (Remarks to the Author):

The revised version of the manuscript has improved and some of my concerns have been adequately addressed. However, I still have some concerns about some of the analysis and assumptions made in this study and some small points to clarify the manuscript a bit more.

We thank Reviewer #3 for the positive assessment of our revised manuscript.

Before I address my concerns, I have a tip for future submissions (not only for this paper). I like to clarify that it is not meant as criticism and that I did not let this influence my review of the revised manuscript. Some references in the text to figures and references to line numbers in the response are not correct. I also feel like there are more typos in the revised version. Firstly, the incorrect references make it more difficult for me to review the paper because I have to do more work to see where the revised text is, and what figure is actually meant. Secondly, it makes it feel like the revised version was a bit rushed and I think that is a shame, because it is obvious you have spent a lot of time on this study. For the future, I suggest being more thorough on the last check before submission.

Thank you for your communicating this point to us. We are sorry for the typos and other mistakes. We have now checked formatting and layout more carefully.

My main concern is still how the measurements of the Cape Horn Current (CHC) are framed as being representative of the Antarctic Circumpolar Current (ACC). I am not satisfied with the arguments given in the response. To address the arguments in the response:

(1) The authors refer to the study of Chaigneau and Pizarro (2005; CP05) where an example of a drifter (black drifter in Fig. S1) is provided that leaves the ACC around 100W and returns to the ACC through the CHC. Based on CP05 I would say there are probably a few more drifters that follow a similar trajectory. However, a few drifters do not necessarily reflect the mean flow in this region. Compared to a more recent paper (Strub et al., 2019) I would rather say they are probably not representing the mean flow accurately. I therefore believe that the quality of this paper is insufficient to argue that the CHC is a northern branch of the ACC.

We agree that the CP05 study based on drifters is not optimal to infer longer-term mean circulation dynamics, which would be most relevant for past current strength reconstructions on geological time-scales. We therefore also included satellite altimetry data from Zheng et al. (2023) in the previous version of the manuscript. However, the reviewer correctly noted that we should likewise consider the comprehensive satellite altimetry and model study by Strub et al. (2019). While Strub et al. (2019) primarily focus on the coastal ocean mainly further north (38°S-46°S), their map with geostrophic velocities (Strub et al., Fig. 1) also extends south to the CHC area to ca. 55°S. This map thus includes the CHC area and, in accordance with the drifter data, shows the strong current along within the CHC before entering the Drake Passage. This is likewise consistent with an additional recent remote sensing study in the area by Saldias et al. (2024; *Remote Sensing*).

The CP05 study is probably not conclusive in linking the CHC to the subantarctic ACC as required to characterize the CHC as a northern branch of the ACC. We now reframed our paper accordingly towards common forcing mechanisms of the CHC and the subantarctic ACC which we discuss below referring to the reviewer's points (2) and (3).

(2) The authors also refer to the study of Zheng et al. (2023). However, I believe the citation to be inaccurate. In Zheng et al. (2023) I do not think they say that the CHC is part of the ACC. What they do say is that the CHC is part of the ACC system. A subtle difference, but in my view a very important one.

(3) Other paleo papers are cited where the CHC is also referred to as being part of the ACC. I did not look thoroughly at these papers, but in my opinion, if they argue that the CHC is part of the ACC, then I think those claims are inaccurate.

We comment on points (2) and (3) together. We agree with Reviewer #3's view that the CHC cannot be strictly considered as being a part of the ACC. We therefore adapted the wording that the CHC is part

of the ACC system (lines 84-85). As such, the CHC is located in vicinity of the Northern Boundary (NB) of the ACC (after Park et al., 2019) and is in this sense connected to the northernmost ACC.

In point (3), the reviewer mentions paleo studies involving the CHC. We assume that this mainly refers to Lamy et al., 2015 (*PNAS*) based on sediment-core MD07-3128 located close to our IODP Site U1542. Lamy et al. (2015), for the first time reconstructed CHC current strength on geological time-scales. At that time little has been published on the CHC in the modern oceanographic literature. One such study is Boisvert (1969) describing the prominent poleward flow of CHC reaching to deep water depth along the continental margin (i.e. the water depth of core MD07/3128 and Site1542)

The original CHC view by Lamy et al., (2015) involved interpreting the deep reaching CHC as an important component of the “return” flow to the ACC (also described in Well et al., 2003 (*Deep-Sea Research*). The subsequent drifter study (CP05) then provided further indications of the CHC being part of the ACC “system” as the larger Southeast Pacific circulation is described in there.

The more recent remote sensing-based studies by Strub et al. (2019), Zheng et al. (2023), and Salidas et al (2024) provide a more detailed view on the CHC. Salidas et al (2024) and Zheng et al (2023) suggest that the CHC is mainly driven by the barotropic component of the pressure gradient through sea-level. A major implication for paleoceanographic current reconstructions in general (of e.g. the ACC system) is the adjustment time-scale for changing the density structure of major currents. For the open ocean ACC in the central South Pacific, sortable silt changes generally correspond to the total water transport including wind, baroclinic, and eddy-induced, as even high-resolution core scanning (*Zr/Rb* data) records still average across centuries. Lamy et al. (2024, *Nature*). Though adjustment times of the CHC may be shorter and sedimentation-rates are higher (and thus our temporal resolution of the records), we still assume that these basic assumptions are also valid for the Chilean margin.

Though beyond the resolution of our paleo records, the pronounced seasonal changes in the CHC and the northern adjacent area to beyond 40°S, described by Salidas et al. (2024), provide evidence for the involvement of large-scale atmospheric and oceanic circulation changes. During austral winter and fall, the southward meridional transport, characteristic for the CHC, extends significantly further north and retreats southward during austral spring/summer. These large seasonal changes are connected to the seasonal migration of the South Pacific Gyre and are today related to the Southern Annual Mode (e.g., Qu et al., 2019; *J. Geophys. Res. Ocean*) and thus the Southern Westerly Wind belt. For paleoceanographic time-scales, these seasonal changes imply that a northward extension of subantarctic ACC waters (probably connected to a stronger South Pacific Current (SPC)), would be consistent with a relative northward migration of the Chilean bifurcation during cold periods. Conversely, relatively warm periods would be characterised by a poleward movement of the ACC, SPC and westerly wind circulation. These mechanisms could serve as analogues both for orbital-scale (glacial-interglacial) time-scales and the short-term millennial variations discussed in our manuscript.

Based on these studies (including the PhD study by Qi Zheng (2023) providing additional more detailed information), we changed the overall framing and now discuss our data as reflecting CHC variability controlled by deep ocean forcings (i.e. ACC) and lower latitude processes. We changed and extended the text in several parts (mainly lines 77-85, lines 213-238, lines 302-309).

My main two concerns here are:

(1) The source of the waters in the CHC. In the manuscript CP05 is used to argue that the waters in the CHC are of ACC origin. However, the sea surface temperatures (SSTs) in Fig. 1 show in my opinion

that the water is more likely to be of subtropical origin, which I think is also in line with what Strub et al. (2019) show. Yes, there will be leakage and mixing from the ACC (CP05 show this in my opinion), but based on the SSTs I would argue that most of the waters are of subtropical origin.

Yes, it is possible that the SSTs at our CHC site are influenced by low latitude sources. Following the location of the altimetry-defined Northern Boundary (of the ACC, which separates it from the adjacent subtropical gyres and the SPC), we agree that the surface characteristics of the CHC are probably influenced by both subantarctic water masses approaching South America south of the “bifurcation” as also shown by Strub et al. (2019), and subtropical sources from regions north of the NB and the bifurcation. For our bottom current speed reconstruction at ~1000 m water depth, we likewise assume subantarctic water masses with Antarctic Intermediate Water formed in the Southeast Pacific relatively nearby. In addition, we might have contributions of deep slope water originating substantially further north of the SST and wind bifurcation around 40-45°S (Well et al., 2003; *Deep-Sea Research*). We added a paragraph to the text (lines 222-238).

(2) From Zheng et al. (2023) we can see that the CHC is at best 10% of the volume transport through the Drake Passage. Therefore, I find it difficult to believe that variability in the CHC is representative of ACC variability.

Though we think that 10% of the world’s largest current is still quite significant on a global scale, we agree that our wording “representative of ACC variability” might mislead some readers. We clarified that the CHC variability at geological time-scales would plausibly parallel variations of the northernmost ACC, i.e. primarily in the subantarctic zone in the South Pacific and not on a global scale. Our records show that the millennial-scale variations of CHC strength parallel millennial-scale variation of ACC strength in the Drake Passage close to the Polar Front (Fig. 3). Both records show strengthened currents during the major Antarctic warm events (Antarctic Isotope Maxima - AIMS) of the last glacial period. These observations imply that millennial-scale variations of the CHC were similar to those within the ACC south to the Polar Frontal Zone, at least in the Drake Passage.

I think the main conclusions of this paper are based on the assumption that you are measuring ACC variability in the CHC sediment core. I think this assumption is false and I am not convinced by the arguments provided. The variability could be representative of Southern Hemispheric variability, but that is not how the paper is framed. For me to recommend this paper for publication a major revision on how the paper is framed is necessary where the assumption that ACC variability is measured in the CHC sediment core is dropped.

I do want to note that I appreciate that in the text the measurements are referred to as CHC/ACC strength. I think this is well done. However, it is not done consistently throughout the full text and for all figures.

As discussed in more detail above, we agree with this general statement of Reviewer #3. We therefore, changed the overall framing and now clearly separate variability of the CHC from that of the subantarctic ACC. However, both might, at least partly, reflect similar atmospheric and oceanic forcing mechanisms. Therefore, we keep our comparisons of CHC variations to more pelagic ACC records and discuss major differences and their implications in more detail. We rewrote the relevant parts of the main text, figure captions, and supplementary information.

Major points:

- Line 69: I feel like sources 24, 25 and 26 are wrongly cited here. I must admit I only skimmed through the papers but the general message I think of all three papers is that Southern Ocean warming affects AMOC stability. None of them mention millennial scale variability in the Drake Passage throughflow or the Agulhas current (is Agulhas leakage meant here by the way?) as triggers for Dansgaard- Oeschger

events as is suggested by the main text. Why are these papers cited here/what message should these citations convey?

We agree that this sentence does not relate to the references 24, 25, 26 (Buizert et al., 2015, Oka et al., 2021, Knorr and Lohmann, 2003). As mentioned by the reviewer, these studies discuss Southern Ocean warming affecting AMOC stability. We now adapted our sentence to clearly mention the role of Southern Ocean temperature affecting AMOC, separate from the link between Agulhas leakage (Beal et al., 2015) and the Drake Passage throughflow (Wu et al., 2021), which potentially affect AMOC. We replaced Agulhas current by Agulhas leakage (lines 64, 66).

- As I understand it correctly now, there is no relation between the millennial scale SST variability and the millennial scale CHC variability? (lines 266 – 311). Do I understand this correctly?

We are sorry for the potential misunderstanding here. We rewrote this part and now state that the CHC strength events are not in all cases corresponding to SST warmings. In light of Reviewer #3's valid point that at least part of the SST signal might originate from the low latitudes (today), and the CHC strength is primarily a deeper water signal with different thresholds of change, these deviations would be easier to explain. We changed the text accordingly (line 302-309).

- Like my first review on the relation between SST events and CHC/ACC events, Fig. 6 shows to me that there is not necessarily a relation between CO₂ events and CHC/ACC events. Also here there are many events only occurring in one of the two variables, sometimes the CHC/ACC event is leading, and sometimes the CO₂ event is leading. I suggest presenting the CHC/ACC – CO₂ events in a similar fashion as the CHC/ACC – SST events.

Although we acknowledge that comparing interhemispheric variability by glacial cycle is more reliable to address age model uncertainties, we directly compared CHC/ACC strengthening events to CO₂ rise events, as both records are located in the Southern Hemisphere.

Similar to a concern from Reviewer #2, while interhemispheric comparison poses greater challenge (e.g., bipolar seesaw), Site U1542 and the Eastern South Pacific follow an "Antarctic timing," with both CO₂ and CHC events considered contemporaneous on geological timescales. In this comparison we highlight that the magnitude of millennial-scale events and glacial terminations differs, particularly for CO₂ rise. This suggests that factors other than ACC reacceleration also contribute to increase the atmospheric CO₂ levels during terminations. Following Reviewer #3 suggestion, we now present in figure 6 the CHC/ACC– CO₂ events in a similar fashion as the CHC/ACC – SST events. We keep the previous figure as a new supplementary figure to detail the correlation of millennial-scale events that are appearing in both record in less than 7 kyr (corresponding to the suborbital limit), as only major CHC events are compared with 2-kyr average CO₂ variability, which minimizes interpretative uncertainty. Although more speculative, this one-to-one CHC to CO₂ comparison suggests a correlation at millennial timescales.

- Line 492: '... our findings shed light on the impact of the DP throughflow on Atlantic circulation on millennial timescales...'. Can you explain to me what is meant with this statement? Because I did not fully get that from this study.

We rephrased this sentence to better clarify that we provide evidence that the last glacial CHC changes paralleled variations in Drake Passage throughflow and were linked to changes in Atlantic circulation (i.e., Wu et al., 2020 *Nat. comm.*), most likely extending back to ~790 kyr based on our new data.

- In view of open science, I encourage the authors to also publish the code used to make the figures and perform the analysis with.

We used published scientific software such as Analyseries (Paillard et al., 1996), and PAST (Hammer and Harper 2001, Pal Elec). The maps are realised with Ocean Data View (Schlitzer 2022, <https://odv.awi.de/>). Other figures are realised using the commercial software Grapher. We added this information to the methods.

Minor points:

- Line 71: 'abrupt changes' does not sit well with me. Abrupt changes with respect to what? I suggest changing this sentence to something like: '... understanding of millennial scale dynamics in the Southern Ocean is ...'.

Thank you, done

- Line 127: '... the major part deviates into the southward branch, the CHC.' This needs a reference.

Thank you, done

- Line 148: Use capitals for Supplementary Figure 4.

We now give a more complete equation for each step, thank you

- Line 159: 2c should be 2b I think.

Thank you, done

- Line 196: 2f should be 2g I think.

Thank you, done

- Line 209: G/I should be G/IG I think.

Thank you, done

- Line 242: 'millennial-scale changes'. Is it possible to mention in what direction these changes are? If it is possible to be specific here, then I think that is better.

We specified, mentioning here the millennial-scale strengthening in ACC strength

- Line 244: I still do not understand this sentence about the amplification of millennial-scale variability.

The amplification refers to the previous sentence and to the higher amplitude of the millennial-scale changes in more coastal oceanic settings (i.e. CHC) and likewise because of the higher sedimentation-rates. In contrast, more pelagic sediment records are either affected by smoothing through bioturbation or generally limited resolution to resolve millennial-scale climate change. We rephrased the sentence accordingly (lines 268-269).

- Line 323: 3f should be 3d I think.

Thank you, done

- Line 363: 620 kyr should be 621 kyr I think.

Thank you for your precision, done

- Line 406: From lines 266 – 311 I understood there is not really a relation between SSTs and ACC strength millennial scale variability. Here it is stated there is a relation. Can you explain this a bit more?

We rephrased to clarify this sentence (lines 485–487) which links our interhemispheric comparison (i.e., AMOC, SST and CHC) to recent modern observations. We think that this part is relevant to discuss our results with modern observations.

- Line 433: 6c should be 6d I think.

Thank you, done

- Line 439: 6c should be 6d I think.

Thank you, done

- Line 440: I suggest changing ‘parameters’ to ‘processes’.

We agree that ‘processes’ is a better fit in the sentence

- Line 513: Here Cape-Horn current is used instead of Cape Horn current. I suggest being consistent over the entire manuscript.

Thank you, done

Figures:

- Figure 1: I still disagree with calling the arrow above the Northern Boundary (NB) around 48 degrees the ACC. It falls outside of the definition of the ACC, so it should not be called the ACC in my opinion. I know that in Lamy et al. (2015) they also call the ocean circulation at this latitude the ACC, but also that is, in my opinion, not correct. I suggest calling the arrow the South Pacific Current (SPC) as in Strub et al. (2019).

We changed figures 1 and supplementary figure 1 in agreement with the oceanographic study of Strub et al., 2019.

- Figure 1: The northward current here is named the PCC. In the main text this current is called the Humboldt current (line 127). I suggest being consistent in naming this current.

Thank you, we now are consistent using Humboldt Current, commonly used in the literature.

- Figure 1: there is a typo in the caption: ‘Cape Horn Current’

Thank you, corrected

- Figure 2f: why is not a linear fit used?

During interglacials, the CHC and the DP throughflow reach a plateau at a maximum of 120% of interglacial values. In contrast, central South Pacific records exceed 160% of Holocene values during MIS 11. This indicates that Drake Passage throughflow strength reaches a maximum during interglacials and the current strength cannot increase, whereas temperatures are not constrained by such a plateau. This difference of behaviour in warmer periods likely explains why a polynomial fit performs better than a linear fit.

- Figure 2: Typo in the caption: Antarctic circumpolar current Strength.

Thank you, corrected

- Figure 4: I think subfigures (c) – (f) can be described a bit more in the caption.
- Figure S5: Is it possible to include the locations of the cores in Figure S6i?
- Figure S6: subfigure (i) is not labelled.
- Figure S6: The fronts in subfigure (i) are not mentioned in the caption.

We modified figures S5 and S6, both including now a labelled map displaying the locations of the cores and the major oceanic fronts.

- Figure S8: In the caption it is not explained what the purple, blue and orange markers represent.

Thank you, corrected

- Figure S8 and S11: the use of the colors for the markers is confusing because they have the same color as the CHC/ACC, SST, and planktic d18O lines. I suggest using different colors for the markers.

We hope the new colors of the markers will be less confusing.

- Figures 4, 5, 6 and S8: the same color (or at least very similar) is used for the planktic d18O record and the dCO₂/dt records. It might be nice to use a different color for the dCO₂/dt records for clarity.

We agree and hope that the new color better distinguish the dCO₂/dt records

References:

- Strub, P. T., James, C., Montecino, V., Rutllant, J. A., and Blanco, J. L.: Ocean circulation along the southern Chile transition region (38–46S): Mean, seasonal and interannual variability, with a focus on 2014–2016, Prog. Oceanogr., 172, 159–198, <https://doi.org/10.1016/j.pocean.2019.01.004>, 2019.

Reviewer Reports on the Third Revision

Reviewer #2 (Remarks to the Author):

The revised version of the manuscript 790,000 years of millennial-scale Cape Horn Current variability and interhemispheric linkages by Rigalleau et al., has been significantly improved and my concerns have been adequately addressed. Just a minor comment: please define ESP in line 343. I commend the authors on their extensive efforts and recommend their work for publication.

We thank the reviewer for this positive statement and the additional comments.

Reviewer #3 (Remarks to the Author):

1. Line 78: As I understand it from the sources is that the CHC originates solely from the bifurcation of the SPC into the CHC and the Humboldt Current, meaning I would not mention the ACC here. This is also mentioned in line 272.

We agree with the reviewer and do not mention the ACC anymore here.

2. Lines 214-217: This statement needs a reference.

Thank you, we added the missing reference.

3. Lines 217-220: This sentence confuses me a bit. The first part is quite clear, though I suggest rewording the sentence a bit; e.g. ‘... total water transport including wind-driven, barotropic and eddy-induced transport.’ I’m not sure what is meant in the second part. As it is worded now, it suggests to me that there are processes active now, that weren’t active in the past. However, I don’t think that is what is meant here. I suppose what is meant here is that it is impossible to determine the contribution of for example these three transport mechanisms on the total transport based on the proxies that are used. I suggest clarifying this sentence.

We agree, and reworded the text accordingly in order to show that the presently proxy data do not allow to distinguish the individual oceanographic processes.

4. Line 222: What ‘low-latitude forcings’ are meant here?

We specified what we mean with “low-latitude forcings” and added also the reference accordingly.